# Water stable isotopes spatio-temporal variability in Antarctica in 1960-2013: observations and simulations from the ECHAM5-wiso atmospheric general circulation model

5   Sentia Goursaud[1,2], Valérie Masson-Delmotte[1], Vincent Favier[1], Anaïs Orsi[2], Martin Werner[3]

[1]Univ. Grenoble Alpes, CNRS, IGE, F-38000 Grenoble, France

10   [2]LSCE (Institut Pierre Simon Laplace, UMR CEA-CNRS-UVSQ 8212, Université Paris Saclay), Gif-sur-Yvette, France

[3] Alfred Wegener Institute, Helmholtz Centre for Polar and Marine Research, Bremerhaven, Germany

15   *Correspondence to*: Sentia Goursaud (sentia.goursaud@lsce.ipsl.fr)

**Abstract.**

Polar ice core water isotope records are commonly used to infer past changes in Antarctic temperature, motivating an improved understanding and quantification of the temporal relationship between $\delta^{18}O$ and temperature. This can be achieved using simulations performed by atmospheric general circulation models equipped with water stable isotopes. Here, we evaluate the skills of the high resolution water-isotope-enabled atmospheric general circulation model ECHAM5-wiso (the European Centre Hamburg Model), nudged to European Centre for Medium-range Weather Forecasts (ECMWF) reanalysis using simulations covering the period 1960-2013 over the Antarctic continent.

We compare model outputs with field data, first with a focus on regional climate variables and, second on water stable isotopes, using our updated dataset of water stable isotope measurements from precipitation, snow and firn/ice core samples. ECHAM5-wiso simulates a large increase in temperature from 1978 to 1979, possibly caused by a discontinuity in the European Reanalyses (ERA) linked to the assimilation of remote sensing data starting in 1979.

Although some model-data mismatches are observed, the (precipitation minus evaporation) outputs are found to be realistic products for surface mass balance. A warm model bias over Central East Antarctica and a cold model bias over coastal regions explain first-order $\delta^{18}O$ model biases by too strong isotopic depletion on coastal areas and underestimated depletion inland. At the second order, despite these biases, ECHAM5-wiso correctly captures the observed spatial patterns of deuterium excess. The results of model-data comparisons for the inter-annual $\delta^{18}O$ standard deviation differ when using precipitation or ice core data. Further studies should explore the importance of deposition and post-deposition processes affecting ice core signals and not resolved in the model.

These results build trust in the use of ECHAM5-wiso outputs to investigate the spatial, seasonal and inter-annual $\delta^{18}O$-temperature relationships. We thus make the first Antarctica-wide synthesis of prior results. First, we show that local spatial or seasonal slopes are not a correct surrogate for inter-annual temporal slopes, leading to the conclusion that the same isotope-temperature slope cannot be applied for the climatic interpretation of Antarctic ice core for all time scales. Finally, we explore the phasing between the seasonal cycles of deuterium excess and $\delta^{18}O$, as a source of information on changes in moisture

sources affecting the $\delta^{18}$O-temperature relationship. The few available records and ECHAM5-wiso show different phase relationships in coastal, intermediate and central regions.

This work evaluates the use of the ECHAM5-wiso model as a tool for the investigation of water stable isotopes in Antarctic precipitation, and calls for extended studies to improve our understanding of such proxies.

**Keywords.**

Water isotopes, isotope modelling, model evaluation, mass balance, Antarctic ice sheet, paleoclimate, reconstruction

## 1. Introduction

The Antarctic climate has been monitored from sparse weather stations, providing instrumental records starting at best in 1957 (Nicolas and Bromwich, 2014). Water stable isotopes in Antarctic ice cores are key to expand the documentation of spatio-temporal changes in polar climate and the hydrologic cycle (Jouzel et al., 1997) for the recent past (PAGES 2k Consortium, 2013; Stenni et al., 2017a) as well as for glacial-interglacial variations (Jouzel et al., 2007; Schoenemann et al., 2014). Water stable isotopes measured along ice cores were initially used to infer Antarctic past temperatures using the spatial isotope-temperature slope (Lorius et al., 1969). The focus on inter-annual variations is motivated by the goal to quantify temperature changes at the Earth's surface, including Antarctica, during the last millennia, to place current changes in the perspective of recent natural climate variability (Jones et al., 2016), to understand the drivers of this variability, and to test the ability of climate models to correctly represent it. This timescale is relevant for the response of the Antarctic climate to e.g. volcanic forcing, and for the Antarctic climate fingerprint of large-scale modes of variability such as ENSO and the Southern Annular Mode (Smith and Stearns, 1993; Turner, 2004; Stammerjohn et al., 2008; Schroeter et al., 2017). The various climate signals potentially recorded in precipitation isotopic composition are, however, difficult to disentangle.

First, the original signal from precipitation may be altered due to deposition and post-deposition processes (e.g. Sokratov and Golubev, 2009; Jones et al., 2017; Münch et al., 2017; Laepple et al., 2018). Wind erosion and sublimation during or after precipitation have long been known to affect ice core records (Eisen et al., 2008; Grazioli et al., 2017). Other processes such as melt and diffusion processes can also alter the preservation of isotopic signals in firn and ice and cause smoothing of the initial snowfall signals (Johnsen, 1977; Whillans and Grootes, 1985; Johnsen et al., 2000; Jones et al., 2017). So far, the mechanisms of such post-deposition processes on the alteration of the initial precipitation signals are not fully understood and quantified (Touzeau et al., 2017). Second, the Antarctic snowfall isotopic composition may be affected by the origin of moisture and the associated evaporation conditions, or by changes in the relationship between condensation and surface temperature, as well as by changes in the intermittency of precipitation (e.g. Sime et al., 2009; Hoshina et al., 2014; Touzeau et al., 2016). Although the surface snow isotopic composition signal has classically been interpreted as a precipitation-weighted

deposition signal (Krinner and Werner, 2003), recent studies evidenced isotopic exchanges between the Antarctic snow surface and the atmosphere associated with snow metamorphism occurring at the diurnal and sub-annual scales (Steen-Larsen et al., 2014; Casado et al., 2016; Ritter et al., 2016; Touzeau et al., 2016).

5   Second, the climatic interpretation of water stable isotopes in Antarctic ice cores is still challenging. Quantitative approaches have relied on empirical relationships as well as the use of theoretical and atmospheric models including water stable isotopes. Pioneer studies evidenced a close linear relationship between the spatial distribution of water stable isotopes and local temperature (e.g. Lorius and Merlivat, 1975), and explained this feature as the result of the distillation along air mass trajectories. Thereupon, 10  local temperature (i.e. at a specific site) was reconstructed using $\delta^{18}O$ measurements and based on the slope of the aforementioned spatial empirical relationship, as a surrogate for relationships at annual to multi-annual scales. However, recent data syntheses have shown that other effects had to be taken into account (e.g. Masson-Delmotte et al., 2008): It was found that the Antarctic snowfall isotopic composition is also linked to the initial vapour isotopic composition (Stenni et al., 2016), atmospheric transport 15  pathways (Schlosser et al., 2008; Dittmann et al., 2016), Antarctic sea ice extent (Bromwich and Weaver, 1983; Noone and Simmonds, 2004; Holloway et al., 2016) and local condensation temperature, itself related to surface temperature through complex boundary layer processes (Krinner et al., 2007). Evaporation conditions, transport and boundary layer processes may vary through time, from seasonal (Fernandoy et al., 2018) to annual or multi-annual scale, thereby potentially distorting the quantitative 20  relationship between snow isotopic composition and local surface air temperature estimated empirically for present day conditions (Jouzel et al., 1997).

Model studies have been key to explore quantitatively the spatio-temporal aspects of the relationships between precipitation isotopic composition and temperature (Jouzel et al., 2000). Mixed cloud isotopic models have been used to propose a coherent interpretation of $\delta^{18}O$ and $\delta D$ data in terms of changes in 25  site and source temperatures (Uemura et al., 2012), or to simulate isotopic variations along individual atmospheric trajectories (Dittmann et al., 2016). However, such theoretical distillation models rely on the closure assumption at the ocean surface to calculate the initial evaporation isotopic composition, and do not account for atmospheric dynamics and mixing of air masses (Jouzel and Koster, 1996; Delmotte et

al., 2000). Atmospheric general circulation models equipped with water stable isotopes offer a physically coherent, three-dimensional framework to investigate the weather and climate drivers of Antarctic precipitation isotopic composition (Jouzel et al., 2000). They play a key role in assessing how different boundary conditions (e.g. changes in orbital forcing, changes in atmospheric greenhouse gas concentration) affect the simulated relationships between precipitation isotopic composition and climate variables. Most of these simulations support that the present-day isotope-temperature spatial relationship is a good approximation for the relationships between glacial conditions and today (Delaygue et al., 2000; Werner et al., in prep.), with one exception (Lee et al., 2008). One study used climate projections in response to increased atmospheric $CO_2$ concentration to explore isotope-temperature relationships in a world warmer than today, and suggested a changing temporal isotope-temperature relationship, due to changing covariance between temperature and precipitation (Sime et al., 2009). Several observational and modelling studies have also evidenced different isotope-temperature relationships between the spatial relationship and those calculated at the seasonal (Morgan and van Ommen, 1997), or at the inter-annual scale (Schmidt et al., 2007).

Our study is motivated by the need for a synthesis over all of Antarctica, using a proper interpretation of processes that affect water stable isotopes on the appropriate spatial and temporal scales. It aims to address the following questions: (i) what is the performance of a state-of-the-art atmospheric general circulation model with respect to existing Antarctic observations of spatio-temporal variations in temperature, surface mass balance, precipitation and snow isotopic composition for present-day? (ii) what can we learn from such a model for the regional relationships between isotopic composition from the precipitation and temperature at the inter-annual scale for the recent past, and considering all of Antarctica?

For this purpose, we focus on the high resolution atmospheric general circulation model equipped with water stable isotopes ECHAM5-wiso (the European Centre Hamburg model) which demonstrated remarkable skills for Antarctica (Werner et al., 2011). We explore a simulation performed for the period 1960-2013, where the atmospheric model is nudged to the European Reanalyses (ERA) ERA-40 and ERA-interim reanalyses (Uppala et al., 2005), ensuring that the day-to-day simulated variations are coherent with the observed day-to-day variations in synoptic weather and atmospheric circulation (see Butzin et al., 2014 for more explanation). This framework is crucial to perform comparisons between

simulations and observations for temporal variations. Second, we compile a database of precipitation, snow, and firn/ice isotopic composition, using data from precipitation sampling and ice core records, considering $\delta^{18}O$ and deuterium excess (hereafter, d). These methods are described in Section 2. We then compare the model outputs with the available datasets (Section 3). After evaluating the near-surface temperature and the surface mass balance (hereafter SMB) (Section 3.1), we focus on the water stable isotopes (Section 3.2). We emphasize spatial patterns, the magnitude of inter-annual variability (Sections 3.2.1 and 3.2.4), the pattern and the amplitude of seasonal variations (Section 3.2.2 and 3.2.4). We explore the simulated and estimated isotope-temperature relationships (Section 3.2.3) and the relationships between d and $\delta^{18}O$ (Section 3.2.3). Highlighting the strengths and limitations of the model (Section 3.3), we use the simulation framework to explore the $\delta^{18}O$-temperature relationship (Section 4.1) and the phase lag between seasonal variations in d and $\delta^{18}O$ (Section 4.2). Finally, we focus on the implications of our results for the climatic interpretation of water stable isotope records for seven Antarctic regions (central plateau, coastal Indian, Weddell Sea coast, West Antarctic ice sheet, Victoria Land and Droning Maud Land regions). The Antarctica2k group (Stenni et al., 2017b) indeed identified these seven Antarctic regions, which are geographically and climatically consistent, to produce regional temperature reconstructions using ice core records. The results of our study thus contribute to the reconstruction of past Antarctic climate spanning the last 2000 years (the Antarctica2k initiative) of the Past Global Changes (PAGES) PAGES2K project (PAGES 2k Consortium, 2013), by providing quantitative calibrations of the regional temperature reconstructions using ice core water stable isotope records.

## 2. Material and methods

### 2.1 Observations and reanalysis products

### 2.1.1 Temperature and surface mass balance instrumental records

Station temperature records have been extracted from the READER database (https://legacy.bas.ac.uk/met/READER) (Turner et al., 2004). We have selected surface stations data following two conditions: to cover the 7 Antarctic regions aforementioned (see section 1 and Fig. 1) with at least one station for each, and to cover the period 1960-2013. As a result, we have selected Neumayer,

Mawson, Vostok, Casey, Dumont d'Urville (hereafter DDU), McMurdo, Palmer and Esperanza station surface data. Due to the short duration of surface station records for the 90-180° W sector, we have added data from the automatic weather station (hereafter, AWS) of Dome C, but we have used it with caution as these records are associated with a warm bias in thermistor measurements due to solar radiation when the wind speed is low (Genthon et al., 2011). Finally, we extracted the reconstruction of temperature for Byrd station by Bromwich et al. (2013), based on AWS data and infilled with observational reanalysis data. No record meets our criteria for the Weddell Sea coast region (Fig. 1).

SMB data have been extracted from the quality-controlled GLACIOCLIM-SAMBA (GC) database (Favier et al., 2013). We have selected data spanning the twentieth century, corresponding to 3242 punctual values, which have been then clustered within the corresponding ECHAM5-wiso grid cells for calculation of gridded annual average values. As described by Favier et al. (2013), the spatial coverage of SMB field data is particularly poor in the Antarctic Peninsula, in West Antarctica and along the margins of ice sheet. As a result, SMB is not correctly sampled at elevations between 200 and 1000 m a.s.l., where accumulation rates are the highest. In central Antarctica, areas characterized by wind glaze and megadunes are also insufficiently documented.

### 2.1.2 ERA reanalyses

The ECHAM5-wiso model run for this study is nudged to ERA-40 (Uppala et al., 2005) and ERA-interim (Dee et al., 2011) global atmospheric reanalyses produced by the European Centre for Medium-Range Weather Forecasts (ECMWF). ERA-40 covers the period 1957-2002 at a daily resolution, with a spatial resolution of 125 km x 125 km. ERA-Interim covers the period 1979 to present at a 6-hourly resolution, and with a spatial resolution of 0.75° x 0.75°.

For comparison with instrumental records and ECHAM5-wiso outputs, we have extracted 2-meter temperature outputs (hereafter 2m-T) over the periods 1960-1978 and 1979-2013 for ERA-40 and ERA-Interim, respectively, at grid cells closest to the stations where meteorological measurements have been selected (see previous section). We have then calculated annual averages.

### 2.1.3 A database of Antarctic water stable isotopic composition from precipitation, surface snow and firn/ice core records

This database consists of water stable isotope measurements performed on different types of samples (precipitation, surface snow or shallow ice cores), and at different time resolutions (sub-annual, annual or multi-annual average values) (see S1 in Supplementary Material). Sample data consist of $\delta^{18}O$ and/or $\delta D$, providing d, if both $\delta^{18}O$ and $\delta D$ have been measured. Altogether, we have gathered data from:

- (1) 101 high resolution ice core records, including 79 annually resolved records, and 18 records with sub-annual resolution (including 5 records with both $\delta^{18}O$ and $\delta D$ data). These data have been extracted from the Antarctica2k data synthesis (Stenni et al., 2017b) with a filter for records spanning the interval 1979-2013, thus restricting the original 122 ice cores to a resulting 101 ice cores data. Primary data sources, geographical coordinates and covered periods are reported in Supplementary Material Table S1.

- (2) average surface snow isotopic composition data compiled by Masson-Delmotte et al. (2008) (available on [http://www.lsce.ipsl.fr/Phocea/Pisp/index.php?nom=valerie.masson](http://www.lsce.ipsl.fr/Phocea/Pisp/index.php?nom=valerie.masson)), expanded with datasets from Fernandoy et al. (2012); in this case, the averaging period is based on different time periods, with potential not continuous records (see Supplementary Material Table S1).

- (3) precipitation records extracted from the International Atomic Energy Agency / Global Network of Isotopes in Precipitation (IAEA / GNIP) network (IAEA/WMO, 2016) with monthly records available for 4 Antarctic Stations, complemented by daily records for 4 Antarctic stations from individual studies. Precipitation records from Vostok are available but are excluded from our analysis, due to an insufficient number of measurements (29). See orange part of Table S1 in Supplementary Material.

Each of the 1205 locations have given an individual index number. Data have been processed to calculate time-averaged values (available at 1089 locations for $\delta^{18}O$ values, 879 locations for $\delta D$ and 770 locations for d). The ice core records with sub-annual resolution were averaged at annual resolution over the period 1979-2013, resulting in 88 ice core records for $\delta^{18}O$ and only 5 for d. Most precipitation records are not continuous and do not cover a full year, preventing the calculation of annual mean values. We have also used sub-annual records from 22 highly resolved ice cores (including 18 records giving access to $\delta^{18}O$

and 5 records giving access to d) and precipitation sampling from 8 stations to characterise the seasonal amplitude. For ice core records, we have only calculated the yearly amplitude from available measurements, as chronologies cannot be established at monthly scales. Note that this database is publically available on the PANGAEA data archive (https://www.pangaea.de/?t=Cryosphere).

## 2.2 ECHAM5-wiso model and simulation

The atmospheric general circulation model (AGCM) ECHAM5-wiso (Roeckner et al., 2003; Werner et al., 2011) captures the global pattern of precipitation and vapour isotopic composition, including the spatial distribution of annual mean precipitation isotopic composition over Antarctica (Masson-Delmotte et al., 2008). Several studies using ECHAM5-wiso have been dedicated to model-data comparisons for temporal variations in other regions (e.g. Siberia, Greenland) (Butzin et al., 2014; Steen-Larsen et al., 2016).

The ECHAM5-wiso outputs analysed in this study consists of daily values simulated over the period 1960-2013. ECHAM5-wiso was nudged to atmospheric reanalyses from ERA-40 (Uppala et al., 2005) and ERA-interim (Dee et al., 2011), which are shown to have good skills for Antarctic precipitation (Wang et al., 2016), surface pressure fields, and vertical profiles of winds and temperatures. The ocean surface boundary conditions (sea-ice included) are also prescribed based on ERA-40 and ERA-interim data. Isotope values of ocean surface isotopic composition are based on a compilation of observational data (Schmidt et al., 2007). The simulation was performed at a T106 resolution (which corresponds to a mean horizontal grid resolution of approx. 1.1 ° x 1.1 °) with 31 vertical model levels.

## 2.3 Methods for model-data comparisons

In the model, we have extracted specific daily variables for comparison with available data, and then averaged it. We have extracted daily 2-m temperature outputs (hereafter 2m-T) for comparison with surface air instrumental records, daily (precipitation minus evaporation) outputs (hereafter P-E) for comparison with SMB data, and daily precipitation isotopic composition outputs for comparison with measurements of isotopic composition data in the precipitation. For ice core data, we averaged daily precipitation isotopic composition weighted by the daily amount of precipitation.

For each specific site, we selected the model grid cell including the coordinates of the site. When comparing model outputs with the database of surface data (time-averaged SMB and isotopic composition), available data have been averaged within each model grid cell.

Time selection was dependent on the variables. The 2-m T outputs have been compared with temperature records for the period 1960-2013, based on annual averages and selecting same years as in the data (see Section 3.1.1). The comparison with other datasets (SMB, snow and water stable isotopes from firn/ice cores) is restricted to the period 1979-2013, due to concerns about the skills of the reanalyses used for the nudging prior to 1979 in Antarctica (see next section). Daily (P-E) outputs were all extracted over the whole period 1979-2013 and averaged (see Section 3.1.2). For comparison with the surface isotopic database (Section 3.2.1), daily precipitation isotopic composition was averaged by weighting by the daily amount of precipitation over the whole period 1979-2013. For the inter-annual variability (same Section) or annual values (e.g. for d outputs, see Section 4), daily precipitation isotopic composition was averaged by weighting by the daily amount of precipitation for each year of the period 1979-2013. For sub-annual isotopic composition, we used precipitation isotopic compositions (amplitude and mean seasonal cycle) and highly resolved ice cores (amplitude only). Precipitation isotopic composition data consist of a very small number of measurements, sometimes taken before 1979 (e.g. observations from DDU consist of 19 measurements during 1973), and thus model precipitation isotopic composition outputs were extracted at the very exact sampling date. Then, monthly averages were performed and mean seasonal cycles were calculated. The resulting mean seasonal cycles of precipitation isotopic composition were obtained the same way in both precipitation data and the model. For comparison with the mean seasonal amplitude of the highly resolved ice cores, the mean seasonal amplitude was calculated from the mean seasonal cycle based on the monthly averages (weighted by the precipitation amount) over the period covered by the ice core record. .

Finally, for the spatial linear relationships, calculations reported for each grid cell are based on the relationship calculated by including the 24 grid cells ($\pm$ 2 latitude steps. $\pm$ 2 longitude points) surrounding the considered grid cell.

Our comparisons are mainly based on linear regressions. Note that through all the manuscript, we consider a linear relationship to be significant for p-value<0.05.

### 3. Model skills

In this section, we assess ECHAM5-wiso skills, with the perspective to use the model outputs for the interpretation of water stable isotope data. In polar regions, isotopic distillation is driven by fractionation occurring during condensation, itself controlled by condensation temperature (Dansgaard, 1964). We thus first compare ECHAM5-wiso outputs with regional climate records, as this comparison may explain potential isotopic biases. This includes a comparison with reanalyses, in order to explore the role of nudging in model-data mismatches. We then compare ECHAM5-wiso outputs with our isotopic database.

### 3.1 Temperature and surface mass balance

### 3.1.1 Comparison with instrumental temperatures records and ERA outputs

We compare time-series of instrumental temperature records (filled circles and dashed lines, Fig. 2) with model outputs (solid lines, Fig. 2), from 1960 to 2013. This comparison first highlights local offsets between observed and simulated mean values at each site, without a systematic overall warm or cold bias. Table 1 reports the statistical analysis of annual differences between observations and simulations (observed mean, mean difference between the data and the model outputs, observed versus simulated standard deviation). ECHAM5-wiso has a cold bias for 7 out of 10 stations. While this bias is less than 2°C for Dronning Maud Land (Mawson and Neumayer) and over the Peninsula (Palmer and Esperanza), it reaches 7°C for the Coastal Indian region (Casey and Dumont d'Urville) and is very strong over the Victoria Land region (McMurdo), reaching 15°C. This cold bias may be due to the model resolution and the location of coastal stations in the ice-free region, where the small-scale topographic features are not accounted for at the model resolution. In contrast, ECHAM5-wiso has a warm bias for all the stations located inland (Vostok, Dome C and Byrd). Werner et al (2011) also reported this warm bias for the central Antarctic plateau, and suggested that it could be linked to problems in simulating correctly the polar atmospheric boundary layer. Our comparison also shows that the simulated inter-annual temperature variability is larger than observed for 7 out of 9 sites, and particularly overestimated for locations such as DDU, Mc Murdo and Palmer, where the cold bias is large.

Figure 2 depicts a sharp simulated increase in temperature from 1978 to 1979 for all stations, except for the Peninsula region (Esperanza and Palmer). Such a feature is not displayed in instrumental records, with

one exception, at McMurdo (Fig. 2). As a result, the model-data correlation coefficient for McMurdo is higher over 1960-2013 than over 1979-2013 (Table 2), possibly because it is dominated by the sharp increase just prior to 1979. For all other stations, the correlation coefficient is significantly higher in 1979-2013 than in 1960-2013. In order to assess whether ECHAM5-wiso reproduces the temperature bias displayed by ERA-40 (Bromwich et al., 2007), we compare outputs from ERA-40 and ERA-interim (green bars, Fig. 3) with ECHAM5-wiso outputs (purple bars, Fig. 3) nudged by these reanalyses (i.e. over 1960-1978 and 1979-2013, respectively) and with the station temperature data (horizontal black lines, Fig. 3).

All data sets reveal a cold bias simulated by both the reanalyses and ECHAM5-wiso at all stations but Byrd and Vostok over the two periods (only over 1960-1978 for Neumayer and Esperanza), but this bias is larger over the period 1960-1978 compared to the period 1979-2013. This finding supports our earlier suggestion for Dumont d'Urville (Goursaud et al., 2017) that the 1978-1979 shift simulated by ECHAM5-wiso arises from the nudging to ERA-40 reanalyses. We note that mean values and the amplitude of inter-annual variations are different for ECHAM5-wiso and ERA (not shown), as expected from different model physics, despite the nudging technique. This finding has lead us to restrict, as possible, the subsequent analysis of the ECHAM5-wiso outputs to the period 1979-2013.

For this period, marked by small temperature variations, we note that the correlation coefficient between data and model outputs (Table 2) is very small for McMurdo (r=0.2) and rather small for Vostok (r=0.6), questioning the ability of our simulation to resolve the drivers of inter-annual temperature variability at these locations. We observe that the model reproduces the amplitude of inter-annual variations, with a tendency to underestimate the variations as shown by model-data slopes from 0.6 to 1°C per °C. As a result, ECHAM5-wiso underestimates the magnitude of inter-annual temperature variability for these central regions of the West and East Antarctic ice sheet. It will therefore be important to test whether similar caveats arise for water isotopes.

### 3.1.2 Comparison with GLACIOCLIM database accumulation

For each grid cell where at least one stake record is available, we have calculated the ratio of the P-E values (which we use as a surrogate for accumulation) simulated by ECHAM5-wiso to the averaged SMB

estimate for that grid region based on stake measurements (Fig. 4a). Due to the limited number of grid cells containing SMB data points from 1979 to 2013 (100 cells), located almost only on the East Antarctic Ice Sheet, we have decided to use the dataset covering the entire twentieth century (521 cells) spread over the continent.

The spatial distribution of SMB is well captured by ECHAM5-wiso, with decreasing SMB values from the coast to the interior plateau (Fig. 4a). However, the model quantitatively shows some discrepancies when compared with the GC database. The area-weighted (by the model grid cells) mean GC SMB is 141.3 mm w.e. $y^{-1}$ while the simulated area-weighted mean P-E over the same model grid is 126.6 mm w.e. $y^{-1}$. This underestimation covers 69.7% of the compared areas. The 30.3% remaining areas associated

with an overestimation of the model are located in sparse regions like in the north of the plateau, and over coastal areas (Fig. 4b). Note that the low P-E rates over the plateau (75 mm w.e. $y^{-1,}$ see Fig. 4a) counterbalances the local overestimation at the coast, supporting the ability of ECHAM5-wiso to resolve the integrated surface mass balance for the Antarctic ice sheet. Figures 4c and 4d confirms the global underestimation by the model, with slopes of simulated P-E against GC SMB lower than 1. This aspect

is emphasized for elevations higher than 2200 m a.s.l. (r= 0.74 and rmse=122.8 mm w.e. $y^{-1}$ for elevation lower than 2200 m a.s.l., and r =0.83 and rmse=55 mm w.e. $y^{-1}$ for elevations higher than 2200 m a.s.l., with "r" the correlation coefficient, and "rmse" the root mean square error). The correlation coefficient (considering all elevations) is 0.79, reflecting the non-homogenous bias over the whole continent. This can be due first to a failure in the representativity of SMB spatial variability, when averaging GC data

within ECHAM5-wiso grid cells, due to a too small number of point measurements. Second, the model grid resolution may be too coarse to reproduce coastal topography and thus associated amounts of precipitation. Finally, several key processes such as the blowing snow erosion and deposition are not taken into account into the model. For instance, the lowest value from the GC database is -164 mm w.e. $y^{-1}$, measured at the Bahia del Diablo glacier, a small glacier covering important elevation ranges in a

narrow spatial scale between the front and the summit. It was the only one within the corresponding model grid cell, so the resulting GC value within this grid cell could not be representative of the model scale, and vice versa the simulated P-E value is not representative of this small glacier-wide value.

When considering the whole Antarctic grounded ice sheet, the area-weighted P-E simulated by the model amounts to 164.4 mm w.e. y$^{-1}$. This value falls within the highest values of the 11 simulations displayed by Monaghan et al. (2006), varying from 84 to 188 mm w.e. y$^{-1}$. However, the high range of values between the different simulations illustrates the uncertainties related to the SMB model, mainly due to model resolution which is crucial to reproduce the impact of topography on precipitations and to non-resolved physical processes (e.g. drifting snow transport, including erosion, deposition and sublimation of drifting snow particles, and clouds microphysics) (Favier et al., in press). Moreover, this simulated value is very close to the best estimations of Antarctic grounded ice sheet SMB, which range between 143.4 mm w.e. y$^{-1}$ (Arthern et al., 2006) and 160.8 mm w.e. y$^{-1}$ (Lenaerts et al., 2012). This simulated value is also very close to the one obtained by Agosta et al. (2013) for the LMDZ4 model over the period 1981-2000 (160 mm w.e. y$^{-1}$), but slightly lower than with the SMHiL model forced by LMDZ4 (189 mm w.e. y$^{-1}$).

To conclude, albeit the ECHAM5-wiso simulation presented in this study has a relatively coarse resolution (110 km x 110 km compared to 15 km x 15 km for the SMHiL model forced by LMDZ4), and does not resolve processes contributing in the SMB (e.g. drifting snow processes), the P-E outputs are realistic products when compared with SMB data.

### 3.2  Comparison with water stable isotope data

Limited by the availability of the data, we could only study model skills with respect to spatio-temporal patterns, including seasonal and inter-annual variations, as well as for the simulated relationships between $\delta^{18}O$ and temperature. We have also extended the model-data comparison to the second-order parameter, d.

### 3.2.1 $\delta^{18}O$ time-averaged values and inter-annual variability

The model-data difference of the time-averaged values is positive for 88% of all grid cells, suggesting a systematic underestimation of isotopic depletion by ECHAM5-wiso (Fig. 5a). The few areas where ECHAM5-wiso overestimates the isotopic depletion are restricted to coastal regions. This pattern is coherent with the temperature anomalies: ECHAM5-wiso produces too low isotopic values where

ECHAM5-wiso has a cold bias, likely causing too strong distillation towards coastal areas, and too high isotopic values inland, where the warm bias limits the distillation strength. The statistical distribution of model-data $\delta^{18}O$ differences (not shown) shows a wide range but an interquartile range (50% of all values) of 1.4 to 3.9 ‰, therefore within 1.3 ‰ from the median. We conclude that, beyond the systematic offset

linked to climatic biases, ECHAM5-wiso correctly captures the spatial gradient (continental effect) of annually averaged $\delta^{18}O$ data. These results also suggest that the spatial distribution of annual mean $\delta^{18}O$ values from shallow ice cores is driven by transport and condensation processes well resolved by ECHAM5-wiso, with probably secondary effects of non-resolved processes such as snow drift, wind erosion, and snow metamorphism. The largest deviations are encountered in coastal regions, where the

model resolution is too low to resolve correct topography, advection and boundary layer processes (e.g. small scale storms, katabatic winds). Katabatic winds also have the potential to enhance ventilation-driven post-deposition processes (Waddington et al., 2002; Neumann and Waddington, 2004).

$\delta^{18}O$ inter-annual standard deviation is underestimated by the model for 92% of the 179 grid cells where this comparison can be performed (Fig. 5b).The interquartile range of the ratio between the simulated and

observed standard deviation varies from 0.4 to 0.6 (not shown), with an underestimation by a factor of 2 for about 50% of the grid cells.  No such underestimation of inter-annual standard deviation was identified for the simulated temperature.

We now focus on our model-data comparison on precipitation data. Both precipitation isotopic composition and temperature measurements are available for only 8 locations, and for short time periods

(Table 3). These data evidence the altitude and continental effect with increased isotopic depletion from Vernadsky (averaged $\delta^{18}O$ of -9.9 ‰) to Dome F (averaged $\delta^{18}O$ of -61.3 ‰). For 5 out of the 8 records, the isotopic depletion is stronger in ECHAM5-wiso than observed (Dome C included). The observations depict an enhanced inter-daily $\delta^{18}O$ standard deviation for inland sites, from 3.1 ‰ at Vernadsky to 10.8‰ at Dome F. The simulated $\delta^{18}O$ inter-daily standard deviation is 1.1 to 3.8 times larger than observed,

ranging from 5.1 to 19.2 ‰. For the exact same time period corresponding to the short precipitation isotopic records, ECHAM5-wiso simulates colder than observed temperatures at all stations but at Dome F and Dome C, i.e. over the plateau. This finding is consistent with results from ice core records reported previously, and consistent with the isotopic systematic biases. From this limited precipitation dataset,

there is no systematic relationship between model biases for temperature (mean value or standard deviation) and for $\delta^{18}O$, in contrast with the outcomes of the model-data comparison using the whole dataset, including surface snow. At Dome C, ECHAM5-wiso underestimates the standard deviation of temperature, but strongly overestimates the standard deviation of $\delta^{18}O$.

As a conclusion, while $\delta^{18}O$ time-averaged model-data biases are consistent with temperature biases using all the dataset, no systematic relationship emerge between model biases for temperature and $\delta^{18}O$ measured in precipitation.

### 3.2.2 $\delta^{18}O$ seasonal amplitude

High-resolution $\delta^{18}O$ data allow us to explore seasonal variations. This includes 18 ice core records with
sub-annual resolution, 4 IAEA / GNIP monthly precipitation datasets and 4 daily precipitation monitoring records.

In order to quantify post-deposition effects in ice cores, we calculated the ratio of the three first seasonal amplitudes by the mean seasonal amplitude in sub-annual ice cores (See Supplementary Information S2). We find a mean ratio of $1.40 \pm 0.47$. We explored whether this ratio was related to annual accumulation
rates (See Supplementary Information S3), without any straightforward conclusion. We also observe that five ice cores depict a ratio lower than 1, including one with a mean yearly accumulation of 15 cm w.e. y$^{-1}$, a feature which may arise from inter-annual variability in the precipitation seasonal amplitude or in post-deposition processes. This empirical analysis shows that a loss of seasonal amplitude due to post-deposition processes is likely in most cases, with an average loss of the seasonal amplitude of
approximately 70% compared to the amplitude recorded in the upper part of the firn cores (first three years).

We have calculated the mean of the $\delta^{18}O$ annual amplitude (i.e. maximum – minimum values within each year) in ice core records (triangles in Fig. 7a) and the mean seasonal amplitude of precipitation time series (circles in Fig. 7a) for comparison with ECHAM5-wiso outputs (Fig. 7, Table 4). Unfortunately, a too
small number of measurements (19 daily measurements) were monitored at DDU, preventing from the representation of the full seasonal cycle. The data depict the largest seasonal amplitude in the central Antarctic plateau, reaching up to 25.9 ‰ at Dome F. ECHAM5-wiso underestimates the seasonal

amplitude (by 14 to 69%) when compared to precipitation data, but overestimates the seasonal amplitude when compared to ice core data (from 11 to 71% ). The overestimation when comparing with ice core data is consistent with the attenuation of signal by post-deposition effects (as aforementioned) rather than a model bias.

The simulated mean seasonal $\delta^{18}O$ amplitude increases gradually from coastal regions to central Antarctica (more than 15 ‰ and up to 25 ‰ for some areas) (Fig. 7a and Fig. 8c, solid lines). The model-data comparison suggests that this pattern is correct, and that the model may underestimate the inland seasonal amplitude. As previously reported for annual mean values, systematic offsets are also identified for seasonal variations, with a systematic overestimation of monthly isotopic levels inland (e.g. for Dome

C and Dome F), and a systematic underestimation on the coast (e.g. for Vernadsky and Halley). The model-data mismatch is largest during local winter months.

Minima are observed and simulated in winter (May-September) at most locations, except for Rothera and Vernadsky where the data show a minimum in July but the model produces a minimum in late autumn (April). Maximum values are observed and simulated in local summer (December-January); a secondary

maximum is also sometimes observed and simulated in late winter (August/September). Data from Marsh station show maxima in January, April and August, whereas the model only produces a single summer maximum value.

In summary, we report no systematic bias of the seasonal temperature amplitude (Fig. 8a). The seasonal pattern for the temperature is similar compared to $\delta^{18}O$, with minima in winter and largest model-data

mismatch in winter. Secondary minima or maxima cannot be discussed with confidence, as they have low amplitudes. We also highlight that model-data offsets are larger in winter. Note that precipitation and d seasonal cycles are described in Section 3.2.4.

### 3.2.3 $\delta^{18}O$ – T relationships

Table 5 reports the temporal $\delta^{18}O$ – T relationships established from precipitation and temperature

observations, and those simulated by ECHAM5-wiso. This calculation is based on daily or monthly values (depending on the sampling resolution), and includes seasonal variations. The data display significant linear relationships for all sites but Marsh (p-value=0.07), with an increased strength of the correlation

coefficient from the coast (e.g. r= 0.38 at Rothera) to the East Antarctic plateau (e.g. r= 0.88 at Dome F). The lowest slopes are identified in the Peninsula region, with a mean slope of 0.32 ‰ °C$^{-1}$ for Rothera and Vernadsky, while the highest slopes occur over the East Antarctic Plateau, with a mean slope of 0.68 ‰ °C$^{-1}$ for Dome C and Dome F. These temporal slopes appear mostly lower than spatial slopes and those

expected a Rayleigh distillation with a single moisture source (typically 0.8 ‰ °C$^{-1}$).

In the ECHAM5-wiso model, as for the data, the simulated isotope-temperature relationship is statistically significant for all sites but Marsh (p-value=0.06). However, correlation coefficients are very small for Rothera and Vernadsky which are thus excluded from further analyses. In the simulation, correlation coefficients are the highest for Halley, Dome C and Dome F (up to 0.55), and the lowest for Neumayer

(as low as 0.29). The slope is the lowest at Neumayer, with a value of 0.29 ‰ °C$^{-1}$, increases at Halley with a value of 0.48 ‰ °C$^{-1}$, and is the highest over the plateau with values of 0.70 ‰ °C$^{-1}$ at Dome C and up to 0.94 ‰ °C$^{-1}$ at Dome F.

To summarize, ECHAM5-wiso tends to underestimate the strength of the isotope-temperature relationship, but correctly simulates a larger strength of the correlation in the central Antarctic Plateau

compared to coastal regions. There are significant differences in the isotope-temperature slopes for both coastal and central plateau locations. While there is some agreement (e.g. for for Dome F and Halley), the model produces also non-realistic slopes, with for instance, a much larger slope than observed at Dome C.

### 3.2.4 The δD-δ$^{18}$O relationship and d patterns

The δ$^{18}$O- δD linear relationship is expected to be affected by different kinetic fractionation processes, for instance associated with changes in evaporation conditions. We first compare the δ$^{18}$O- δD linear relationship in the available precipitation and ice core data, and simulated by ECHAM5-wiso (Table 6). Significant correlation is observed for all observational datasets but Marsh, as expected from meteoric samples, assuming correct preservation of samples and accurate isotopic measurements. We stress that

the smallest correlation coefficient is identified at Vernadsky (r=0.96), suggesting potential artifacts for this record. In the observations, the δD-δ$^{18}$O slope varies across regions. While slopes higher than for the global meteoric water line (i.e. >8 ‰ ‰$^{-1}$) are identified at DDU and in Dronning Maud Land, lower

slopes are identified in the Antarctic Peninsula (6.6 to 7.0 ‰ ‰$^{-1}$) and in the central East Antarctic plateau (6.5 and 6.4 ‰ ‰$^{-1}$ at Dome C and Dome F respectively). In the model, outputs also display significant linear relationships. They show higher values of the slope than observed in the Antarctic Peninsula, at DDU, and at Dome F, and lower than observed for the other regions, including Dome C. These results appear coherent with associated coastal versus inland temperature and isotopic distillation biases.

Figure 6 compares the spatial patterns of the d time-averaged model-data difference (characterized at 293 grid cells in our database, see Fig. 6a), the situation is contrasted with 50% of positive and negative differences. We can identify systematic trends, with an underestimation of the mean d levels in ECHAM5-wiso for the central East Antarctic Plateau and the Peninsula, and an overestimation above Victoria Land (Fig. 6a). Due to the temperature dependency of equilibrium fractionation coefficients leading to a gradual deviation from the meteoric water line (calculated at the global scale, where the coefficient of 8 results from the average equilibrium fractionation coefficients), d increases when temperature decreases (Masson-Delmotte et al., 2008; Touzeau et al., 2016).For central Antarctica, the d bias is thus consistent with the warm bias and the lack of isotopic depletion. The upper and lower quartiles of the model-data differences range within ±1.5 ± 0.1 ‰, suggesting that the model outputs remain close to those observed. The d pattern is similar to that of $\delta^{18}$O: ECHAM5-wiso underestimates the d standard deviation for 90 % of grid cells, with an interquartile range comparable to the one for the ratio of standard deviations for $\delta^{18}$O (Fig. 6b). Table 7 displays the comparison of the statistics between d in the observations and in ECHAM5-wiso. In the observations, the time-averaged d is particularly low in the Peninsula (-3.6 to 8.6 ‰), intermediate in coastal regions of Dronning Maud Land, Victora Land and Adélie Land (4.4 to 8.6 ‰), and very high in the central Antarctic Plateau (up to 17.5 ‰ for Dome C). Lower coastal values and higher inland values are captured by ECHAM5-wiso, albeit with large offsets for each site, reaching several per mille. These findings are consistent with the map showing the time-averaged precipitation d simulated by ECHAM5-wiso over the period 1979-2013 (Fig. 6a), with very low coastal values (close to zero) and increasing values towards the interior of Antarctica, reaching values higher than 16 ‰ on the plateau. ECHAM5-wiso mainly under-estimates the d intra-annual standard deviation for 10 sites out of 15 (Table 7 and Fig. 6b).

Figure 8d depicts the mean d seasonal patterns of the precipitation data and corresponding model outputs. The data show different patterns from one location to another. While d measured at Neumayer, Halley and Rothera displays a maximum in autumn (March-April), it appears in late autumn (May) at Marsh and in winter (June-August) at Vernadsky. Maxima for central stations are observed later, in May-July for

Dome C and July-September for Dome F. In short, most coastal areas are associated with a maximum d in autumn while central areas are associated with a later maximum d, i.e. in winter or late winter, thus in anti-phase with $\delta^{18}O$ and temperature. The seasonal amplitude increases from the coast to the plateau. In the model, for central areas, a first d maximum is simulated earlier than observed (February-March for Dome F, and May-June for Dome C), followed by a second maximum in late winter (August for Dome F

and September for Dome C). For coastal areas, the amplitude of the simulated d signal is too small to unequivocally estimate the timing of the maximum. Note the very low value simulated at DDU in July, which appears to be an outlier when comparing this value with the average modelled d value for all days in August 1973 (+5.9 ‰). No link emerges between the modelled seasonal patterns in d and in temperature (Fig. 8a), accumulation (Fig. 8b) nor $\delta^{18}O$ (Fig. 8c).

Finally, Table 8 reports the d mean seasonal amplitude values for the precipitation data and ice core records, as well as for the model outputs covering the observation. They clearly show an increase in d seasonal amplitude from the coast to the plateau (see also Fig. 7b), with values varying from 6.7 ‰ at Halley to 41 ‰ at Dome C. ECHAM5-wiso systematically underestimates the d mean seasonal amplitude when compared with precipitation data, while it systematically overestimates it when compared with ice

core data (from 9.4 to 15.5 ‰), with the exception of the GIP ice core. Again, we cannot rule out a loss of amplitude in ice core data compared to the initial precipitation signal, due to the temporal resolution and to post-deposition effects.

### 3.3 Strength and limitations of the ECHAM5-wiso model outputs

The isotopic model-data time-averaged biases appear coherent with temperature. A warm bias over

Central East Antarctica and a cold bias over coastal regions lead to a too low and too strong isotopic depletion, respectively. Temperature and distillation biases also explain the underestimation of d above the central East Antarctic plateau.

However, some characteristics are not explained by model skills for temperature. At sub-annual time scales, ECHAM5-wiso always overestimates the standard deviation of $\delta^{18}O$ in precipitation (Table 3), but results for d are mixed (Table 7). ECHAM5-wiso always underestimates seasonal amplitude of $\delta^{18}O$ and d in precipitation but always overestimates seasonal amplitude of $\delta^{18}O$ and d in firn/ice cores (Table 4

and 8). Differences between the model and firn/core data are at least partially due to diffusion processes, but no clear reason can be given for the other isotopic biases.

We do not find any clear link between other model biases for d and those for temperature or $\delta^{18}O$.

Sampling Antarctic snowfall remains challenging (Fujita and Abe, 2006; Landais et al., 2012; Schlosser et al., 2016; Stenni et al., 2016). Sampling is likely to fail to capture small events, and may also collect

surface snow transported by winds or hoar. Snow samples may undergo sublimation before collection. The fact that ECHAM5-wiso appears to overestimates the variability of precipitation isotopic composition may be related to an improper characterisation of the full day-to-day variability of real world precipitation from daily precipitation sampling. Alternatively, this feature may also arise from a lack of representation of small scale processes (boundary layer processes, wind characteristics, snow-atmosphere interplays) in

ECHAM5-wiso. These processes may contribute to a local source of Antarctic moisture (through local recycling), reducing the influence of large-scale moisture transport (resolved by ECHAM5-wiso nudged to reanalyses) on the isotopic composition of precipitation and its day-to-day variability.

Caveats also limit the interpretation of the comparison of ECHAM5-wiso precipitation outputs with surface snow or shallow ice core data. Such records are potentially affected by post-deposition processes,

such as wind scoring, erosion, snow metamorphism in-between precipitation events and diffusion.

Our apparently contradictory findings for model-data comparisons with respect to inter-annual variations (from ice cores) and inter-daily variations (from precipitation data) call for more systematic comparisons between $\delta^{18}O$ records of precipitation and ice cores at the same locations, over several years.

## 4. Use of ECHAM5-wiso outputs for the interpretation of ice core records

In this section, we use the model outputs to help in the interpretation of ice core data: we quantify the inter-annual isotope-temperature relationships (Section 4.2), and characterize the spatial distribution of seasonal $\delta^{18}O$-d phase lag. Based on the confidence we can have in the model for each of the seven

aforementioned regions (See Section 1 and Fig.1), we formulate recommendations for the future use of ECHAM5-wiso outputs (Section 4.3).

## 4.1 Spatial and temporal isotope-temperature relationships

First, we use ECHAM5-wiso to investigate spatial $\delta^{18}$O-temperature relationships (Fig. 9a and 9b), and then inter-annual (Fig. 9c and Fig. 9d) and seasonal relationships (Fig. 9e and Fig. 9f). For spatial relationships, the strength of the linear correlation coefficient is higher than 0.8. The spatial slope shows regional differences. It is generally smaller near the coasts (less than 0.8 ‰ °C$^{-1}$), with the exception of Dronning Maud Land , and increases at elevations higher than 2500 m a.s.l., with values above 1.2 ‰ °C$^{-1}$ in large areas. Furthermore, ECHAM5-wiso simulates spatial heterogeneity of the gradient in the central East Antarctic plateau, around Dome C, Dome A and Dome F. Such variability may arise from the simulated intermittency of precipitation, and from differences in condensation versus surface temperature. At the inter-annual scale (Fig. 9c and 9d), results are not significant for large areas encompassing the Dronning Maud Land region, the Antarctic Peninsula, the Transantarctic Mountain region, the Ronne and Filchner ice shelve regions, part of Victoria Land and along the Wilkes Land coast. For the whole continent, the correlation coefficient varies between 0.5 and 0.6 (with few values reaching 0.6 at the upper limit and 0.3 at the lower limit). Where correlations are significant, the inter-annual $\delta^{18}$O-temperature slope increases from the coasts (0.3‰ °C$^{-1}$ to 0.6‰ °C$^{-1}$) to the inland regions, where it can exceed 1‰ °C$^{-1}$ for some high elevation locations. The low correlation may be due to the small range of mean annual temperature over the period 1979-2013 and is not necessary indicative of a weak sensitivity to temperature change.

Finally, at the seasonal scale, results are significant almost over the whole continent (with the exception of two little areas in Peninsula and East Antarctica) and the correlation coefficients are equal to one everywhere but along the coastal regions in the Indian Ocean sector, where the correlation coefficient can decrease down to 0.75. Slopes are lower than for spatial and inter-annual relationships, with values from 0.0 to 0.3 ‰ °C$^{-1}$ along the coast (higher over Dronning Maud Land and the Ross Ice Shelf region), around 0.5 ‰ °C$^{-1}$ inland for altitudes lower than 2500 m a.s.l. (with the exception of lower values above the Transantarctic Mountains), and up to 0.8 ‰ °C$^{-1}$ over the East Antarctic Plateau.

To conclude the coherent framework provided by the ECHAM5-wiso simulation covering the period 1979-2013 shows that annual $\delta^{18}O$ and surface temperature are only weakly linearly related in several areas. This suggests that the inter-annual variability of $\delta^{18}O$ is controlled by other processes, for instance associated with synoptic variability and changes in moisture source characteristics (Sturm et al., 2010; Steiger et al., 2017). Moreover, our results rule out the application of a single isotope-temperature slope for all Antarctic ice core records on the inter-annual time scale, and that the seasonal isotope-temperature slope is not a surrogate for scaling inter-annual $\delta^{18}O$ to temperature.

We have also used the simulation to explore linear relationships between d and surface air temperature, without any significant results (not shown).

## 4.2 $\delta^{18}O$ –d phase lag

D has originally been interpreted as a proxy for relative humidity at the moisture source (Jouzel et al., 2013; Pfahl and Sodemann, 2014; Kurita et al., 2016). However, recent studies of Antarctic precipitation data combined with back-trajectory analyses did not support this interpretation (e.g. Dittmann et al., 2016; Schlosser et al., 2017), calling for further work to understand the drivers of seasonal d variations. The phase lag between d and $\delta^{18}O$ was initially explored to identify changes in evaporation conditions (Ciais et al., 1995). In ECHAM5-wiso, this phase lag is calculated as the lag that gives the highest correlation coefficient between d and $\delta^{18}O$ (Fig. 10), using the mean seasonal cycle from monthly averaged values. If there were no seasonal change in moisture origin and climatic conditions during the initial evaporation process, one would expect d to be in anti-phase with $\delta^{18}O$, due to the impact of condensation temperature on equilibrium fractionation. For regions with small seasonal amplitude in condensation temperature, a constant initial isotopic composition at the moisture source would imply a stable d year round. In such regions, the simulated phase lag likely therefore reflects seasonal changes in the d of the initial moisture source. The comparison with precipitation data (Section 3.2.4) showed that ECHAM5-wiso had low seasonal amplitude in coastal regions (Fig. 8d), making the discussion of seasonal maxima difficult. These comparisons are also limited by the duration of the precipitation records. Here, we use the full simulation (1979-2013) to investigate the phase lag between the mean seasonal cycle of d and $\delta^{18}O$. Clear spatial patterns are identified for the distribution of this phase lag (see Fig. 10). At intermediate elevations

(between 1000 and 3000 m a.s.l.), d seasonal variations occur in phase (within 2 months) with the seasonal cycle of $\delta^{18}O$ (and surface air temperature). By contrast, a phase lag of several months is identified over coastal areas and over the central East Antarctic Plateau. Along the Wilkes Land coast and the Dronning Maud Land region, the time lag is between two and four months, below 1000 m a.s.l. and 500 m a.s.l. respectively. Over the West Antarctic Ice Sheet, the phase lag is higher than two months below 500 m a.s.l. and can even reach six months (indicating an anti-phase between d and $\delta^{18}O$). Over the Central East Antarctic Plateau (above 3000 m elevation), the phase lag reaches several months again, especially near Dome C. Obtaining longer precipitation records and comparing the phase lag identified in precipitation and surface snow records would be helpful to understand whether post-deposition processes, which are not included in ECHAM5-wiso, affect this phase lag. The different characteristics of seasonal d changes suggest different seasonal changes in moisture origin at coastal, intermediate and central plateau regions, supporting the identification of specific coastal versus inland regions to assess the isotope-temperature relationships. Note that the few available data sets are in line with the simulation.

**4.3 Recommendations for the different regions of Antarctica**

In this section, we summarize our findings, based on the model-data comparisons and the analysis of model outputs for the 7 Antarctic regions selected by the Antarctica2k program, as shown in Fig.1 (Stenni et al., 2017a). The regions depend on geographical and climatic characteristics. Results from Section 3 were averaged over each region and are given in Table 9.

We first discuss the systematic model biases. The maximum time-averaged model-data differences (3.8 ‰ and 2.6 ‰ for $\delta^{18}O$ and d respectively) are identified in the Weddell Sea area. Minimum time averaged model-data differences occur in different regions for $\delta^{18}O$ and d (Victoria Land and Dronning Maud Land respectively).

For inter-annual standard deviation, the model-data mismatch is smallest for Victoria Land (ratio of 1.1 and 1.0 for $\delta^{18}O$ and d respectively). Results for $\delta^{18}O$ show that the simulated inter-annual variability can be considered close to reality (model-data ratio higher than 0.7) only for Victoria Land and the plateau, acceptable (model-data ratio higher than 0.5) for the Weddell Sea area and the West Antarctic Ice Sheet, but significantly different from observations in the other three regions. The model-data mismatch is larger

for d inter-annual variability, with acceptable inter-annual variability only for Victoria Land and the plateau. However, these results are clearly limited by the low number of observational records for some regions.

Table 10 provides a brief overview of ECHAM5-wiso outputs for our 7 regions of interests, in terms of mean climate and isotopic variables, their standard deviation, seasonal amplitude, and the calculated regional $\delta^{18}O$ – T relationship. The main findings are again the highest slope simulated for the central Antarctic plateau, followed by the Dronning Maud Land and West Antarctic Ice Sheet regions, and weak correlations in some regions (Weddell Sea, Antarctic Peninsula), where water stable isotope outputs are not good predictors of inter-annual temperature change within ECHAM5-wiso, together with low correlations and slopes for the other coastal regions (Indian Ocean sector, Victoria Land).

## 5. Conclusions and perspectives

This study presents a systematic evaluation of a present-day Antarctic climate simulation using the ECHAM5-wiso atmospheric circulation model equipped with water stable isotopes. For this simulation, covering the period 1960-2013, the model has been nudged to ERA atmospheric reanalyses. In particular, we tested its ability to correctly capture time-averaged values, inter-annual variations, and seasonal cycles in surface mass balance, temperature, and precipitation isotopic composition in Antarctica. As possible, we discarded model results prior to 1979, as model-data differences prior to 1979 may arise from uncertainties in the reanalyses, prior to the period where satellite data were assimilated.

Despite some divergences, simulated P-E are found to be a good surrogate for SMB. Most artefacts in modelled $\delta^{18}O$ are coherent with those for temperature, with systematic biases in different regions. Some of these artefacts may be linked to the nudging method and the reanalyses. Model-data comparisons are limited by data availability and by the fact that deposition and post-deposition processes are not considered in the simulation. This is particularly true for precipitation amounts, where there is a lack of direct measurements, and isotopic analysis for many regions at a multi-annual time scale. A systematic comparison between water isotope measurements from precipitation and surface snow or ice core samples is needed for further in-depth studies of this topic. We note a lower quantitative performance from ECHAM5-wiso for d (time-averaged values and inter-annual standard deviations) than for $\delta^{18}O$, beyond

its remarkable ability to resolve the spatial distribution of time-averaged d values. Our findings confirm several other studies conducted in other regions highlighting the fact that atmospheric models including ECHAM5-wiso tend to under-estimate the variability of d in surface vapour (e.g. Steen-Larsen et al, 2016). Expanding earlier site-specific studies, we show that the strength and slope of the $\delta^{18}O$-temperature

linear relationship is dependent on the time scale in Antarctica over the four last decades. This findings has implications for past temperature reconstructions using ice core records. Finally, interesting results emerge for regional differences in the phase lag between the mean seasonal cycle in $\delta^{18}O$ and d, calling for further studies to better characterise this feature in precipitation and ice core records, and better understand its implications of these lags for the representation of seasonal changes in moisture source

effects.

Our study would deserve to be expanded to other atmospheric models equipped with water stable isotopes, and other nudged simulations using different reanalyses datasets, to assess the robustness of our findings. Furthermore, obtaining more high-resolution ice core records is crucial to be able to better assess model skills for inter-annual variations. More measurements of precipitation, surface snow and vapour

monitoring for water isotopes would also help to better characterize deposition and post-deposition processes, their implication for model-data evaluation studies, and for an improved climatic interpretation of ice core records.

**Acknowledgements**

This study has been supported by the ASUMA project supported by the ANR (Agence Nationale de la

Recherche, Project n°: ANR-14-CE01-0001), which funded the PhD grant of Sentia Goursaud and the publication costs of this manuscript.

## Tables

**Table 1: Differences between observed (READER) and simulated (ECHAM5-wiso) annual surface air temperature: observed average (noted as "observed μ", in °C), average difference (noted as "μ differences", in °C), standard deviation from observations (noted as "observed σ", in °C) and standard deviation from the model (noted as "simulated σ", in °C) for the period 1979-2013.**

|  | Neumayer | Mawson | Vostok | Casey | Dome C | DDU | McMurdo | Byrd | Palmer | Esperanza |
|---|---|---|---|---|---|---|---|---|---|---|
| Observed μ (°C) | -16.0 | -11.2 | -55.4 | -9.2 | -51.1 | -10.7 | -13.4 | -26.9 | -1.5 | -5.1 |
| μ differences (°C) | -0.8 | -1.6 | 3.2 | -7.3 | 1.7 | -7.2 | -14.9 | 1.4 | -2.8 | -0.3 |
| Observed σ (°C) | 0.67 | 0.74 | 0.99 | 0.92 | 0.97 | 0.66 | 0.74 | 1.2 | 0.33 | 1.1 |
| Simulated σ (°C) | 0.79 | 0.71 | 1.00 | 1.10 | 1.20 | 0.80 | 1.70 | 1.4 | 0.96 | 0.84 |

**Table 2: Linear relationship between surface temperatures (in °C) from station instrumental records and ECHAM5-wiso outputs (in °C) over the period for 1960-2013 and 1979-2013: the slope (in °C °C⁻¹), the correlation coefficient (noted as "r") and the p-value. Data are not reported for 1960-2013 for stations for which records only cover the second period (1979-2013). Numbers in brackets correspond to standard errors.**

| | Period 1960-2013 slope (in °C °C⁻¹) | r | p-value | Period 1979-2013 slope (in °C °C⁻¹) | r | p-value |
|---|---|---|---|---|---|---|
| Neumayer | | | | 0.8 (<0.1) | 0.8 | <0.001 |
| Mawson | 0.5 (0.2) | 0.4 | 0.002 | 0.8 (<0.1) | 0.9 | <0.001 |
| Casey | 1.1 (0.2) | 0.6 | <0.001 | 0.9 (0.1) | 0.9 | <0.001 |
| Dome C | | | | 1.0 (<0.1) | 0.9 | <0.001 |
| DDU | 1.0 (0.4) | 0.4 | 0.004 | 1.1 (0.1) | 0.9 | <0.001 |
| McMurdo | 0.8 (<0.1) | 0.8 | <0.001 | 0.3 (0.2) | 0.2 | 0.2 |
| Byrd | 1.1 (0.2) | 0.7 | <0.001 | 0.8 (0.1) | 0.8 | <0.001 |
| Palmer | | | | 0.7 (0.3) | 0.7 | 0.05 |
| Esperanza | 0.7 (<0.1) | 0.7 | <0.001 | 0.7 (<0.1) | 0.9 | <0.001 |

**Table 3: Comparison between measurements from precipitation samples and ECHAM5-wiso simulated precipitation isotopic composition for grid cells closest to sampling locations over the same period than the data (at daily or monthly scale, when the name of the station is associated with an asterisk). We report the mean value ± the standard deviation for δ¹⁸O (in ‰) and for temperature (°C).**

| | Number of points | Data | | Model | 5 |
| --- | --- | --- | --- | --- | --- |
| | | Temperature (°C) | $\delta^{18}O$ (‰) | Temperature (°C) | $\delta^{18}O$ (‰) |
| Rothera* | 194 | -12.9 ± 3.4 | -4.0 ± 4.1 | -12.3 ± 6.4 | -6.7 ± 5.3 |
| Vernadsky* | 372 | -9.9 ± 3.1 | -3.1 ± 3.6 | -13.5 ± 6.0 | -9.0 ± 7.2 |
| Halley* | 552 | -22.0 ± 5.5 | -18.7 ± 1.7 | -25.7 ± 7.1 | -20.1 ± 7.6 |
| Marsh* | 19 | -12.1 ± 4.1 | -3.4 ± 3.0 | -10.4 ± 5.1 | -4.2 ± 3.6 |
| Dome F | 351 | -61.3 ± 10.8 | -54.7 ± 12.6 | -58.3 ± 12.2 | -53.4 ± 12.1 |
| Dome C | 501 | -58.0 ± 8.6 | -55.2 ± 13.8 | -59.6 ± 17.4 | -52.9 ± 10.9 |
| DDU | 19 | -18.0 ± 3.8 | | -23.4 ± 5.1 | -21.3 ± 7.8 |
| Neumayer | 336 | -20.8 ± 6.6 | -13.4 ± 8.0 | -21.3 ± 7.9 | -15.8 ± 7.9 |

**Table 4: $\delta^{18}O$ mean seasonal amplitude (in ‰) calculated for precipitation and sub-annual ice core data, as well as simulated by ECHAM5-wiso for the same time period than the data. The time resolution used in the model corresponds to the time resolution of the precipitation data, and to the annual scale for the ice core data (i.e. yearly averages based on daily precipitation isotopic composition weighted by the amount of daily precipitation). The data type is identified as 1 for precipitation samples and 2 for ice core data.**

| Station | Type | $\delta^{18}O$ observed amplitude (‰) | ECHAM5-wiso averaged over the observed period (‰) |
|---|---|---|---|
| Rothera | 1 | 4.1 | 1.9 |
| Vernadsky | 1 | 4.1 | 2.3 |
| Halley | 1 | 13.2 | 6.7 |
| Marsh | 1 | 10.4 | 7.3 |
| Dome F | 1 | 25.9 | 15.3 |
| Dome C | 1 | 20.1 | 13.5 |
| DDU | 1 | 6.1 | 3.7 |
| Neumayer | 1 | 12.8 | 7.9 |
| USITASE-1999-1 | 2 | 7.2 | 13.2 |
| USITASE-2000-1 | 2 | 4.8 | 10.6 |
| USITASE-2000-2 | 2 | 7.7 | 10.4 |
| USITASE-2000-4 | 2 | 4.0 | 12.0 |
| USITASE 2000-5 | 2 | 5.2 | 13.7 |
| USITASE-2000-6 | 2 | 2.8 | 14.2 |
| USITASE-2001-1 | 2 | 7.3 | 9.4 |
| USITASE-2001-2 | 2 | 7.3 | 12.0 |
| USITASE-2001-4 | 2 | 6.2 | 8.9 |
| USITASE-2001-5 | 2 | 6.8 | 9.0 |
| USITASE-2002-1 | 2 | 4.2 | 12.2 |
| USITASE-2002-2 | 2 | 6.3 | 10.7 |
| USITASE-2002-4 | 2 | 5.3 | 13.8 |
| NUS 08-7 | 2 | 3.4 | 16.6 |
| NUS 07-1 | 2 | 2.1 | 14.8 |
| WDC06A | 2 | 4.0 | 10.8 |
| IND25 | 2 | 5.3 | 12.6 |
| GIP | 2 | 15.1 | 16.8 |

**Table 5: Slope (in ‰ °C$^{-1}$), correlation coefficient and p-value of the $\delta^{18}O$ -temperature linear relationship from precipitation measurements over the available period and at daily or monthly (when the name of the station is associated with an asterisk) scale depending of the time resolution of the data, and from the ECHAM5-wiso model over the observed period at the time resolution of the data. Numbers in brackets correspond to the standard errors.**

| | Number of points | Data | | | ECHAM5-wiso over the observed period | | |
| --- | --- | --- | --- | --- | --- | --- | --- |
| | | slope (‰ °C$^{-1}$) | r | p-value | slope (‰ °C$^{-1}$) | r | p-value |
| Rothera* | 194 | 0.31 (0.06) | 0.38 | <0.001 | 0.01 (0.03) | 0.23 | <0.001 |
| Vernadsky* | 372 | 0.32 (0.04) | 0.39 | <0.001 | 0.09 (0.02) | 0.25 | <0.001 |
| Halley* | 552 | 0.47 (0.02) | 0.76 | <0.001 | 0.48 (0.02) | 0.68 | <0.001 |
| Marsh* | 19 | 0.61 (0.31) | 0.44 | 0.07 | 0.47 (0.23) | 0.43 | 0.06 |
| Dome F | 351 | 0.76 (0.02) | 0.88 | <0.001 | 0.70 | 0.62 | <0.001 |
| Dome C | 501 | 0.59 (0.02) | 0.64 | <0.001 | 0.94 (0.07) | 0.55 | <0.001 |
| Neumayer | 336 | 0.57 (0.03) | 0.69 | <0.001 | 0.29 (0.06) | 0.29 | <0.001 |

**Table 6: Slope (in ‰ ‰$^{-1}$), correlation coefficient and p-value of the δ$^{18}$O-δD linear relationship from precipitation measurements (top of the table) and ice core data (bottom of the table) over the available period and at daily or monthly scale (identified with an asterisk), and from the ECHAM5-wiso model over the observed period at the time resolution of the data for the precipitation and at the annual scale for the ice core data. Numbers into brackets correspond to the standard errors.**

| | Number of points | Observations slope (‰ ‰$^{-1}$) | r | p-value | ECHAM5-wiso slope (‰ ‰$^{-1}$) | r | p-value |
|---|---|---|---|---|---|---|---|
| Rothera* | 194 | 7.0 (<0.1) | 9.81E-01 | <0.001 | 7.9 (<0.1) | 9.97E-01 | <0.001 |
| Vernadsky* | 372 | 6.6 (<0.1) | 9.62E-01 | <0.001 | 7.8 (0) | 9.96E-01 | 0 |
| Halley* | 552 | 7.8 (0) | 9.91E-01 | 0 | 7.8 (<0.1) | 9.95E-01 | <0.001 |
| Marsh* | 19 | 7.1 (<0.1) | 9.80E-01 | <0.001 | 8.0 (<0.1) | 9.94E-01 | <0.001 |
| Dome F | 351 | 6.4 (<0.1) | 9.92E-01 | <0.001 | 7.3 (<0.1) | 9.91E-01 | <0.001 |
| Dome C | 501 | 6.5 (0) | 9.89E-01 | 0 | 6.3 (<0.1) | 9.73E-01 | <0.001 |
| DDU | 19 | 8.5 (<0.1) | 9.92E-01 | <0.001 | 9.0 (<0.1) | 9.86E-01 | <0.001 |
| Neumayer | 336 | 7.9 (<0.1) | 9.90E-01 | <0.001 | 7.8 (<0.1) | 9.98E-01 | <0.001 |
| NUS 08-7 | 256 | 8.6 (<0.1) | 9.96E-01 | <0.001 | 8.0 (<0.1) | 9.95E-01 | <0.001 |
| NUS 07-1 | 118 | 8.3 (<0.1) | 9.94E-01 | <0.001 | 7.6 (<0.1) | 9.94E-01 | <0.001 |
| WDC06A | 540 | 8.2 (0) | 9.95E-01 | 0.00 | 8.1 (<0.1) | 9.98E-01 | <0.001 |
| IND25 | 349 | 8.2 (<0.1) | 9.82E-01 | <0.001 | 7.8 (<0.1) | 9.94E-01 | <0.001 |
| GIP | 495 | 7.8 (0) | 9.90E-01 | 0 | 8.3 (<0.1) | 9.98E-01 | <0.001 |

**Table 7: Mean value ± standard deviation (in ‰) of sub-annual d in observational time series at daily or monthly scale (identified with an asterisk) for the precipitation and for the ice core data, and simulated d by ECHAM5-wiso for the same time period as the observations for precipitation and at the annual scale for the ice core. Mean values which are overestimated by ECHAM5-wiso are written in italic.**

|  | Data (‰) | ECHAM5-wiso over the observed period (‰) |
|---|---|---|
| Rothera* | -1.1 ± 5.7 | *2.9 ± 1.5* |
| Vernadsky* | -1.5 ± 7.0 | *3.5 ± 1.5* |
| Halley* | 5.77 ± 6.1 | 2.8 ± 1.9 |
| Marsh* | 8.6 ± 7.0 | 2.4 ±1.7 |
| Dome F | 17.4 ± 19.5 | 15.3 ± 14.2 |
| Dome C | 17.5 ± 15.2 | 14.2 ± 24.3 |
| DDU | 5.9 ± 4.5 | 2.8 ± 9.5 |
| Neumayer | 8.7 ± 5.6 | 2.4 ± 4.7 |
| NUS 08-7 | 5.0 ± 2.7 | *6.6 ± 1.0* |
| NUS 07-1 | 5.8 ± 2.3 | *7.0 ± 1.1* |
| WDC06A | 3.6 ± 1.5 | *4.5 ± 0.5* |
| IND25 | 4.4 ± 19.2 | 4.1 ± 0.9 |
| GIP | 6.0 ± 4.4 | 2.1 ± 0.9 |

**Table 8: d mean seasonal amplitude (in ‰) calculated for precipitation at daily or monthly scale (identified with an asterisk) and sub-annual ice core data, as well as simulated by ECHAM5-wiso for the same time period as each record. The data type is identified as 1 for precipitation samples and 2 for ice core records. Amplitude values that are overestimated by ECHAM5-wiso are written in italic.**

| Station | Type | Observed amplitude (‰) | ECHAM5-wiso outputs for the observed period (‰) |
|---|---|---|---|
| Rothera* | 1 | 10.7 | 3.1 |
| Vernadsky* | 1 | 11.8 | 2.1 |
| Halley* | 1 | 6.7 | 3.8 |
| Marsh* | 1 | 25.8 | 4.5 |
| Neumayer | 1 | 7.3 | 5.3 |
| Dome F | 1 | 40.1 | 12.2 |
| Dome C | 1 | 41.0 | 25.4 |
| NUS 08-7 | 2 | 3.5 | *14.0* |
| NUS 07-1 | 2 | 1.9 | *15.8* |
| WDC06A | 2 | 1.0 | *16.5* |
| IND25 | 2 | 2.3 | *11.7* |
| GIP | 2 | 17.8 | 6.9 |

**Table 9: Evaluation of ECHAM5-wiso model for 7 Antarctic regions: East Antarctic Plateau, Coastal Indian, Weddel Sea, Peninsula, West Antarctic Ice Sheet, Victoria Land and Dronning Maud Land (7). We regionally averaged the time-averaged $\delta^{18}$O mean (model – data) differences (in ‰), the inter-annual $\delta^{18}$O standard deviation (model/data) ratio, the time-averaged d mean (model – data) differences (in ‰), the inter-annual d (model-data) standard deviation ratio using only precipitation data. Green cells correspond to parameters for which we support the validity of the use of ECHAM5-wiso for the considered region, orange cells to parameters we suggest some cautions and red cells to parameters we suggest not to use ECHAM5-wiso outputs for the considered region. Numbers in brackets correspond to the number of data points.**

| Region | Plateau | Coastal Indian | Weddell Sea | Peninsula | West Antarctic Ice Sheet | Victoria Land | Dronning Maud Land |
|---|---|---|---|---|---|---|---|
| $\delta^{18}$O mean difference (in ‰) | 2.5 (551) | 1.8 (68) | 3.8 (38) | 2.6 (30) | 3.7 (48) | 0.6 (246) | 1.1 (70) |
| $\delta^{18}$O standard deviation ratio | 0.9 (62) | 0.3 (3) | 0.6 (12) | 0.1 (1) | 0.5 (7) | 1.1 (2) | 0.3 (13) |
| d mean difference (in ‰) | 0.4 (402) | 1.3 (20) | -2.6 (12) | -1.1 (25) | -0.6 (31) | 2.3 (232) | 0.2 (18) |
| d standard deviation ratio | 0.7 (62) | 0.3 (3) | 0.4 (12) | 0.1 (1) | 0.3 (7) | 1.0 (2) | 0.2 (13) |

**Table 10: Exploration of the ECHAM5-wiso model outputs (1979-2013) for 7 Antarctic regions: East Plateau, Coastal Indian, Weddel Sea, Peninsula, West Antarctic Ice Sheet, Victoria Land and Dronning Maud Land (7). For each of the following variables: precipitation (in mm w.e. y$^{-1}$), temperature (in °C), δ$^{18}$O (in ‰) and d (in ‰), we regionally averaged the annual mean values (lines 1 to 4), the inter-annual standard deviation (lines 5 to 8), and the mean seasonal amplitude (lines 9 to 12). Finally, we calculated the statistics of the inter-annual δ$^{18}$O -temperature linear relationship: the slope (noted as "a", in ‰ °C$^{-1}$), the correlation coefficient (noted as "r") and the p-value for each region.**

| Regions | | Plateau | Coastal Indian | Weddell Sea | Peninsula | West Antarctic Ice Sheet | Victoria Land | Dronning Maud Land |
|---|---|---|---|---|---|---|---|---|
| Time-averaged values | Precipitation (in cm w.e. y$^{-1}$) | 6.7 | 40.7 | 9.0 | 68.8 | 25.9 | 14.1 | 24.3 |
| | Temperature (in °C) | -39.8 | -20.1 | -29.3 | -14.2 | -24.2 | -27.7 | -19.7 |
| | δ$^{18}$O (in ‰) | -42.3 | -24.3 | -30.6 | -18.9 | -26.6 | -28.8 | -25.2 |
| | d (in ‰) | 6.9 | 4.7 | 3.6 | 3.1 | 3.3 | 3.3 | 3.8 |
| Inter-annual standard deviation | Precipitation (in cm w.e. y$^{-1}$) | 0.6 | 4.2 | 1.6 | 9.0 | 2.5 | 2.4 | 3.3 |
| | Temperature (in °C) | 0.5 | 0.7 | 0.8 | 0.9 | 0.7 | 0.7 | 0.5 |
| | δ$^{18}$O (in ‰) | 0.6 | 0.4 | 0.8 | 0.4 | 0.7 | 0.9 | 0.4 |
| | d (in ‰) | 0.4 | 0.4 | 0.4 | 0.2 | 0.4 | 0.5 | 0.5 |
| Mean seasonal amplitude | Precipitation (in cm w.e. y$^{-1}$) | 5.2 | 28.4 | 7.2 | 45.0 | 19.2 | 12.2 | 19.3 |
| | Temperature (in °C) | 23.9 | 16.5 | 24.2 | 17.8 | 21.7 | 24.4 | 18.1 |
| | δ$^{18}$O (in ‰) | 10.9 | 4.4 | 12.2 | 4.1 | 8.8 | 10.7 | 7.3 |

| | | | | | | | | |
|---|---|---|---|---|---|---|---|---|
| | d (in ‰) | 7.3 | 4.3 | 4.7 | 2.5 | 4.1 | 5.1 | 4.2 |
| Inter-annual $\delta^{18}O$-temperature relationship | a | 0.7 | 0.2 | 0.3 | 0.1 | 0.6 | 0.6 | 0.4 |
| | r | 0.6 | 0.4 | 0.3 | 0.3 | 0.6 | 0.5 | 0.4 |
| | p-value | 5.9E-05 | 8.5E-03 | 4.5E-02 | 6.5E-02 | 1.9E-04 | 2.7E-03 | 2.2E-02 |

**Figures**

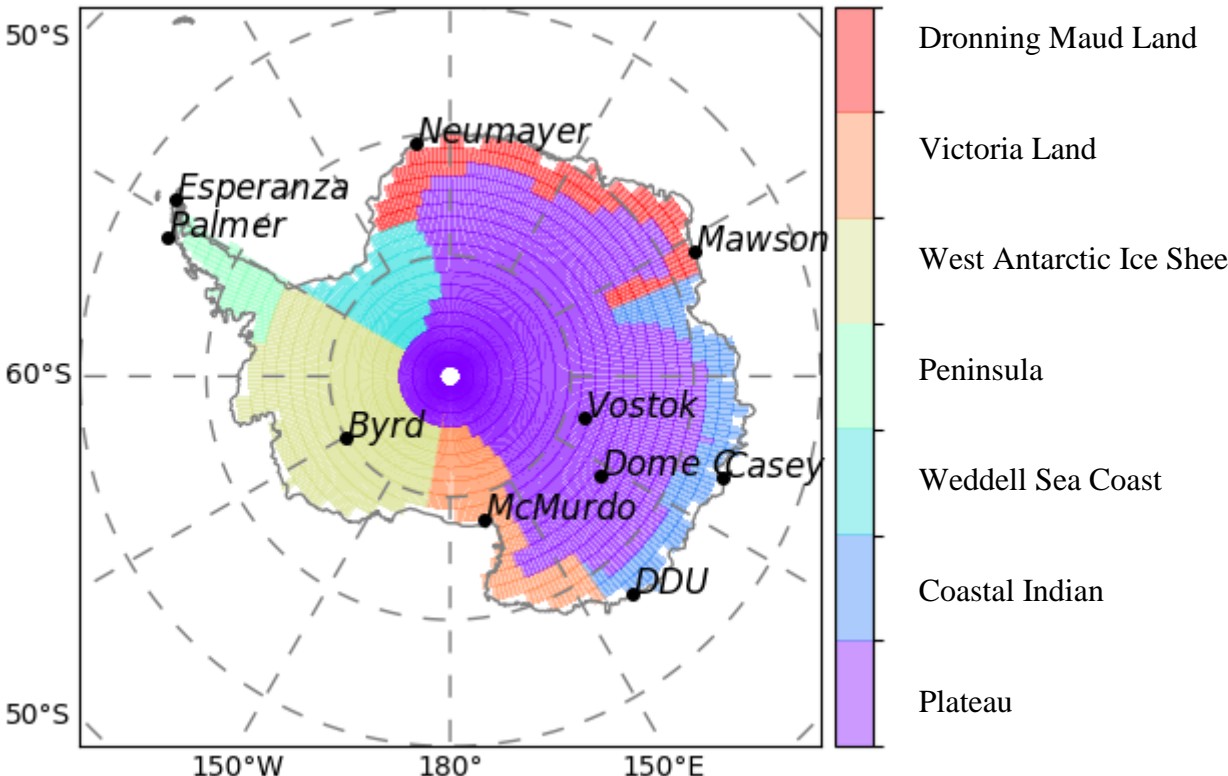

Figure 1: Spatial distribution of Antarctica in 7 regions: East Antarctic plateau, coastal Indian, Weddell sea, West Antarctic Ice Sheet, Victoria Land and Droning Maud Land regions; and the location of the selected READER surface stations: Neumayer, Mawson, Vostok, Dome C, Casey, Dumont d'Urville (noted as "DDU"), McMurdo, Byrd, Palmer and Esperanza stations.

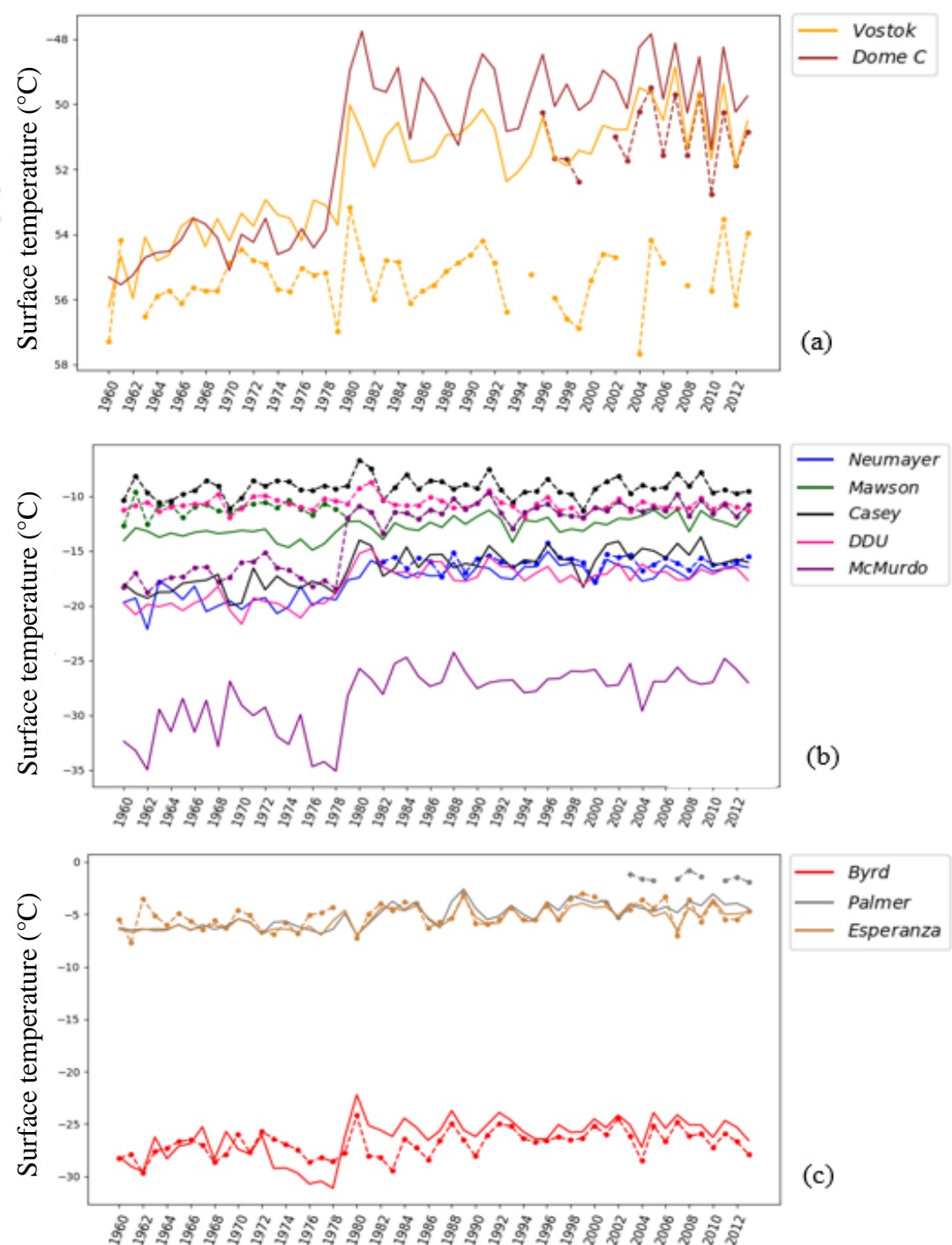

**Figure 2: Surface air temperature (in °C) from station instrumental records (points and dashed lines) and simulated by the ECHAM5-wiso model (solid lines) over the period 1960-2013 for (a) the Plateau, (b) Coastal East Antarctic Ice Sheet and (c) the West Antarctic Ice Sheet. Note that the plots were organized by regions to make it more readable: inland (a), coastal (b) and West Antarctic Ice Sheet plus Peninsula (c).**

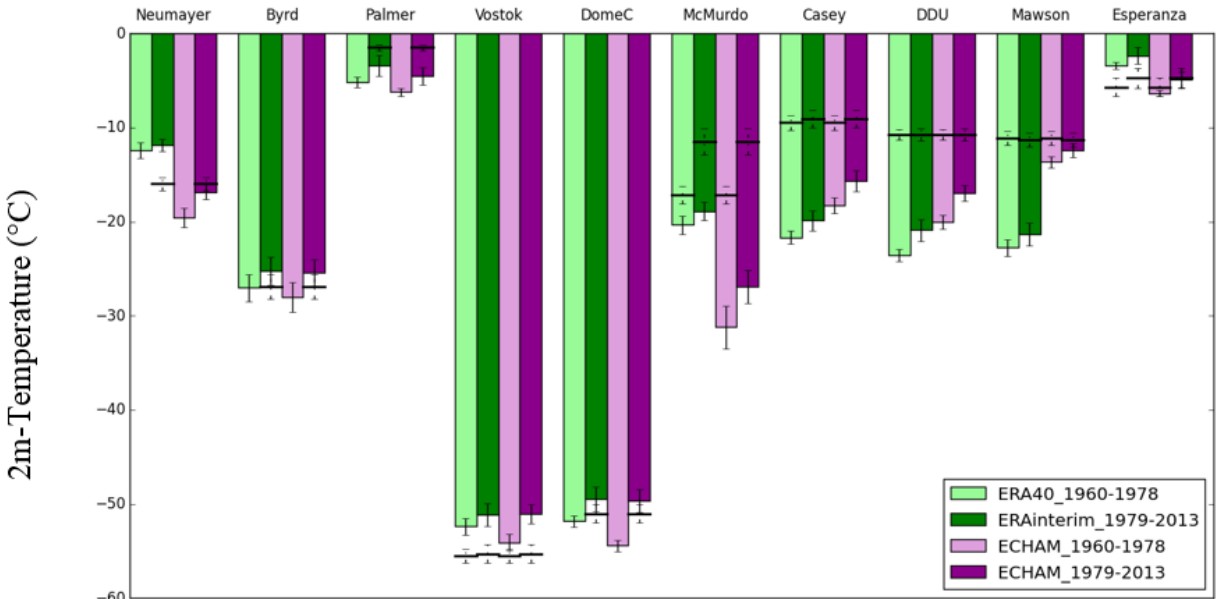

**Figure 3: 2m-temperature outputs (in °C) from ERA-40 (light green), ERA-interim (dark green) and ECHAM5-wiso outputs over the periods 1960-1979 (light purple) and 1979-2013 (dark purple) at the locations of Neumayer, Byrd, Palmer, Vostok, Dome C, McMurdo, Casey, Dumont d'Urville (written as DDU), Mawson and Esperanza stations. Horizontal black lines correspond to the mean data. Vertical black lines correspond to inter-annual standard deviations: dotted lines are associated with data, while solid lines are associated with model outputs (ERA or ECHAM).**

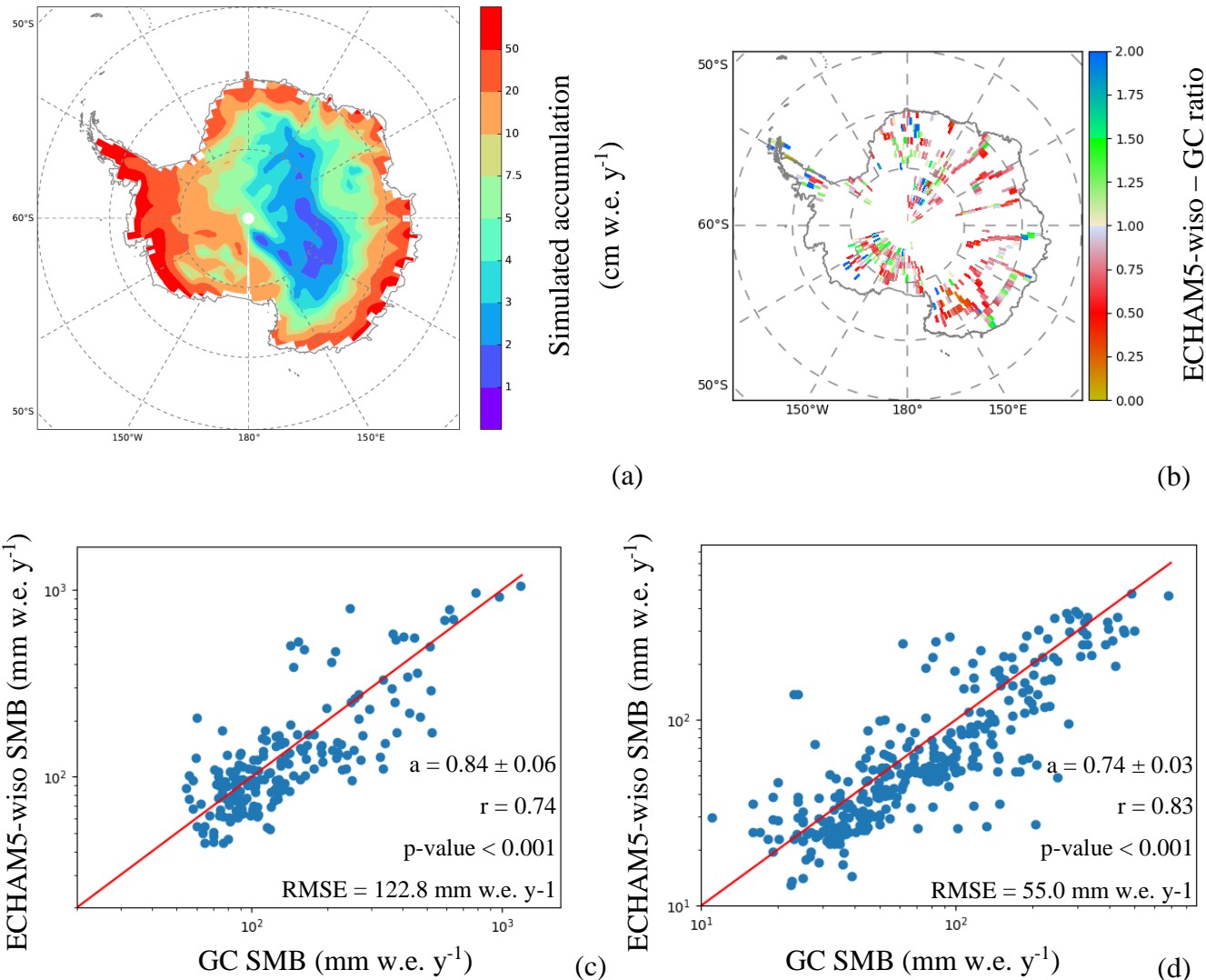

**Figure 4: Comparison of the Glacioclim (hereafter, and noted in the plots "GC") SMB database averaged within the ECHAM5-wiso grid cells and the SMB (i.e. precipitation – evaporation) simulated by the model, with first the spatial distribution of the accumulation (a) as simulated by the model (in cm w.e. y⁻¹), (b) the ratio of the ECHAM5-wiso annual accumulation (precipitation minus evaporation) to the GC averaged SMB (no unit), and finally GC averaged SMB values against SMB values simulated by the model (blue dots) associated with the corresponding linear relationships (red solid line), displayed at the logarithm scale, fore elevation ranges of 0-2200 m a.s.l. (with the upper limit excluded) (c) and 2200-4000 m a.s.l. (d).**

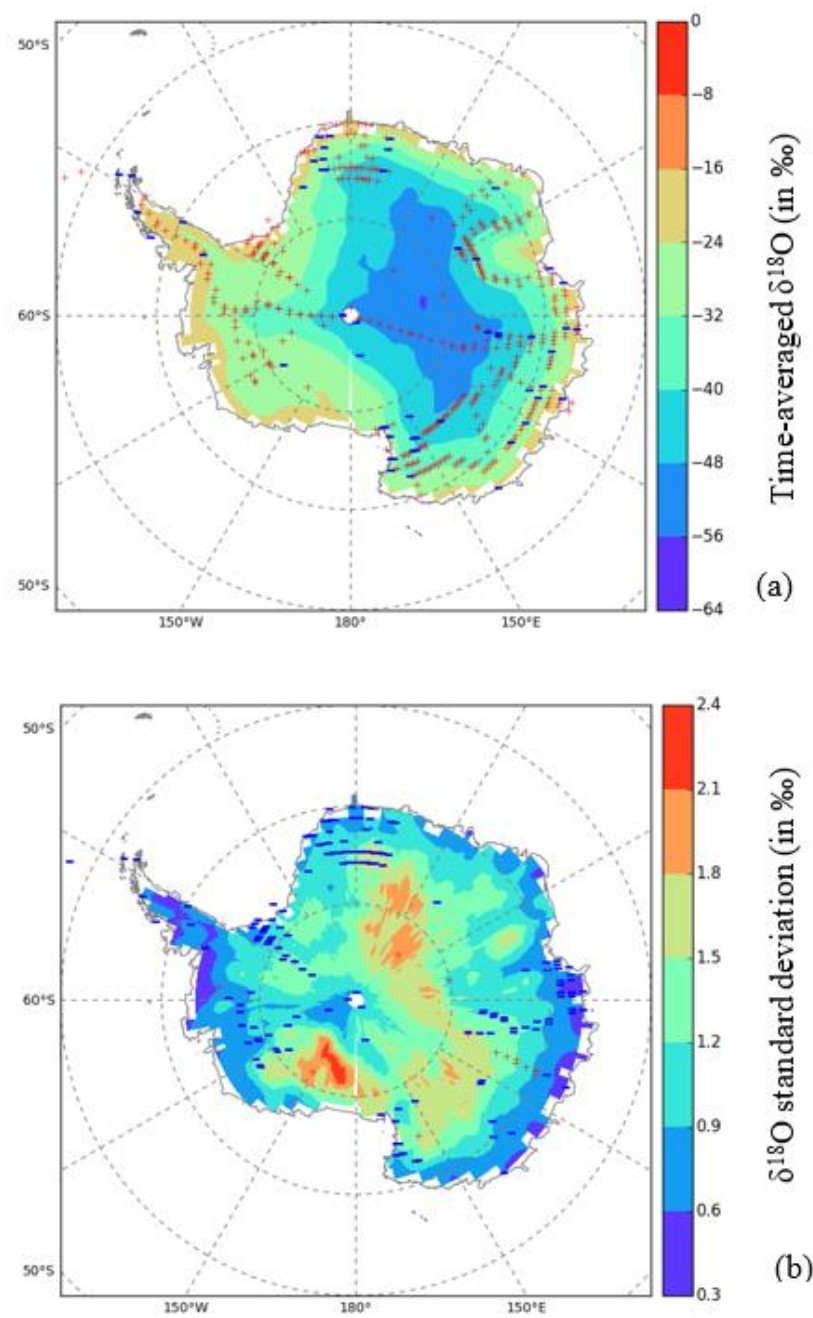

**Figure 5: Maps displaying model-data comparisons for δ18O time-averaged values (a) and inter-annual standard deviations (b). Backgrounds correspond to ECHAM5-wiso simulations over the period 1979-2013, while signs correspond to the model-data comparison. For the tim-averaged values, thecomparison consists in calculating the model-data differences. Red "+" symbols indicate a positive model-data difference while blue "-" symbol correspond to a negative model-data difference. For the inter-annual standard deviations, the comparison consists in calculating the ratio of the simulated value to the corresponding grid cell data. Red "+" symbols indicate a ratio higher than 1 while blue "-" symbol correspond to a model/data ratio lower than 1.**

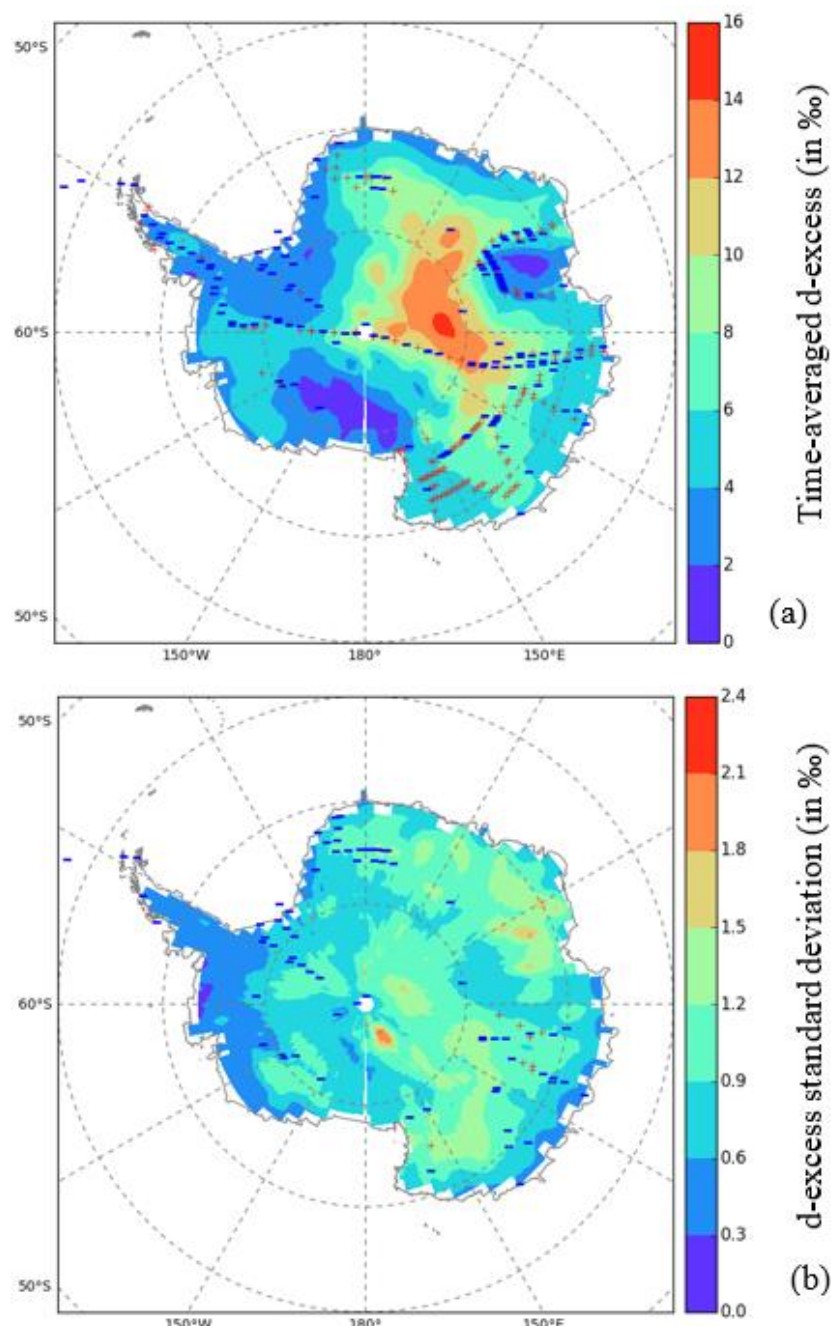

**Figure 6: Maps displaying model-data comparisons for d time-averaged (in ‰, a) values and inter-annual standard deviations (in ‰, b). Backgrounds correspond to ECHAM5-wiso simulations over the period 1979-2013, while signs correspond to the model-data comparison. For the time-averaged values, the comparison consists in calculating the model-data differences. Red "+" symbols indicate a positive model-data difference while blue "-" symbol correspond to a negative model-data difference. For the inter-annual standard deviation, the comparison consists in calculating the ratio of the simulated value to the corresponding grid point data. Red "+" symbols indicate a ratio higher than 1 while blue "-" symbol correspond to a model/data ratio lower than 1.**

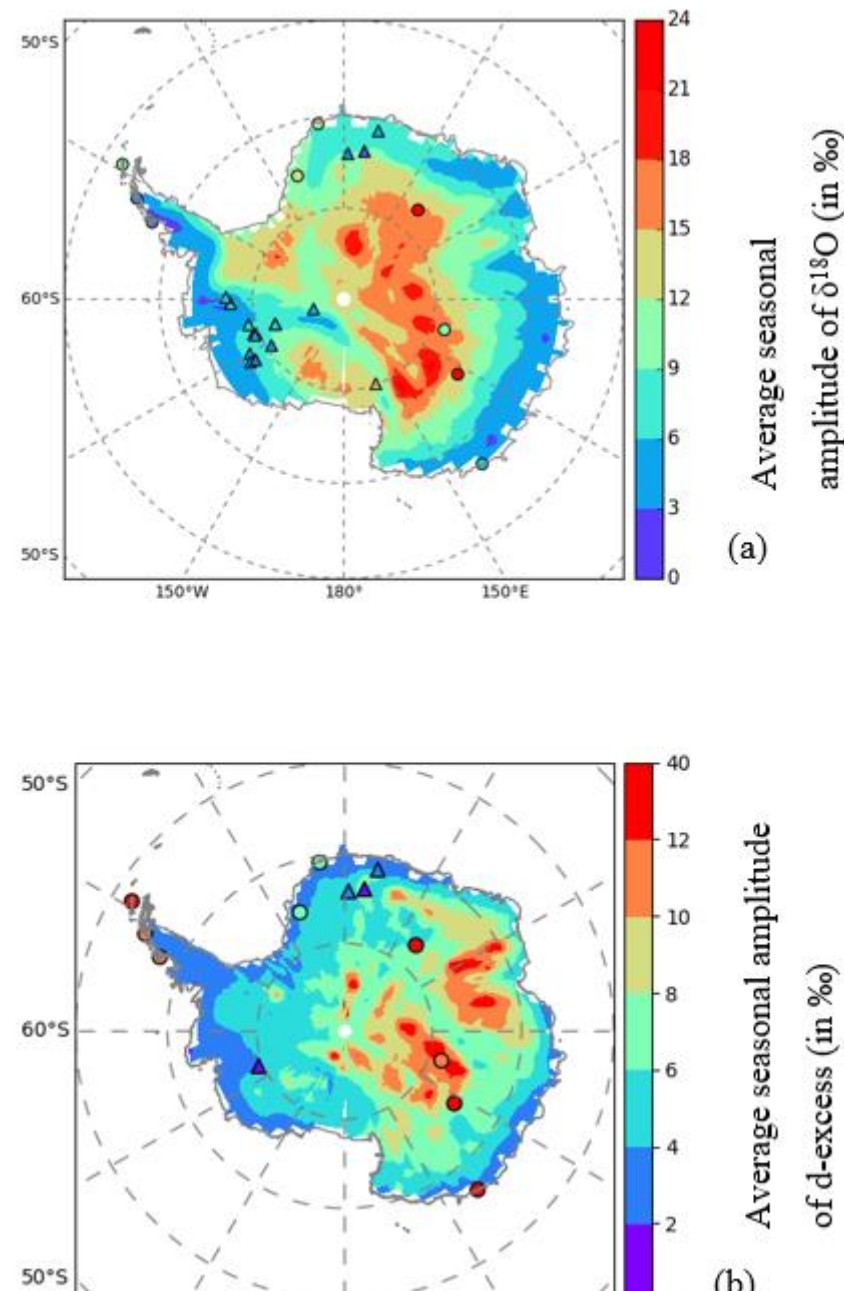

**Figure 7: Average seasonal amplitude of precipitation δ¹⁸O (a) and d (b) (in ‰) simulated by ECHAM5-wiso (color shading) over the period 1979-2013 and calculated from precipitation data (circles) and ice core records (triangles) over their respective available periods.**

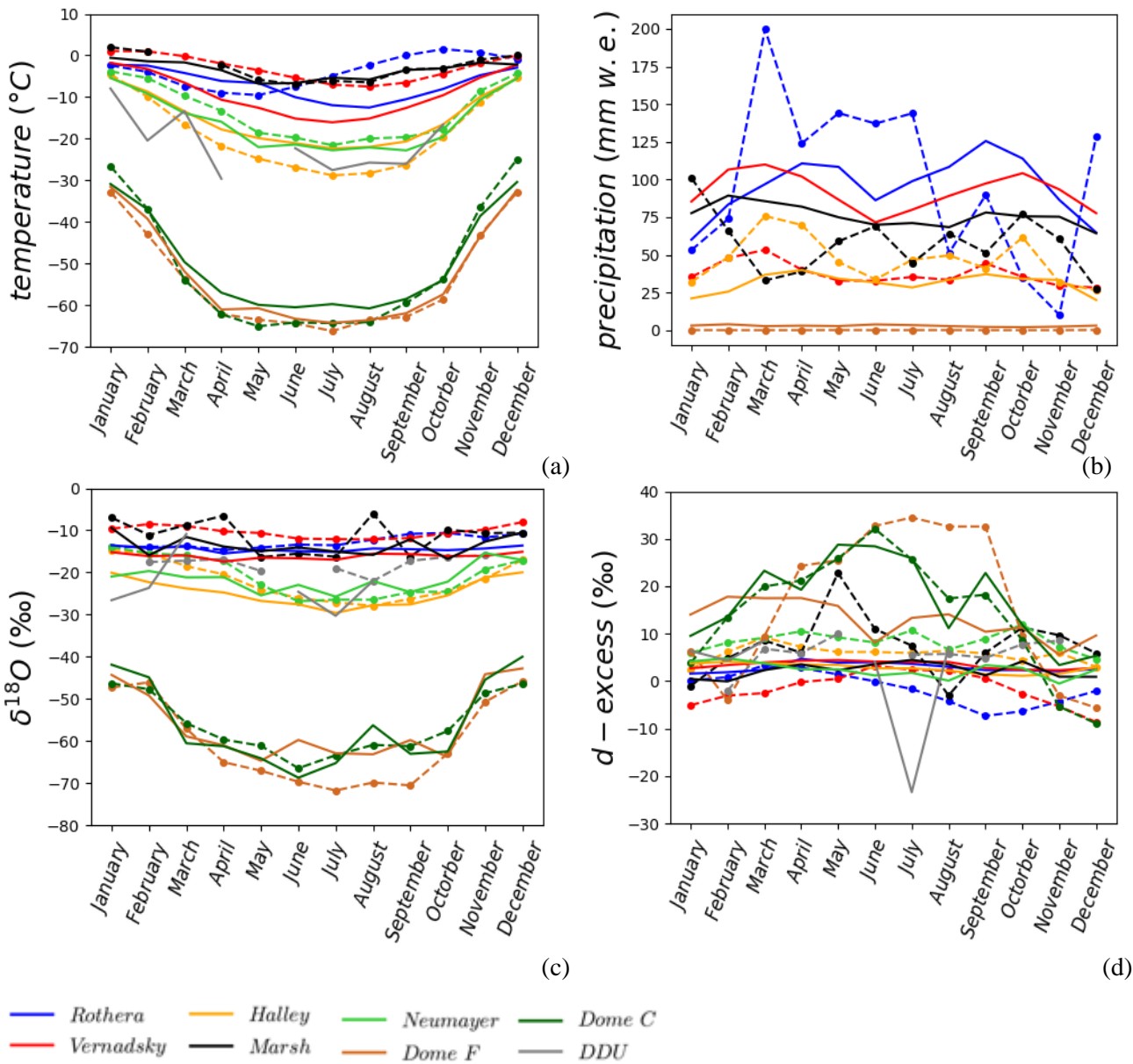

5  **Figure 8: Average seasonal cycles from precipitation data over the available period (dashed lines with points) and simulated by the ECHAM5-wiso model over the period 1979-2013 (solid lines) of the temperature (in °C) (a), the precipitation (in mm w.e. y⁻¹), the precipitation δ¹⁸O (in ‰) (c) and the deuterium excess (in ‰) (d). Data are shown for different durations, depending on sampling, while model results are showed for the period 1979-2013. The number of points used for the observations are given in Table 3.**

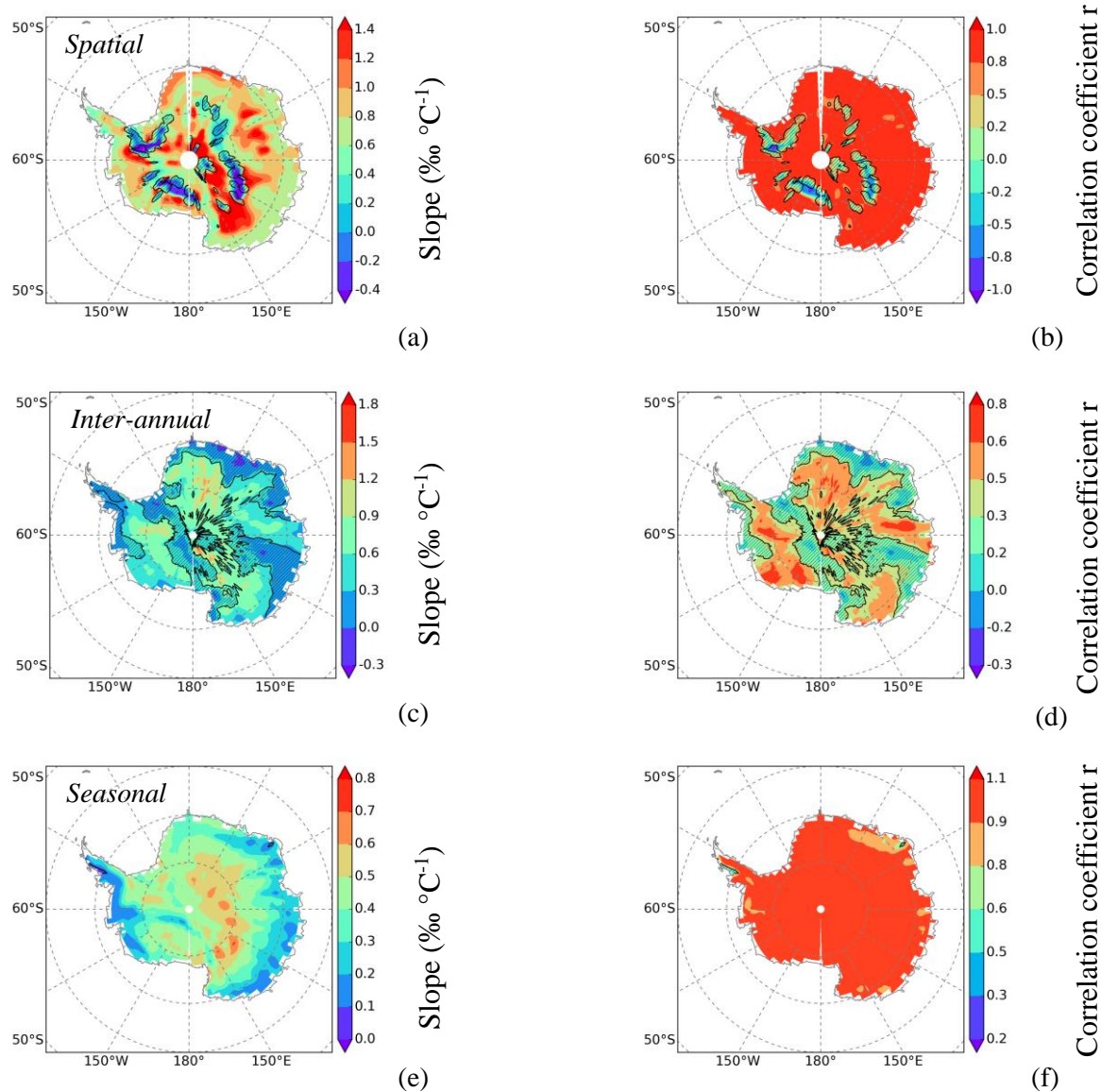

**Figure 9: Linear analysis of annual ECHAM5-wiso outputs from 1979-2013 for the temporal δ18O-temperature relationship (using the 2-meter temperature and the precipitation weighted δ18O). Maps show the slope of the linear regression (‰ °C⁻¹) at the right side (a, c and e) and the correlation coefficient at the left side (b, d, and f). The upper plots uses outputs at the spatial scale (a and b), the middle plots at the inter-annual scale (c and d) and the lower plots at the seasonal scale (e and f). Areas where results of the**
5  **linear analysis are not significant are hatched (p-value>0.05).**

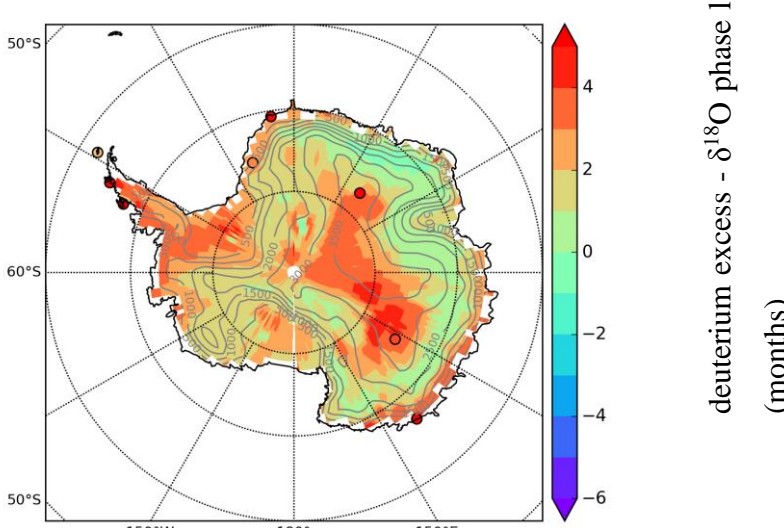

**Figure 10: Best correlated phase lag between the mean seasonal cycle of deuterium excess and that of δ¹⁸O simulated by ECHAM5-wiso over the period 1979-2013 (color shading) and calculated from precipitation data (circles) . The sign informs on the sign of the correlation between δ¹⁸O and d (e.g. positive numbers correspond to a correlation, while negative numbers correspond to an anti-correlation). The absolute value corresponds to the lag (in months) between δ¹⁸O and deuterium excess corresponding to the highest correlation of monthly averaged values. This figure also displays the Antarctic topography, with isohypses (in m a.s.l.).**

Agosta, C., Favier, V., Krinner, G., Gallée, H., Fettweis, X., and Genthon, C.: High-resolution modelling of the Antarctic surface mass balance, application for the twentieth, twenty first and twenty second centuries, Climate dynamics, 41, 3247-3260, 2013.

Arthern, R. J., Winebrenner, D. P., and Vaughan, D. G.: Antarctic snow accumulation mapped using polarization of 4.3-cm wavelength microwave emission, Journal of Geophysical Research: Atmospheres, 111, 2006.

Bromwich, D. H., and Weaver, C. J.: Latitudinal displacement from main moisture source controls δ18O of snow in coastal Antarctica, Nature, 301, 145, 10.1038/301145a0, 1983.

Bromwich, D. H., Fogt, R. L., Hodges, K. I., and Walsh, J. E.: A tropospheric assessment of the ERA-40, NCEP, and JRA-25 global reanalyses in the polar regions, Journal of Geophysical Research: Atmospheres, 112, 2007.

Bromwich, D. H., Nicolas, J. P., Monaghan, A. J., Lazzara, M. A., Keller, L. M., Weidner, G. A., and Wilson, A. B.: Central West Antarctica
among the most rapidly warming regions on Earth, Nature Geoscience, 6, 139, 2013.

Butzin, M., Werner, M., Masson-Delmotte, V., Risi, C., Frankenberg, C., Gribanov, K., Jouzel, J., and Zakharov, V. I.: Variations of oxygen-18 in West Siberian precipitation during the last 50 years, Atmospheric Chemistry and Physics, 14, 5853-5869, 2014.

Casado, M., Landais, A., Masson-Delmotte, V., Genthon, C., Kerstel, E., Kassi, S., Arnaud, L., Picard, G., Prie, F., and Cattani, O.:
Continuous measurements of isotopic composition of water vapour on the East Antarctic Plateau, Atmospheric Chemistry and Physics, 16,
15   8521-8538, 2016.

Ciais, P., White, J., Jouzel, J., and Petit, J.: The origin of present-day Antarctic precipitation from surface snow deuterium excess data, Journal of Geophysical Research: Atmospheres, 100, 18917-18927, 1995.

Dansgaard, W.: Stable isotopes in precipitation, Tellus, 16, 436-468, 1964.

Dee, D., Uppala, S., Simmons, A., Berrisford, P., Poli, P., Kobayashi, S., Andrae, U., Balmaseda, M., Balsamo, G., and Bauer, P.: The ERA-
Interim reanalysis: Configuration and performance of the data assimilation system, Quarterly Journal of the Royal Meteorological Society, 137, 553-597, 2011.

Delaygue, G., Jouzel, J., Masson, V., Koster, R. D., and Bard, E.: Validity of the isotopic thermometer in central Antarctica: limited impact of glacial precipitation seasonality and moisture origin, Geophysical Research Letters, 27, 2677-2680, 2000.

Delmotte, M., Masson, V., Jouzel, J., and Morgan, V. I.: A seasonal deuterium excess signal at Law Dome, coastal eastern Antarctica: a
southern ocean signature, Journal of Geophysical Research: Atmospheres, 105, 7187-7197, 2000.

Dittmann, A., Schlosser, E., Masson-Delmotte, V., Powers, J. G., Manning, K. W., Werner, M., and Fujita, K.: Precipitation regime and stable isotopes at Dome Fuji, East Antarctica, Atmospheric Chemistry and Physics, 16, 6883-6900, 2016.

Eisen, O., Frezzotti, M., Genthon, C., Isaksson, E., Magand, O., van den Broeke, M. R., Dixon, D. A., Ekaykin, A., Holmlund, P., and Kameda, T.: Ground-based measurements of spatial and temporal variability of snow accumulation in East Antarctica, Reviews of
Geophysics, 46, 2008.

Favier, V., Agosta, C., Parouty, S., Durand, G., Delaygue, G., Gallée, H., Drouet, A.-S., Trouvilliez, A., and Krinner, G.: An updated and quality controlled surface mass balance dataset for Antarctica, Cryosphere, 7, p. 583-p. 597, 2013.

Favier, V., Krinner, G., Amory, C., Gallée, H., Beaumet, J., and Agosta, C.: Antarctica-Regional Climate and Surface Mass Budget, Current Climate Change Reports, in press.
Fernandoy, F., Meyer, H., and Tonelli, M.: Stable water isotopes of precipitation and firn cores from the northern Antarctic Peninsula region as a proxy for climate reconstruction, The Cryosphere, 6, 313, 2012.

Fernandoy, F., Tetzner, D., Meyer, H., Gacitúa, G., Hoffmann, K., Falk, U., Lambert, F., and MacDonell, S.: New insights into the use of stable water isotopes at the northern Antarctic Peninsula as a tool for regional climate studies, The Cryosphere, 12, 1069-1090, 10.5194/tc-12-1069-2018, 2018.
Fujita, K., and Abe, O.: Stable isotopes in daily precipitation at Dome Fuji, East Antarctica, Geophysical Research Letters, 33, n/a-n/a, 10.1029/2006GL026936, 2006.

Genthon, C., Six, D., Favier, V., Lazzara, M., and Keller, L.: Atmospheric temperature measurement biases on the Antarctic plateau, Journal of Atmospheric and Oceanic Technology, 28, 1598-1605, 2011.

Goursaud, S., Masson-Delmotte, V., Favier, V., Preunkert, S., Fily, M., Gallée, H., Jourdain, B., Legrand, M., Magand, O., and Minster, B.:
A 60-year ice-core record of regional climate from Adélie Land, coastal Antarctica, The Cryosphere, 11, 343-362, 2017.

Grazioli, J., Madeleine, J.-B., Gallée, H., Forbes, R. M., Genthon, C., Krinner, G., and Berne, A.: Katabatic winds diminish precipitation contribution to the Antarctic ice mass balance, Proceedings of the National Academy of Sciences, 114, 10858-10863, 2017.

Holloway, M. D., Sime, L. C., Singarayer, J. S., Tindall, J. C., Bunch, P., and Valdes, P. J.: Antarctic last interglacial isotope peak in response to sea ice retreat not ice-sheet collapse, Nature communications, 7, 2016.
Hoshina, Y., Fujita, K., Nakazawa, F., Iizuka, Y., Miyake, T., Hirabayashi, M., Kuramoto, T., Fujita, S., and Motoyama, H.: Effect of accumulation rate on water stable isotopes of near-surface snow in inland Antarctica, Journal of Geophysical Research: Atmospheres, 119, 274-283, 2014.

Johnsen, S.: Stable isotope homogenization of polar firn and ice, Isotopes and impurities in snow and ice, 1, 1977.

Johnsen, S. J., Clausen, H. B., Cuffey, K. M., Hoffmann, G., Schwander, J., and Creyts, T.: Diffusion of stable isotopes in polar firn and ice:
the isotope effect in firn diffusion, Physics of ice core records, 159, 121-140, 2000.

Jones, J. M., Gille, S. T., Goosse, H., Abram, N. J., Canziani, P. O., Charman, D. J., Clem, K. R., Crosta, X., De Lavergne, C., and Eisenman, I.: Assessing recent trends in high-latitude Southern Hemisphere surface climate, Nature Climate Change, 6, 917, 2016.

Jones, T., Cuffey, K., White, J., Steig, E., Buizert, C., Markle, B., McConnell, J., and Sigl, M.: Water isotope diffusion in the WAIS Divide ice core during the Holocene and last glacial, Journal of Geophysical Research: Earth Surface, 122, 290-309, 2017.

Jouzel, J., and Koster, R. D.: A reconsideration of the initial conditions used for stable water isotope models, Journal of Geophysical Research: Atmospheres, 101, 22933-22938, 1996.

Jouzel, J., Alley, R. B., Cuffey, K., Dansgaard, W., Grootes, P., Hoffmann, G., Johnsen, S. J., Koster, R., Peel, D., and Shuman, C.: Validity of the temperature reconstruction from water isotopes in ice cores, Journal of Geophysical Research: Oceans, 102, 26471-26487, 1997.

Jouzel, J., Hoffmann, G., Koster, R., and Masson, V.: Water isotopes in precipitation:: data/model comparison for present-day and past
climates, Quaternary Science Reviews, 19, 363-379, 2000.

Jouzel, J., Masson-Delmotte, V., Cattani, O., Dreyfus, G., Falourd, S., Hoffmann, G., Minster, B., Nouet, J., Barnola, J.-M., and Chappellaz, J.: Orbital and millennial Antarctic climate variability over the past 800,000 years, science, 317, 793-796, 2007.

Jouzel, J., Delaygue, G., Landais, A., Masson-Delmotte, V., Risi, C., and Vimeux, F.: Water isotopes as tools to document oceanic sources of precipitation, Water Resources Research, 49, 7469-7486, 2013.

Krinner, G., and Werner, M.: Impact of precipitation seasonality changes on isotopic signals in polar ice cores: a multi-model analysis, Earth and Planetary Science Letters, 216, 525-538, 2003.

Krinner, G., Magand, O., Simmonds, I., Genthon, C., and Dufresne, J.-L.: Simulated Antarctic precipitation and surface mass balance at the end of the twentieth and twenty-first centuries, Climate Dynamics, 28, 215-230, 2007.

Kurita, N., Hirasawa, N., Koga, S., Matsushita, J., Steen-Larsen, H. C., Masson-Delmotte, V., and Fujiyoshi, Y.: Influence of large-scale
atmospheric circulation on marine air intrusion toward the East Antarctic coast, Geophysical Research Letters, 43, 9298-9305, 2016.

Laepple, T., Münch, T., Casado, M., Hoerhold, M., Landais, A., and Kipfstuhl, S.: On the similarity and apparent cycles of isotopic variations in East Antarctic snow pits, The Cryosphere, 12, 169, 2018.

Landais, A., Ekaykin, A., Barkan, E., Winkler, R., and Luz, B.: Seasonal variations of 17O-excess and d-excess in snow precipitation at Vostok station, East Antarctica, Journal of Glaciology, 58, 725-733, 2012.

Lee, J. E., Fung, I., DePaolo, D. J., and Otto-Bliesner, B.: Water isotopes during the Last Glacial Maximum: New general circulation model calculations, Journal of Geophysical Research: Atmospheres, 113, 2008.

Lenaerts, J., Den Broeke, M., Berg, W., Meijgaard, E. v., and Kuipers Munneke, P.: A new, high-resolution surface mass balance map of Antarctica (1979–2010) based on regional atmospheric climate modeling, Geophysical Research Letters, 39, 2012.

Lorius, C., Merlivat, L., and Hagemann, R.: Variation in the mean deuterium content of precipitations in Antarctica, Journal of Geophysical
Research, 74, 7027-7031, 1969.

Lorius, C., and Merlivat, L.: Distribution of mean surface stable isotopes values in East Antarctica; observed changes with depth in coastal area, CEA Centre d'Etudes Nucleaires de Saclay, 1975.

Masson-Delmotte, V., Hou, S., Ekaykin, A., Jouzel, J., Aristarain, A., Bernardo, R., Bromwich, D., Cattani, O., Delmotte, M., Falourd, S., Frezzotti, M., Gallée, H., Genoni, L., Isaksson, E., Landais, A., Helsen, M., Hoffmann, G., Lopez, J., Morgan, V., Motoyama, H., Noone,
D., Oerter, H., Petit, J.-R., Royer, A., Uemura, R., Schmidt, G., Schlosser, E., Simões, J., Steig, E. J., Stenni, B., Stievenard, M., Van den Broeke, M. R., Van De Wal, R. S., Van de Berg, W., Vimeux, F., and White, J. W.: A review of Antarctic surface snow isotopic composition: observations, atmospheric circulation, and isotopic modeling*, Journal of Climate, 21, 3359-3387, 2008.

Monaghan, A. J., Bromwich, D. H., and Wang, S.-H.: Recent trends in Antarctic snow accumulation from Polar MM5 simulations, Philosophical Transactions of the Royal Society of London A: Mathematical, Physical and Engineering Sciences, 364, 1683-1708, 2006.

Morgan, V., and van Ommen, T. D.: Seasonality in late-Holocene climate from ice-core records, The Holocene, 7, 351-354, 1997.

Münch, T., Kipfstuhl, S., Freitag, J., Meyer, H., and Laepple, T.: Constraints on post-depositional isotope modifications in East Antarctic firn from analysing temporal changes of isotope profiles, The Cryosphere, 11, 2175, 2017.

Neumann, T. A., and Waddington, E. D.: Effects of firn ventilation on isotopic exchange, Journal of Glaciology, 50, 183-194, 2004.

Nicolas, J. P., and Bromwich, D. H.: New Reconstruction of Antarctic Near-Surface Temperatures: Multidecadal Trends and Reliability of
Global Reanalyses*,+, Journal of Climate, 27, 8070-8093, 2014.

Noone, D., and Simmonds, I.: Sea ice control of water isotope transport to Antarctica and implications for ice core interpretation, Journal of Geophysical Research: Atmospheres, 109, 2004.

Pfahl, S., and Sodemann, H.: What controls deuterium excess in global precipitation?, Climate of the Past, 10, 771, 2014.

Ritter, F., Steen-Larsen, H. C., Werner, M., Masson-Delmotte, V., Orsi, A., Behrens, M., Birnbaum, G., Freitag, J., Risi, C., and Kipfstuhl,
S.: Isotopic exchange on the diurnal scale between near-surface snow and lower atmospheric water vapor at Kohnen station, East Antarctica, The Cryosphere, 10, 1647-1663, 10.5194/tc-10-1647-2016, 2016.

Roeckner, E., Bäuml, G., Bonaventura, L., Brokopf, R., Esch, M., Giorgetta, M., Hagemann, S., Kirchner, I., Kornblueh, L., and Manzini, E.: The atmospheric general circulation model ECHAM 5. PART I: Model description, 2003.

Schlosser, E., Oerter, H., Masson-Delmotte, V., and Reijmer, C.: Atmospheric influence on the deuterium excess signal in polar firn:
implications for ice-core interpretation, Journal of glaciology, 54, 117-124, 2008.

Schlosser, E., Stenni, B., Valt, M., Cagnati, A., Powers, J. G., Manning, K. W., Raphael, M., and Duda, M. G.: Precipitation and synoptic regime in two extreme years 2009 and 2010 at Dome C, Antarctica–implications for ice core interpretation, Atmospheric Chemistry and Physics, 16, 4757-4770, 2016.

Schlosser, E., Dittmann, A., Stenni, B., Powers, J. G., Manning, K. W., Masson-Delmotte, V., Valt, M., Cagnati, A., Grigioni, P., and Scarchilli, C.: The influence of the synoptic regime on stable water isotopes in precipitation at Dome C, East Antarctica, The Cryosphere, 11, 2345, 2017.

Schmidt, G. A., LeGrande, A. N., and Hoffmann, G.: Water isotope expressions of intrinsic and forced variability in a coupled ocean-atmosphere model, Journal of Geophysical Research: Atmospheres, 112, 2007.

Schoenemann, S. W., Steig, E. J., Ding, Q., Markle, B. R., and Schauer, A. J.: Triple water-isotopologue record from WAIS Divide, Antarctica: Controls on glacial-interglacial changes in 17Oexcess of precipitation, Journal of Geophysical Research: Atmospheres, 119, 8741-8763, 2014.

Schroeter, S., Hobbs, W., and Bindoff, N. L.: Interactions between Antarctic sea ice and large-scale atmospheric modes in CMIP5 models, The Cryosphere, 11, 789, 2017.

Sime, L., Wolff, E., Oliver, K., and Tindall, J.: Evidence for warmer interglacials in East Antarctic ice cores, Nature, 462, 342, 2009.

Smith, S. R., and Stearns, C. R.: Antarctic pressure and temperature anomalies surrounding the minimum in the Southern Oscillation index, Journal of Geophysical Research: Atmospheres, 98, 13071-13083, 1993.

Sokratov, S. A., and Golubev, V. N.: Snow isotopic content change by sublimation, Journal of Glaciology, 55, 823-828, 2009.

Stammerjohn, S., Martinson, D., Smith, R., Yuan, X., and Rind, D.: Trends in Antarctic annual sea ice retreat and advance and their relation to El Niño–Southern Oscillation and Southern Annular Mode variability, Journal of Geophysical Research: Oceans, 113, 2008.

Steen-Larsen, H., Masson-Delmotte, V., Hirabayashi, M., Winkler, R., Satow, K., Prié, F., Bayou, N., Brun, E., Cuffey, K., and Dahl-Jensen, D.: What controls the isotopic composition of Greenland surface snow?, Climate of the Past, 10, 377, 2014.

Steen-Larsen, H., Risi, C., Werner, M., Yoshimura, K., and Masson-Delmotte, V.: Evaluating the skills of isotope-enabled General Circulation Models against in-situ atmospheric water vapor isotope observations, Journal of Geophysical Research: Atmospheres, 2016.

Steiger, N. J., Steig, E. J., Dee, S. G., Roe, G. H., and Hakim, G. J.: Climate reconstruction using data assimilation of water isotope ratios from ice cores, Journal of Geophysical Research: Atmospheres, 122, 1545-1568, 2017.

Stenni, B., Scarchilli, C., Masson-Delmotte, V., Schlosser, E., Ciardini, V., Dreossi, G., Grigioni, P., Bonazza, M., Cagnati, A., Karlicek, D., Risi, C., Udisti, R., and Valt, M.: Three-year monitoring of stable isotopes of precipitation at Concordia Station, East Antarctica, The Cryosphere, 10, 2415-2428, 10.5194/tc-10-2415-2016, 2016.

Stenni, B., Curran, M. A., Abram, N. J., Orsi, A., Goursaud, S., Masson-Delmotte, V., Neukom, R., Goosse, H., Divine, D., and Van Ommen, T.: Antarctic climate variability on regional and continental scales over the last 2000 years, Climate of the Past, 13, 1609, 2017a.

Stenni, B., Curran, M. A. J., Abram, N. J., Orsi, A., Goursaud, S., Masson-Delmotte, V., Neukom, R., Goosse, H., Divine, D., van Ommen, T., Steig, E. J., Dixon, D. A., Thomas, E. R., Bertler, N. A. N., Isaksson, E., Ekaykin, A., Frezzotti, M., and Werner, M.: Antarctic climate variability at regional and continental scales over the last 2,000 years, Clim. Past Discuss., 2017, 1-35, 10.5194/cp-2017-40, 2017b.

Sturm, C., Zhang, Q., and Noone, D.: An introduction to stable water isotopes in climate models: benefits of forward proxy modelling for paleoclimatology, Climate of the Past, 6, 115-129, 2010.

Touzeau, A., Landais, A., Stenni, B., Uemura, R., Fukui, K., Fujita, S., Guilbaud, S., Ekaykin, A., Casado, M., and Barkan, E.: Acquisition of isotopic composition for surface snow in East Antarctica and the links to climatic parameters, The Cryosphere, 10, 837-852, 2016.

Touzeau, A., Landais, A., Morin, S., Arnaud, L., and Picard, G.: Numerical experiments on isotopic diffusion in polar snow and firn using a multi-layer energy balance model, Geosci. Model Dev. Discuss., 2017, 1-58, 10.5194/gmd-2017-217, 2017.

Turner, J.: The El Niño–Southern Oscillation and Antarctica, International Journal of climatology, 24, 1-31, 2004.

Turner, J., Colwell, S. R., Marshall, G. J., Lachlan-Cope, T. A., Carleton, A. M., Jones, P. D., Lagun, V., Reid, P. A., and Iagovkina, S.: The SCAR READER project: toward a high-quality database of mean Antarctic meteorological observations, Journal of Climate, 17, 2890-2898, 2004.

Uemura, R., Masson-Delmotte, V., Jouzel, J., Landais, A., Motoyama, H., and Stenni, B.: Ranges of moisture-source temperature estimated from Antarctic ice cores stable isotope records over glacial-interglacial cycles, Climate of the Past, 8, 1109-1125, 2012.

Uppala, S. M., Kållberg, P., Simmons, A., Andrae, U., Bechtold, V. d., Fiorino, M., Gibson, J., Haseler, J., Hernandez, A., and Kelly, G.: The ERA-40 re-analysis, Quarterly Journal of the Royal Meteorological Society, 131, 2961-3012, 2005.

Waddington, E. D., Steig, E. J., and Neumann, T. A.: Using characteristic times to assess whether stable isotopes in polar snow can be reversibly deposited, Annals of Glaciology, 35, 118-124, 2002.

Wang, Y., Ding, M., van Wessem, J., Schlosser, E., Altnau, S., van den Broeke, M. R., Lenaerts, J. T., Thomas, E. R., Isaksson, E., and Wang, J.: A comparison of Antarctic Ice Sheet surface mass balance from atmospheric climate models and in situ observations, Journal of Climate, 2016.

Werner, M., Langebroek, P. M., Carlsen, T., Herold, M., and Lohmann, G.: Stable water isotopes in the ECHAM5 general circulation model: Toward high-resolution isotope modeling on a global scale, Journal of Geophysical Research: Atmospheres, 116, 2011.

Whillans, I., and Grootes, P.: Isotopic diffusion in cold snow and firn, Journal of Geophysical Research: Atmospheres, 90, 3910-3918, 1985.