# Peer review of "Water stable isotopes spatio-temporal variability in Antarctica in 1960-2013: observations and simulations from the ECHAM5-wiso atmospheric general circulation model"

_Climate of the Past, 2017_

## Referee Comment (RC1) · Anonymous Referee #1 · 23 Jan 2018

General Comments This paper addresses novel and relevant scientific questions within the scope of CP. The authors use an impressive collection of data to assess the skill of the ECHAM5-wiso model. The main scope of the paper is to evaluate spatial, seasonal and interannual $\delta$18O-temperature relationships, as well as deuterium excess and $\delta$18O phasing. This information is important for correctly interpreting certain climate records in Antarctica, especially when using shallow ice core records of a few decades length.

Minor revisions are required for publication, as well as one major revision and/or clarification.

Specific Comments * My biggest concern is how the authors addressed water isotope diffusion in shallow ice core records. The majority of diffusion occurs in the upper ∼10-20 meters of the ice sheet, thus this will have a significant effect on the results of this study (i.e. for ice core data from 1979-2013, or for any data extending beyond a few years in length). Can the authors clarify whether any consideration was given to the attenuation of the seasonal and multi-year variations due to diffusion?

For example, at a typical inland West Antarctic site (mean annual temp = -30.3C, accumulation = 0.23 m/yr), the annual d18O and dD signal amplitudes will decrease by about ∼50% in 30 years (calculated using a Johnsen firn model and a Herron-Langway densification model). For a colder site (temp = -40.3C, accum = 0.12 m/yr), the amplitudes decrease by 67% in 30 yrs. And for a warmer site with high accum (temp = -25.3C, accum = 0.38 m/yr), the amplitudes are decreased by 37% in 30 years. These are quick calculations, but show the importance of diffusion.

Could firn diffusion be the cause of model-data mismatch? If so, and I think this is the case, the authors should either make these calculations and include the corrections in the paper, or state a few examples of signal attenuation for different temperatures and accumulation that are relevant to the ice core sites used in the paper. On the other hand, if I have misunderstood the results, please provide clarifications and explain why.

* Please explain "nudging", and perhaps use different wording in the paper. While this may be common terminology, it is not immediately clear what it means, nor does it appear to be defined in the main text of the paper. I would also suggest a short, 1-sentence explanation in the introduction that explains the relevance of slopes for ice core isotope-temperature relationships, etc.

* Can you please confirm that for any averaged isotope data, that the same averaging was done in the model. If not, please state why, and how this could affect results. Also, please provide a clarification on how averaging could reduce the amplitude of the

observed seasonal and multi-year signals.

* The authors state once that "... a stationary isotope-temperature slope cannot be applied for the climatic interpretation of Antarctic ice core." (pg 3, line 1-2). This is an important point. I think this point should be made in the Conclusion as well, specifically that the results of this study (or atleast some of the results) may not hold in the deeper past (greater than a few decades). Please be clear in your assessment of the relevance for paleoclimate interpretations. This has the potential to be misunderstood.

* In many instances, the citations are dated. There are many more recent studies that should be cited in this manuscript. I encourage the authors to provide citations of more recent studies.

Technical Corrections

pg 2 line 7 - nudged? please explain what this means somewhere in the introduction, and possibly change the wording.

pg 2 line 15-17 - the description is unclear

pg 2 line 28 - slopes? "We show that local spatial or seasonal slopes" the relevance of slopes should be defined in the introduction so certain readers are not left wondering what this means

pg 3 line 6 - "This work valuates" - evaluates?

pg 4 line 4 - consider saying "the hydrologic cycle" rather than "water cycle"

pg 4 line 6-8: "Their climate interpretation is however limited, first by the alteration of the signal due to deposition and post-deposition processes, and second by the complexity of all parameters affecting the Antarctic snowfall isotopic composition" —— cite sources.

For Antarctica, one of the more in-depth studies of "post-depositional processes" is Jones et al., 2017 "Water isotope diffusion in the WAIS Divide ice core during the

Holocene and last glacial"Âădoi:10.1002/2016JF003938. Also provide citations for depositional processes and "complexity of all parameters" - perhaps you mean isotopic recharge, etc?

pg 4 line 6: "Their"? Who are they?

pg 4 line 6-8: "Their climate interpretation is however limited, first by the alteration of the signal due to deposition and post-deposition processes, and second by the complexity of all parameters affecting the Antarctic snowfall isotopic composition." - please use another word other than limited. I think you mean to say that post depositional processes alter the original signal, which must be accounted for in climate interpretations?

pg 4 line 13-16, "However, recent studies cast doubt on this assumption, evidencing isotopic exchanges between the Antarctic snow surface and the atmosphere associated with snow metamorphism occurring at the diurnal and sub-annual scales (Ritter et al., 2016;Casado et al., 2016;Touzeau et al., 2016)." —- consider citing Steen-Larsen et al.?

pg 4 line 16-18: Again, the most recent diffusion study I have seen is Jones et al. 2017, it provides important information with an Antarctic perspective, and it should be cited here. There are important points in Jones et al. 2017 that improve on Sigfus Johnsen's 2000 paper.

pg 4 line 18-19: "So far, the overall importance of such post-deposition processes on the alteration of the initial precipitation signals cannot be quantified." — This is not true. The alteration of the initial precip signal can be determined reasonably well by fitting a Gaussian to the data. Similarly, the Johnsen firn diffusion model, to the first order, is also a reasonable model for signal alteration. However, there are physical mechanisms that are still not understood.

pg 5 line 7: "$\delta$18O and deuterium", should be "$\delta$18O and $\delta$D (D refers to deuterium)" - something like this would be more consistent

pg 5 line 19: is this really the only exception??? "with one exception (Lee et al., 2008)."

pg 6 line 2: what is motivating "interannual scale" research, I suggest mentioning why this matters in the introduction

pg 7 line 11: "cautious" - caution?

pg 7 line 23: nudged to, what does this mean?

pg 8 line 10: just use dD rather than deuterium to avoid confusion, and make sure to define D, see above comment

pg 8 line 18-19: unclear what this means, "the averaging period may be heterogeneous, including subintervals within 1960-2013, or longer time periods."

pg 11 line 3-5: "While this bias is small (less than 2°C)" - this is not small, please re-word

pg 11 line 8: "above the ice sheet" - what does this mean?

pg 12 line 1-2: "despite the nudging technique (not shown)." - what exactly is not shown? As mentioned previously, please explain nudging.

pg 14 line 21-24: "The largest deviations are encountered in coastal regions, where either the model resolution is too low to resolve advection and boundary layer processes (e.g. katabatic winds), or where post-deposition processes may have a larger influence." — Why would post deposition processes have a larger influence? Larger compared to what?

pg 15 line 23: "We have calculated the mean amplitude of the $\delta$18O sub-annual variations" - please clarify what amplitude you are calculating? Monthly?

pg 16 line 2-4: "ECHAM5-wiso underestimates the seasonal amplitude (by 14 to 69%) when compared to precipitation data, but overestimates the seasonal amplitude when compared to ice core data (from 11 to 71%)." — could the seasonal amplitude overestimation in the model be related to diffusion? These overestimations are similar to the annual signal attenuation examples I gave above.

pg 18 line 10-11: this needs more explaining and/or a citation - "Due to the temperature dependency of equilibrium fractionation coefficients, dexcess increases when temperature decreases."

pg 19 lines 16-18: "ECHAM5-wiso systematically underestimates the d-excess mean seasonal amplitude when compared with precipitation data, while it systematically overestimates it when compared with ice core data." could the overestimation be due to diffusion, which would decrease the dxs amplitude? what is the range of overestimation (in percent)?

pg 19 lines 26-27, pg 20 lines 1-2: "ECHAM5-wiso always underestimates seasonal amplitude of $\delta$ 18O and d-excess in precipitation but always overestimates seasonal amplitude of $\delta$ 18O and d-excess in firn/ice cores (Table 4 and 8). Differences between the model and firn/core data might be due to diffusion processes, but no clear reason can be given for the other isotopic biases." - it is not accurate to say "might be due to diffusion", because diffusion must have a substantial effect

---

## Referee Comment (RC2) · Anonymous Referee #2 · 25 Feb 2018

This paper presents updated data and new model simulations for Antarctic isotope distributions. This is a worthy contribution, and appropriate for publication in Climate of the Past.

I do have a several criticisms and technical points.

**Much of the conclusions in the paper are not new, and this should be made clear. For example, in the abstract, it is stated that "local spatial or seasonal slopes are not a correct surrogate for inter-annual temporal slopes". This is a very old result, and

doesn't belong in the abstract. Similarly, the phasing between deuterium excess and d18O has been examined thoroughly in previous work. What is new here? Throughout the paper, I would like to see better delineation between new results and reiteration of old results.

\*\*Too little reference is given to primary results, and it is difficult to determine what data are actually being used. Reference is given to the Stenni et al compilation of ice core data, but it is not clear which of the many records in that data set are actually used. For example, there are multiple cores from Dronning Maud Land and in the vicinity of Byrd Station and the West Antarctic ice sheet divide, but only a few locations are shown on the map? What are the primary data sources? Which of the cores (by latitutde/longitude) are included here? Which are excluded, and why?

\*\*For the temperature data, it is stated that the READER data are used, and that "AWS" data are used for Byrd and Dome C. For Byrd, the best data are the updated record for Byrd, from Bromwich et al,. 2013. This should be used! It is not clear whether it is, nor not.

\*\*There is reference to diffusion, but it's importance is not taken into account.

"ECHAM5-wiso underestimates the seasonal amplitude (by 14 to 69%) when compared to precipitation data, but overestimates the seasonal amplitude when compared to ice core data (from 11 to 71% )." and "At Dome C, ECHAM5-wiso underestimates the standard deviation of temperature, but strongly overestimates the standard deviation of $\delta$18O."

Can diffusion explain these differences? It would be quite straightforward to evaluate whether this is likely. I suspect such difference can be explained entirely by diffusion, as has been pointed out in numerous previous papers. Also, it should be explained here whether EMCHAM-5 does a better or a worse job that ECHAM-4 or other models (GISS, for example). In other words, does ECHAM5 represent an improvement here, or not?

CPD

\*\*It is stated that there is an abrupt warming from 1978 to 1979, \*possibly\* caused by a discontinuity in the European Reanalyses (ERA) linked to the assimilation of remote sensing data starting in 1979..." This is not just "possible" – it's certain. It is very well established that the ERA-interim data are essentially useless prior to 1979. I don't think including the few data-model comparison prior to that time period is useful.

\*\*References in general are inadequate. For example, it is noted that "recent studies cast doubt on this assumption [that isotope can be interpreted as precipitation-weighted deposition signal" and a few recent papers are cited (Ritter et al., 2016;Casado et al., 2016;Touzeau et al., 2016). Those citations are good, but the original idea goes back at least to Waddington (2002: doi 10.3189/172756402781817004), and a number of papers by Steen-Larsen and others have also discussed this, some years prior to 2016. Such papers should be included. Another example is that reference is made to the impact of sea ice on isotopes, but only the very recent paper by Holloway et al 2016 is cited. This idea goes back to at least 1983 (e.g. Bromwich and Weaver, doi:10.1038/301145a0. Another key citation is Noone et al., 2004: doi: 10.1029/2003JD004228.

\*\*There is much discussion about the phase lag between deuterium excess and d18O, but no discussion of why this is important. As the authors will be aware, Pfahl and Sodemann [2014] have suggested a completely different idea (with respect to humidity) about this than the conventional one (about the delay between SST and air temperature). I realize that the present paper is not claiming to solve this puzzle, but some reference to the scientific context would greatly improve the paper.

\*\* The figures could use improvement. Especially, there should be a variable name and units on the color bars of the various maps. For example, Figure 10 should read "lag, in months", on the color axis.

\*\*The calculation of statistical significance, throughout the paper, is not clear. Is autocorrelation accounted for in stating that $p < 0.05$?

\*\*Several recent papers have demonstrated that the logarithmic form of deuterium excess is a much more reliable and robust measure than the traditional linear calculation. The paper really ought to look at this as well. See Markle and others (doi: 10.1038/ngeo2848) and Uemera et al., 2012 (doi: 10.5194/cp-8-1109-2012), Dutsch et al. 2017, 10.1002/2017JD027085

---

## Author Comment (AC1) · 13 Apr 2018

"This paper presents updated data and new model simulations for Antarctic isotope distributions. This is a worthy contribution, and appropriate for publication in Climate of the Past. I do have a several criticisms and technical points."

We thank the second referee for reviewing our manuscript. In the following lines, we answer to the reviewer comment by comment.

"**Much of the conclusions in the paper are not new, and this should be made clear."

[Figure]

Earlier studies focused on individual locations or specific time intervals have evidenced temporal changes in isotope-temperature relationships (e.g. different decadal to seasonal relationships, Masson-Delmotte et al., 2003), or different relationships in warmer than today climatic conditions (e.g. Sime et al., 2009). Our study is a step forward through a systematic quantitative analysis for all Antarctica, confronting seasonal, annual and time-averaged linear relationships. While our key findings are indeed not new, we confirm and expand the lines of evidence supporting these findings through a broader database and a systematic model-data comparison.

"For example, in the abstract, it is stated that "local spatial or seasonal slopes are not a correct surrogate for inter-annual temporal slopes". This is a very old result, and doesn't belong in the abstract."

Indeed, several earlier studies have challenged the use of spatial isotope-temperature relationships as a surrogate for temporal isotope-temperature relationships as it is crucial to assess the uncertainty associated with temperature reconstructions from ice core records. They evidenced spatial variations in this relationship (see for instance a review in Table 1 from Stenni et al. (2016) for the observed $\delta$18O-temperature linear relationship at selected sites) as well as temporal variations. It suggests that different slopes have to be applied to reconstruct temperatures at different locations. Within the Antarctica2k effort, different statistical methods were thus tested to propose Antarctic temperature reconstructions from regional isotopic syntheses (Stenni et al., 2017a), expanding the pioneer work of Schneider et al. (2006). However, many of these earlier studies were based on site specific studies, or pure modeling work, without combining systematically available evidence, or without a systematic benchmarking of models against all available data. While our conclusions are not new, we provide a more systematic approach to such issues than in earlier studies. We have made clear in the revised manuscript that we are revisiting scientific questions using a more comprehensive database (combining precipitation and shallow ice core data) and with a systematic benchmarking of one isotope-enabled model which was run at higher spatial resolution

then in earlier modeling studies, and in a nudged mode facilitating the comparison with short records. We stress that our work is the first to provide a quantitative synthesis of spatial and temporal (seasonal and annual) $\delta$18O-temperature linear relationship (slopes and correlation coefficients) for all Antarctica.

"Similarly, the phasing between deuterium excess and d18O has been examined thoroughly in previous work. What is new here? Throughout the paper, I would like to see better delineation between new results and reiteration of old results." - Most of model-data comparisons for Antarctica have been focused on model skills for spatial gradients in $\delta$18O or $\delta$D. Our study is also the first one to benchmark ECHAM5-wiso outputs systematically using a synthesis of precipitation and ice core data, looking at spatio-temporal variations. In the abstract, we have clarified the novelty of our approach for the $\delta$18O-temperature temperature and the d-excess - $\delta$18O phasing: p.2 l.28 by writing: "Our study confirms key findings obtained from site specific focused studies, using a database of all existing records and a systematic model-data comparison."

"**Too little reference is given to primary results, and it is difficult to determine what data are actually being used. Reference is given to the Stenni et al compilation of ice core data, but it is not clear which of the many records in that data set are actually used. For example, there are multiple cores from Dronning Maud Land and in the vicinity of Byrd Station and the West Antarctic ice sheet divide, but only a few locations are shown on the map? What are the primary data sources? Which of the cores (by latitutde/longitude) are included here? Which are excluded, and why?" The data extracted from the database compiled by the Antarctica2k group were restricted to the period 1979-present, resulting in the selection of 101 ice core data compared to the 122 original ice core data reported in Stenni et al (2017). The primary sources, latitude, longitude and covered periods of all data from our database are detailed in Supplementary Material. We have clarified this in our revised Section 2.1.3;

"(1) 101 high resolution ice core records, including 79 annually resolved records,

and 18 records with sub-annual resolution (including 5 records with both $\delta18O$ and $\delta D$ data). These data have been extracted from the Antarctica2k data synthesis (Stenni et al., 2017b) with a filter for records spanning the interval 1979-2013, thus restricting the original 122 ice cores to a resulting 101 ice cores data. Primary data sources, geographical coordinates and covered periods are reported in Supplementary Material Table S1. (2) average surface snow isotopic composition data compiled by Masson-Delmotte et al. (2008) (available on http://www.lsce.ipsl.fr/Phocea/Pisp/index.php?nom=valerie.masson), expanded with datasets from Fernandoy et al. (2012); in this case, the averaging period is based on different periods, with potential not continuous record (see Supplementary Material Table S1). (3) precipitation records extracted from the International Atomic Energy Agency / Global Network of Isotopes in Precipitation (IAEA / GNIP) network (IAEA/WMO, 2016) with monthly records available for 4 Antarctic Stations, complemented by daily records for 4 Antarctic stations from individual studies. Precipitation records from Vostok are available but have excluded from our analysis, due to a too small number of measurements (29). (see Supplementary Material Table S1).

"**For the temperature data, it is stated that the READER data are used, and that "AWS" data are used for Byrd and Dome C. For Byrd, the best data are the updated record for Byrd, from Bromwich et al,. 2013. This should be used! It is not clear whether it is, nor not." We thank the referee for suggesting to use of the temperature reconstruction of Bromwich et al. (2013). We have repeated our analysis replacing the AWS temperature data with the reconstructed Byrd station data. It made our conclusions more robust, as we found the same results with this longer time series, i.e. (i) a warm model bias (Table 1), (ii) no sharp increase from 1978 to 1979 in the data contrary to the model (Fig. 2c), (iii) stronger correlation coefficient and slopes between observed and simulated temperatures (Table 2, more coherent with the results obtained at other stations.

In revised Table 1, the observed standard deviation changed by 0.1°C (from 1.2 to 1.3

°C), and thus our conclusions about spatial model biases remain unchanged. In Table 2, we can have completed the line for Byrd station: Period 1960-2013 Period 1979-2013 slope (in °C °C-1) r p-value slope (in °C °C-1) r p-value … Byrd 1.1 (0.15) 0.7 <0.001 0.8 (0.12) 0.8 <0.001

Figure 2.c was also modified following this change.

In the text, we revised the description of the used temperature data in Section 2.1.1 p.7 l.17: "Due to the short duration of surface station records for the 90-180° W sector, we have added data from the automatic weather station (hereafter, AWS) of Dome C, but we have used it with caution as these records are associated with a warm bias in thermistor measurements due to solar radiation when the wind speed is low (Genthon et al., 2011). Finally, we extracted the reconstruction of temperature for Byrd station by Bromwich et al. (2013), based on AWS data and infilled with reanalysis data." In Section 3.1.1, we modified only one sentence of our results, p.13 l.15: "We observe that the model reproduces the amplitude of inter-annual variations, with a tendency to underestimate the variations as shown by model-data slopes from 0.7 to 1°C per °C."

"**There is reference to diffusion, but it's importance is not taken into account. "ECHAM5-wiso underestimates the seasonal amplitude (by 14 to 69%) when compared to precipitation data, but overestimates the seasonal amplitude when compared to ice core data (from 11 to 71% )." and "At Dome C, ECHAM5-wiso underestimates the standard deviation of temperature, but strongly overestimates the standard deviation of $\delta 18O$." Can diffusion explain these differences? It would be quite straightforward to evaluate whether this is likely. I suspect such difference can be explained entirely by diffusion, as has been pointed out in numerous previous papers."

We agree that post-deposition effects can explain the overestimation by ECHAM5-wiso of the seasonal amplitude recorded in ice cores, while it underestimates the seasonal amplitude recorded in precipitations. This comment was also made by the first referee. We repeat here the answer to the comment of Reviewer 1. This is an important issue

for the quantitative comparison of seasonal isotopic amplitudes in precipitation model outputs with firn data, potentially affected by diffusion (Johnsen et al, 2000).

The theory of Whillans and Grootes (1985) about isotopic diffusion in firn based on diffusional vapor flux through firn pore spaces, appears compatible with the estimated loss of seasonal amplitude through depth in diverse sites (e.g. Cuffey and Steig, 1998), but this validation is limited by the lack of comprehensive datasets (monitoring of precipitation isotopic composition over multiple years to be compared with the firn records), as well as uncertainties on key parameters. Accounting for firn ventilation effects on sublimation and condensation and disequilibrium between pore-space vapor and snow grains Neumann and Waddington (2004) shows more rapid isotopic changes in the upper few firn meters at low-accumulated site than explained by Whillans and Grootes (1985). Recent studies in Greenland have further evidenced changes in surface snow isotopic signal in between precipitation events, attributed to water vapour exchange associated with snow metamorphism (Steen-Larsen et al., 2014). While there has long been evidence for a loss of seasonal amplitude with depth, a reliable quantification of the effect of diffusion in firn for all available Antarctic records is currently out of reach, and thus cannot be applied to remove this bias in model-data comparisons of seasonal amplitudes.

Here, we just focus on all sub-annual records available from our database. For each record, we calculate the ratio of the seasonal amplitude estimated from the first three first seasonal cycles to the mean seasonal amplitude for all available seasonal cycles along this core. If the seasonal cycle of precipitation isotopic composition was constant through time, the interplay of diffusion and the averaging of seasonal amplitude over multiple seasonal cycles would make this ratio as an indication of the loss of seasonal amplitude, assuming that the amplitude of the first three seasonal cycles is representative of that of precipitation (Table 1 in Supplementary Material). We obtain a mean ratio of 1.40 $\pm$ 0.47. No significant relationship can be identified between this ratio and the corresponding estimated annual accumulation rates (see enclosed Figure 1). We

note that a ratio lower than 1 is obtained in five ice cores, including one with a mean annual accumulation of 15 cm w.e. y-1; this situation is understood to result from inter-annual variations in precipitation isotopic composition and/or diffusion characteristics in the upper firn. Our simple empirical calculation shows that we cannot exclude a loss of seasonal amplitude in the firn data used to estimate the average seasonal isotopic amplitude, due to post-deposition processes; it also shows that the average seasonal amplitude obtained in our firn multi-year records may be affected by an average loss of about 70% (1/1.4) of the seasonal amplitude recorded during the first three years of each firn core. We cannot assess directly the potential distortion of the seasonal amplitude from the initial precipitation to the snow surface due to the lack of systematic precipitation, surface snow and firn multi-year monitoring datasets.

These elements are reported in our revised manuscript, in Section 3.2.2 "$\delta$18O seasonal amplitude", p.16 l.1 as follows: "In order to quantify post-deposition effects in ice cores, we calculated the ratio of the three first seasonal amplitudes by the mean seasonal amplitude in sub-annual ice cores (See Supplementary Information S2). We find a mean ratio of 1.40 $\pm$ 0.47. We explored whether this ratio was related to annual accumulation rates (See Supplementary Information S3), without any straightforward conclusion. We also observe that five ice cores depict a ratio lower than 1, including one with a mean yearly accumulation of 15 cm w.e. y-1, a feature which may arise from inter-annual variability in the precipitation seasonal amplitude or in post-deposition processes. This empirical analysis shows that a loss of seasonal amplitude due to post-deposition processes is likely in most cases, with an average loss of the seasonal amplitude of approximately 70% compared to the amplitude recorded in the upper part of the firn cores (first three years). " Here, S2 and S3 correspond to the attached Figure 1 and Table 1 attached to this response.

We specified that the overestimation of the mean $\delta$18O seasonal amplitude by ECHAM5-wiso compared to ice core data could be due to post-deposition effects: p.16 l.12 "The overestimation when comparing with ice core data could be due to the attenuation of signal by post-deposition effects (as aforementioned) rather than a model bias."
p.20 l.23: "Again, we cannot rule out a loss of amplitude in ice core data compared to
the initial precipitation signal, due to the temporal resolution and to post-deposition ef-
fects.the overestimation when comparing against ice core data, i.e. an attenuation in
the data by post-deposition effects."

"Also, it should be explained here whether EMCHAM-5 does a better or a worse job
that ECHAM-4 or other models (GISS, for example). In other words, does ECHAM5
represent an improvement here, or not?" As explained in Werner et al. (2011) the
ECHAM5 model includes a number of general model improvement as compared to
ECHAM4. For the representation of the atmospheric water cycle, especially in polar
regions like Antarctica, Martin Werner rated the separate prognostic equations were
for cloud ice and cloud liquid water, a new flux-form semi‐Lagrangian transport
scheme for all vapour, liquid water and ice in the atmosphere, and a different cloud
micro-physical scheme as most important. Unfortunately, we have never published a
present-day comparison study of the different isotope models (it was planned with the
SWING2 project several years ago, but never realised). But for an LGM-PI comparison,
please see Figure 5 of Jasechko et al. (2015) for the performance of the different
models over Antarctica.

"**It is stated that there is an abrupt warming from 1978 to 1979, *possibly* caused by
a discontinuity in the European Reanalyses (ERA) linked to the assimilation of remote
sensing data starting in 1979..." This is not just "possible" – it's certain. It is very well
established that the ERA-interim data are essentially useless prior to 1979. I don't
think including the few data-model comparison prior to that time period is useful." We
agree that the he abrupt discontinuity from 1978-1979 in ERA reanalyses has been
well established (Bromwich et al., 2007), so that most climate modelers have focused
their analyses of simulations nudged to ERA reanalyses models to the period starting
in 1979 (e.g. Lenaerts et al., 2012). We also note that some work on ice core data has
explored historical reanalyses as a source of climate information. Finally, the nudged

isotopic simulations give access to a comparison with ice core records independently of the data assimilated in reanalyses (e.g. accumulation and water stable isotopes), as an independent source of. While the finding that reanalyses are not reliable prior to remote sensing of sea ice (1979) is not new, we believe that this is still a valuable information, especially for the research community working on proxy records in natural archives.

"**References in general are inadequate. For example, it is noted that "recent studies cast doubt on this assumption [that isotope can be interpreted as precipitation weighted deposition signal" and a few recent papers are cited (Ritter et al., 2016;Casado et al., 2016;Touzeau et al., 2016). Those citations are good, but the original idea goes back at least to Waddington (2002: doi 10.3189/172756402781817004), and a number of papers by Steen-Larsen and others have also discussed this, some years prior to 2016. Such papers should be included. Another example is that reference is made to the impact of sea ice on isotopes, but only the very recent paper by Holloway et al 2016 is cited. This idea goes back to at least 1983 (e.g. Bromwich and Weaver, doi:10.1038/301145a0. Another key citation is Noone et al., 2004: doi: 10.1029/2003JD004228." We thank Reviewer 1 and 2 for suggesting to add references to pioneer studies. Especially here, we thank reviewer 2 for directing us towards the studies of Bromwich and Weaver (1983), and Noone and Simmonds (2004) that we were not aware of. We thus expanded the citation of (Holloway et al., 2016) with those given, p.15 l.17: (Bromwich and Weaver, 1983;Noone and Simmonds, 2004;Holloway et al., 2016). Also, in response to Reviewer 1, we updated our citations : - In the introduction: o p.4 l.6 (Schoenemann et al., 2014) o p.4 l.10 (Sokratov and Golubev, 2009;Jones et al., 2017;Münch et al., 2017;Laepple et al., 2018) o p.4 l.14 (Smith and Stearns, 1993;Turner, 2004;Stammerjohn et al., 2008;Schroeter et al., 2017) o p.4 l.21 (Hoshina et al., 2014) o p.4 l.25 (Grazioli et al., 2017) o p.5 l.14 (Fernandoy et al., 2018) - Section 3.2.1 o p.16 l.8 (Waddington et al., 2002;Neumann and Waddington, 2004) - Section 4.1 o p.24 l.1 (Sturm et al., 2010;Steiger et al., 2017) - Section 4.2 o p.24 l.8 (Jouzel et al., 2013;Pfahl and Sodemann, 2014;Kurita et al., 2016) o p.24 l. 9

(Schlosser et al., 2017)

"**There is much discussion about the phase lag between deuterium excess and d18O, but no discussion of why this is important. As the authors will be aware, Pfahl and Sodemann [2014] have suggested a completely different idea (with respect to humidity) about this than the conventional one (about the delay between SST and air temperature). I realize that the present paper is not claiming to solve this puzzle, but some reference to the scientific context would greatly improve the paper." Pfahl and Sodemann (2014) analysed global observational deuterium excess records and a diagnostic of moisture sources from atmospheric reanalyses, and found an empirical linear relationship between seasonal variations in deuterium excess in precipitation and relative humidity at the moisture sources. This finding was challenged by related studies of Antarctic precipitation data (Dittmann et al., 2016;Schlosser et al., 2017), calling for further investigations. Several earlier studies had also explored the seasonal relationship between d-excess and $\delta$18O in Antarctic precipitation and snow samples, showing different results in central Antarctica (an antiphase, e.g. Stenni et al., 2016) and coastal Antarctica (e.g. Ciais et al., 1995;Delmotte et al., 2000). While an anti-phase is expected if d-excess would only respond to changes in condensation temperature (through the dependency on temperature of the equilibrium fractionation coefficients and the resulting local meteoric water line), a different phase relationship is understood to reflect a signal related to changes in moisture origin and/or evaporation conditions, which may theoretically include effects of relative humidity as well as effects of sea surface temperature (Jouzel et al., 1982). In this manuscript, we explore this relationship systematically in all available records (this was not done previously) and compare the ECHAM5-wiso outputs with these datasets to test if the model is able to produce realistic seasonal lags (to our knowledge, it was never be done previously either). We better introduced section 4.2 to show the importance of studying the deuterium excess - $\delta$18O phase lag p.24 l.9: "The phase lag between d-excess and $\delta$18O was initially explored to identify changes in evaporation conditions (Ciais et al., 1995). D-excess has been interpreted as a proxy for relative humidity at the moisture source (Pfahl and

Sodemann, 2014). However, recent studies of Antarctic precipitation data combined with back-trajectory analyses did not support this interpretation (e.g. Dittmann et al., 2016;Schlosser et al., 2017), calling for further work to understand the drivers of seasonal d-excess variations."

"** The figures could use improvement. Especially, there should be a variable name and units on the color bars of the various maps. For example, Figure 10 should read "lag, in months", on the color axis." For more clarity, we added the labels of Figures 4, 9 and 10.

"**The calculation of statistical significance, throughout the paper, is not clear. Is autocorrelation accounted for in stating that p<0.05?" We confirm that we consider a linear relationship as significant when the corresponding p-value is lower than 0.05. The criterion of a significant linear relationship are now explicitly indicated at the end of Section 2.3 (method section) p.11 l.22: "Our comparisons are mainly based on linear regressions. Note that through all the manuscript, we consider a linear relationship to be significant for p-value<0.05."

"**Several recent papers have demonstrated that the logarithmic form of deuterium excess is a much more reliable and robust measure than the traditional linear calculation. The paper really ought to look at this as well. See Markle and others (doi: 10.1038/ngeo2848) and Uemera et al., 2012 (doi: 10.5194/cp-8-1109-2012), Dutsch et al. 2017, 10.1002/2017JD027085"

We thank the referee for suggesting to test alternative definition of deuterium excess. We understand that the logarithmic definitions were introduced for large amplitude, glacial-interglacial changes (Uemura et al., 2012) or glacial abrupt events (Markle et al., 2017). However, Uemura et al. (2012) call for caution when interpreting this alternative parameter which does not highlight key second order features (e.g. deglacial lags; obliquity signals) as strongly as d-excess. Moreover, different logarithmic definitions have been introduced by different authors : a second-order polynomial (Uemura et al.,

2012;Markle et al., 2017) or a first-order polynomial approach (Dütsch et al., 2017), preventing comparisons between these studies. Such a logarithm definition removes the $\delta$-scale effect (thus correcting for temperature-dependent d-excess signals, particularly strong at glacial-interglacial scales) but may be sensitive to the air-mixing effects (Risi et al., 2013). We also note that most studies exploring the potential of d-excess to identify changes in moisture origins also used the classical definition of deuterium excess (e.g. Schlosser et al., 2008;Dittmann et al., 2016). Finally, recent studies found similar signals using both definitions (Schoenemann and Steig, 2016). We nevertheless checked the implications of the logarithmic definition with our data (see Figure S1 from the Supplementary Material associated with this response). The first test is performed with the spatial distribution of water isotopes. Using the spatial distribution of data in Antarctica, we obtain a slightly better correlation coefficient using a $\delta$D-$\delta$18O linear relationship compared to using a ln(RD)- ln(R18O) linear relationship (correlation coefficients of 0.9978 and 0.9963 respectively, and slopes of respectively 7.76 ‰ ‰1 and 10.49 ,see S1.a and S1.b from the Supplementary Material). We then performed the same test using daily precipitation data from two completely different sites, the plateauing site of Dome C (Stenni et al., 2016) (See S2 from the Supplementary Material) and the coastal site of Neumayer (Schlosser et al., 2008) (See S3 from the Supplementary Material). For each site, we processed $\delta$D-$\delta$18O and ln(RD)- ln(R18O) linear relationships over all the data, and then over the mean seasonal cycle. As over all time-averaged data points from our database, we observe that the correlation coefficient does not vary much from one definition to another one. Also, the slope is different depending on the definition when focusing on a same site (e.g. 10.977 and 6.720 ‰ ‰1 for the logarithmic and the classical definition respectively using the precipitation data measured at Dome C). However, whatever the considered definition, it differs from one site to another one using the same time scale (e.g. 10.988 and 7.930 for Dome C and Neumayer respectively using the logarithmic definition over the daily data), as well as for different time scales at a same site (e.g. 10.988 and 11.385 for daily data and mean seasonal data respectively at Dome C). As a result, no criterion for this test

allows us to choose one definition rather than another one. Finally, we compared the deuterium excess mean seasonal cycle using both definitions for the aforementioned sites (see S4 from the Supplementary Material). Although the variations are similar for precipitation data from Neumayer, they do not look like the same for precipitation data from Dome C. But the cause responsible for this difference is not understood. We conclude that none of the definition is stable in space and time using our data, opening the issue of the adequate slope to use for an alternative d-excess logarithm definition to be applied to our database and for the model-data comparison. Nevertheless, we prefer to use the classical definition for a better comparison with previous studies.

Bromwich, D. H., and Weaver, C. J.: Latitudinal displacement from main moisture source controls $\delta$18O of snow in coastal Antarctica, Nature, 301, 145, 10.1038/301145a0, 1983. Bromwich, D. H., Fogt, R. L., Hodges, K. I., and Walsh, J. E.: A tropospheric assessment of the ERA-40, NCEP, and JRA-25 global reanalyses in the polar regions, Journal of Geophysical Research: Atmospheres, 112, n/a-n/a, 10.1029/2006JD007859, 2007. Bromwich, D. H., Nicolas, J. P., Monaghan, A. J., Lazzara, M. A., Keller, L. M., Weidner, G. A., and Wilson, A. B.: Central West Antarctica among the most rapidly warming regions on Earth, Nature Geoscience, 6, 139, 2013. Ciais, P., White, J., Jouzel, J., and Petit, J.: The origin of present‐day Antarctic precipitation from surface snow deuterium excess data, Journal of Geophysical Research: Atmospheres, 100, 18917-18927, 1995. Cuffey, K. M., and Steig, E. J.: Isotopic diffusion in polar firn: implications for interpretation of seasonal climate parameters in ice-core records, with emphasis on central Greenland, Journal of Glaciology, 44, 273-284, 1998. Delmotte, M., Masson, V., Jouzel, J., and Morgan, V. I.: A seasonal deuterium excess signal at Law Dome, coastal eastern Antarctica: a southern ocean signature, Journal of Geophysical Research: Atmospheres, 105, 7187-7197, 2000. Dittmann, A., Schlosser, E., Masson-Delmotte, V., Powers, J. G., Manning, K. W., Werner, M., and Fujita, K.: Precipitation regime and stable isotopes at Dome Fuji, East Antarctica, Atmospheric Chemistry and Physics, 16, 6883-6900, 2016. Dütsch, M., Pfahl, S., and Sodemann, H.: The impact of nonequilibrium and

equilibrium fractionation on two different deuterium excess definitions, Journal of Geophysical Research: Atmospheres, 122, 2017. Fernandoy, F., Meyer, H., and Tonelli, M.: Stable water isotopes of precipitation and firn cores from the northern Antarctic Peninsula region as a proxy for climate reconstruction, The Cryosphere, 6, 313, 2012. Fernandoy, F., Tetzner, D., Meyer, H., Gacitúa, G., Hoffmann, K., Falk, U., Lambert, F., and MacDonell, S.: New insights into the use of stable water isotopes at the northern Antarctic Peninsula as a tool for regional climate studies, The Cryosphere, 12, 1069-1090, 10.5194/tc-12-1069-2018, 2018. Genthon, C., Six, D., Favier, V., Lazzara, M., and Keller, L.: Atmospheric temperature measurement biases on the Antarctic plateau, Journal of Atmospheric and Oceanic Technology, 28, 1598-1605, 2011. Grazioli, J., Madeleine, J.-B., Gallée, H., Forbes, R. M., Genthon, C., Krinner, G., and Berne, A.: Katabatic winds diminish precipitation contribution to the Antarctic ice mass balance, Proceedings of the National Academy of Sciences, 114, 10858-10863, 2017. Holloway, M. D., Sime, L. C., Singarayer, J. S., Tindall, J. C., Bunch, P., and Valdes, P. J.: Antarctic last interglacial isotope peak in response to sea ice retreat not ice-sheet collapse, Nature communications, 7, 2016. Hoshina, Y., Fujita, K., Nakazawa, F., Iizuka, Y., Miyake, T., Hirabayashi, M., Kuramoto, T., Fujita, S., and Motoyama, H.: Effect of accumulation rate on water stable isotopes of near‐surface snow in inland Antarctica, Journal of Geophysical Research: Atmospheres, 119, 274-283, 2014. Jasechko, S., Lechler, A., Pausata, F. S. R., Fawcett, P. J., Gleeson, T., Cendón, D. I., Galewsky, J., LeGrande, A. N., Risi, C., Sharp, Z. D., Welker, J. M., Werner, M., and Yoshimura, K.: Late-glacial to late-Holocene shifts in global precipitation $\delta$18O, Clim. Past, 11, 1375-1393, 10.5194/cp-11-1375-2015, 2015. Jones, T., Cuffey, K., White, J., Steig, E., Buizert, C., Markle, B., McConnell, J., and Sigl, M.: Water isotope diffusion in the WAIS Divide ice core during the Holocene and last glacial, Journal of Geophysical Research: Earth Surface, 122, 290-309, 2017. Jouzel, J., Merlivat, L., and Lorius, C.: Deuterium excess in an East Antarctic ice core suggests higher relative humidity at the oceanic surface during the last glacial maximum, Nature, 299, 688, 1982.

[revised manuscript text omitted]

Please also note the supplement to this comment:
https://www.clim-past-discuss.net/cp-2017-118/cp-2017-118-AC1-supplement.pdf

———————————————

[Figure]

**Supplement:**

S1: Linear relationships between (a) $\delta D$ (in ‰) and $\delta^{18}O$ (in ‰) using all (time-averaged) data points from our database, (b) ln(RD) and ln($^{18}O$) using all (time-averaged) data points from our database (excluding 9 outliers, thus making a total of 323 points).

[Figure]

(a)

(b)

S2: Linear relationships between δD (in ‰) and δ¹⁸O (in ‰) (a) using all data points (501 points), (b) data points of the mean seasonal cycle (12 points) of precipitation data measured at Dome C (Stenni et al., 2016), and used in our database, and linear relationships between ln(RD) and ln($^{18}$O) (c) using all data points, (d) data points of the mean seasonal cycle from the same measurements monitored.

[Figure]

S3: Linear relationships between δD (in ‰) and δ¹⁸O (in ‰) (a) using all data points (342 points), (b) data points of the mean seasonal cycle (12 points) of precipitation data measured at Neumayer (Schlosser et al., 2008), and used in our database, and linear relationships between ln(RD) and ln(¹⁸O) (c) using all data points, (d) data points of the mean seasonal cycle from the same measurements monitored.

[Figure]

S4: Mean seasonal cycle using the classical deuterium excel definition (orange solid line), and the logarithmic deuterium defined by Uemura et al. (2012) as $dln = \ln(1 + \delta D) - ((-2.85 \times 10^{-2}) \times \ln(1 + \delta^{18}O))^2 + 8.47 \times \ln(1 + \delta^{18}O))$ (orange solid line) (a) using the precipitation data measured at Dome C (Stenni et al., 2016) and (b) precipitation data measured at Neumayer (Schlosser et al., 2008).

[Figure]

(a)

[Figure]

(b)

Schlosser, E., Oerter, H., Masson-Delmotte, V., and Reijmer, C.: Atmospheric influence on the deuterium excess signal in polar firn: implications for ice-core interpretation, Journal of glaciology, 54, 117-124, 2008.

Stenni, B., Scarchilli, C., Masson-Delmotte, V., Schlosser, E., Ciardini, V., Dreossi, G., Grigioni, P., Bonazza, M., Cagnati, A., and Karlicek, D.: Three-year monitoring of stable isotopes of precipitation at Concordia Station, East Antarctica, The Cryosphere, 10, 2415, 2016.

Uemura, R., Masson-Delmotte, V., Jouzel, J., Landais, A., Motoyama, H., and Stenni, B.: Ranges of moisture-source temperature estimated from Antarctic ice cores stable isotope records over glacial-interglacial cycles, Climate of the Past, 8, 1109-1125, 10.5194/cp-8-1109-2012, 2012.

---

## Author Comment (AC3) · 13 Apr 2018

"General Comments This paper addresses novel and relevant scientific questions within the scope of CP. The authors use an impressive collection of data to assess the skill of the ECHAM5-wiso model. The main scope of the paper is to evaluate spatial, seasonal and interannual $\delta$18O-temperature relationships, as well as deuterium excess and $\delta$18O phasing. This information is important for correctly interpreting certain climate records in Antarctica, especially when using shallow ice core records of a few decades length." "Minor revisions are required for publication, as well as one major

revision and/or clarification." We thank the first referee for reviewing our manuscript. In the following lines, we answer to the reviewer comment by comment.

"Specific Comments * My biggest concern is how the authors addressed water isotope diffusion in shallow ice core records. The majority of diffusion occurs in the upper âĹij10- 20 meters of the ice sheet, thus this will have a significant effect on the results of this study (i.e. for ice core data from 1979-2013, or for any data extending beyond a few years in length). Can the authors clarify whether any consideration was given to the attenuation of the seasonal and multi-year variations due to diffusion? For example, at a typical inland West Antarctic site (mean annual temp = -30.3C, accumulation = 0.23 m/yr), the annual d18O and dD signal amplitudes will decrease by about âĹij50% in 30 years (calculated using a Johnsen firn model and a Herron-Langway densification model). For a colder site (temp = -40.3C, accum = 0.12 m/yr), the amplitudes decrease by 67% in 30 yrs. And for a warmer site with high accum (temp = -25.3C, accum = 0.38 m/yr), the amplitudes are decreased by 37% in 30 years. These are quick calculations, but show the importance of diffusion. Could firn diffusion be the cause of model-data mismatch? If so, and I think this is the case, the authors should either make these calculations and include the corrections in the paper, or state a few examples of signal attenuation for different temperatures and accumulation that are relevant to the ice core sites used in the paper. On the other hand, if I have misunderstood the results, please provide clarifications and explain why." This is an important issue for the quantitative comparison of seasonal isotopic amplitudes in precipitation model outputs with firn data, potentially affected by diffusion (Johnsen et al, 2000). The theory of Whillans and Grootes (1985) about isotopic diffusion in firn based on diffusional vapor flux through firn pore spaces, appears compatible with the estimated loss of seasonal amplitude through depth in diverse sites (e.g. Cuffey and Steig, 1998), but this validation is limited by the lack of comprehensive datasets (monitoring of precipitation isotopic composition over multiple years to be compared with the firn records), as well as uncertainties on key parameters. Accounting for firn ventilation effects on sublimation and condensation and disequilibrium between pore-space vapor and snow

grains Neumann and Waddington (2004) shows more rapid isotopic changes in the upper few firn meters at low-accumulated site than explained by Whillans and Grootes (1985). Recent studies in Greenland have further evidenced changes in surface snow isotopic signal in between precipitation events, attributed to water vapour exchange associated with snow metamorphism (Steen-Larsen et al., 2014). While there has long been evidence for a loss of seasonal amplitude with depth, a reliable quantification of the effect of diffusion in firn for all available Antarctic records is currently out of reach, and thus cannot be applied to remove this bias in model-data comparisons of seasonal amplitudes (Touzeau et al., 2017).

Here, we just focus on all sub-annual records available from our database. For each record, we calculate the ratio of the seasonal amplitude estimated from the first three first seasonal cycles to the mean seasonal amplitude for all available seasonal cycles along this core. If the seasonal cycle of precipitation isotopic composition was constant through time, the interplay of diffusion and the averaging of seasonal amplitude over multiple seasonal cycles would make this ratio as an indication of the loss of seasonal amplitude, assuming that the amplitude of the first three seasonal cycles is representative of that of precipitation (Table 1 in Supplementary Material). We obtain a mean ratio of $1.40 \pm 0.47$. No significant relationship can be identified between this ratio and the corresponding estimated annual accumulation rates (see enclosed Figure 1). We note that a ratio lower than 1 is obtained in five ice cores, including one with a mean annual accumulation of 15 cm w.e. y-1; this situation can be interpreted as resulting from inter-annual variations in precipitation isotopic composition and/or diffusion characteristics in the upper firn. Our simple empirical calculation shows that we cannot exclude a loss of seasonal amplitude in the firn data used to estimate the average seasonal isotopic amplitude, due to post-deposition processes; it also shows that the average seasonal amplitude obtained in our firn multi-year records may be affected by an average loss of about 70% (1/1.4) of the seasonal amplitude recorded during the first three years of each firn core. We cannot assess directly the potential distortion of the seasonal amplitude from the initial precipitation to the snow surface due to the lack of systematic

precipitation, surface snow and firn multi-year monitoring datasets. These elements are reported in our revised manuscript, in Section 3.2.2 "$\delta$18O seasonal amplitude", p.16 l.1 as follows: "In order to quantify post-deposition effects in ice cores, we calculated the ratio of the three first seasonal amplitudes by the mean seasonal amplitude in sub-annual ice cores (See Supplementary Information S2). We find a mean ratio of 1.40 $\pm$ 0.47. We explored whether this ratio was related to annual accumulation rates (See Supplementary Information S3), without any straightforward conclusion. We also observe that five ice cores depict a ratio lower than 1, including one with a mean yearly accumulation of 15 cm w.e. y-1, a feature which may arise from inter-annual variability in the precipitation seasonal amplitude or in post-deposition processes. This empirical analysis shows that a loss of seasonal amplitude due to post-deposition processes is likely in most cases, with an average loss of the seasonal amplitude of approximately 70% compared to the amplitude recorded in the upper part of the firn cores (first three years). " Here, S2 and S3 correspond to the attached Figure 1 and Table 1 attached to this response. We specified that the overestimation of the mean $\delta$18O seasonal amplitude by ECHAM5-wiso compared to ice core data could be due to post-deposition effects: p.16 l.12 "The overestimation when comparing with ice core data could be due to the attenuation of signal by post-deposition effects (as aforementioned) rather than a model bias." p.20 l.23: "Again, we cannot rule out a loss of amplitude in ice core data compared to the initial precipitation signal, due to the temporal resolution and to post-deposition effects.the overestimation when comparing against ice core data, i.e. an attenuation in the data by post-deposition effects."

"* Please explain "nudging", and perhaps use different wording in the paper. While this may be common terminology, it is not immediately clear what it means, nor does it appear to be defined in the main text of the paper."

"Nudging" is a common term used in atmospheric modelling studies, referring to a specific methodology related to data assimilation (e.g. Risi et al., 2013). Details of the nudging used for ECHAM5-wiso are given in Butzin et al. (2014): "the dynamic–

thermodynamic state of the ECHAM model is constrained to reanalysis data by an implicit nudging technique (Krishamurti et al., 1991; the implementation in ECHAM is described by Rast et al., 2013) – i.e. modelled fields of surface pressure, temperature, divergence and vorticity are relaxed to the corresponding ERA-40 and ERA-Interim reanalysis fields (Uppala et al., 2005; Berrisford et al., 2011; Dee et al., 2011). The nudging interval is 6 hours, ensuring that the simulated large-scale atmospheric flow is modelled in agreement with the ECMWF reanalysis data on all analysed timescales." As the manuscript is dedicated to non-modelers, we understand that this word is not understood prima facie from all, and thus referred to Butzin et al. (2014) in the introduction, p.6 l.12: "We explore a simulation performed for the period 1960-2013, where the atmospheric model is nudged to the European Reanalyses (ERA) ERA-40 and ERA-interim reanalyses (Uppala et al., 2005), ensuring that the day-to-day simulated variations are coherent with the observed day-to-day variations in synoptic weather and atmospheric circulation (see Butzin et al., 2014 for more explanation)."

"I would also suggest a short, 1- sentence explanation in the introduction that explains the relevance of slopes for ice core isotope-temperature relationships, etc." As suggested, we added one sentence in the introduction p.4 l.6, to show the relevance in the isotope-temperature slope to infer past temperatures: "Water stable isotope measured along ice cores were initially used to infer Antarctic past temperatures using the isotope-temperature slope (e.g. Lorius et al., 1969)."

"* Can you please confirm that for any averaged isotope data, that the same averaging was done in the model. If not, please state why, and how this could affect results. Also, please provide a clarification on how averaging could reduce the amplitude of the observed seasonal and multi-year signals." For comparison with precipitation data, as detailed in Section 2.3 "Methods for model-data comparison", daily precipitation outputs were extracted in ECHAM5-wiso to correspond to the same days than in the data. Thus, same averages (seasonal, annual, and time-averaged) were then processed in the data and in the model. The comparison between ECHAM5-wiso outputs and ice

core data is different, as monthly and annual model outputs are then calculated from precipitation weighted daily data to mimic ice core signals (without accounting for post-deposition processes, as described above). We have rewritten Section 2.3 "Methods for model-data comparison" clarify the fact that averaging was done exactly in the same way for model and precipitation data: "We have extracted daily 2-m temperature outputs (hereafter 2m-T) for comparison with surface air instrumental records, daily (precipitation minus evaporation) outputs (hereafter P-E) for comparison with SMB data, and daily precipitation isotopic composition outputs for comparison with measurements of isotopic composition data from precipitation samples. For the comparison of annual model outputs with ice core data, we averaged daily precipitation isotopic composition weighted by the daily amount of precipitation. For each specific site, we selected the model grid cell including the coordinates of the site. When comparing model outputs with the database of surface data (time-averaged SMB and isotopic composition), available data have been averaged within each model grid cell. Time selection was dependent on the variables. The 2-m T outputs have been compared with temperature records for the period 1960-2013, based on annual averages and selecting same years than in the data (see Section 3.1.1). The comparison with other datasets (SMB, snow and water stable isotopes from firn/ice cores) is restricted to the period 1979-2013, due to concerns about the skills of the reanalyses used for the nudging prior to 1979 in Antarctica (see next section). Daily (P-E) outputs were all extracted over the whole period 1979-2013 and averaged (see Section 3.1.2). For comparison with the surface isotopic database (Section 3.2.1), daily precipitation isotopic composition were averaged by weighting by the daily amount of precipitation over the whole period 1979-2013. For the inter-annual variability (same Section) or annual values (e.g. for d-excess outputs, see Section 4), daily precipitation isotopic composition were averaged by weighting by the daily amount of precipitation for each year of the period 1979-2013. For sub-annual isotopic composition, we used precipitation isotopic compositions (amplitude and mean seasonal cycle) and highly resolved ice cores (amplitude only). Precipitation isotopic composition data consist of a very small number of measurements,

sometimes taken before 1979 (e.g. observations from DDU consist in 19 measurements during 1973), and thus model precipitation isotopic composition outputs were extracted at the very exact sampling date. Then, monthly averages were performed and mean seasonal cycles were calculated. The resulting mean seasonal cycles of precipitation isotopic composition were obtained the same way in both the precipitation data and the model outputs. For comparison with the mean season amplitude of the highly resolved ice cores, the mean seasonal amplitude was calculated from the mean seasonal cycle based on the monthly averages (weighted by the precipitation amount) over the period covered by the ice core record. Finally, for the spatial linear relationships, calculations reported for each grid cell are based on the relationship calculated by including the 24 grid cells ($\pm$ 2 latitude steps. $\pm$ 2 longitude points) surrounding the considered grid cell." Also, we detailed the time resolution of the outputs in the captions of Tables: - In Table 4: "$\delta$18O mean seasonal amplitude (in ‰ calculated for precipitation and sub-annual ice core data, as well as simulated by ECHAM5-wiso for the same time period than the data. The time resolution used in the model corresponds to the time resolution of the precipitation data, and to the annual scale for the ice core data (i.e. yearly averages based on daily precipitation isotopic composition weighted by the amount of daily precipitation). The data type is identified as 1 for precipitation samples and 2 for ice core data." - In Table 6: "Slope (in ‰ ‰1), correlation coefficient (noted as "r") and p-value of the ïĄ́18O-ïĄ́D linear relationship from precipitation measurements (top of the table) and ice core data (bottom of the table) over the available period and at daily or monthly scale depending of the time resolution of the data, and from the ECHAM5-wiso model over the observed period at the time resolution of the data for the precipitation and at the annual scale for the ice core data (i.e. yearly averages based on daily precipitation isotopic composition weighted by the amount of daily precipitation). Numbers into brackets correspond to the standard errors." - In Table 7: "Mean value (noted as "$\mu$", in ‰) and standard deviation (noted as "$\sigma$", in ‰ of sub-annual d-excess in observational time series at daily or monthly scale (when the name of the station is associated with an asterisk) for the precipitation at the lowest time resolution

for the ice core data, and simulated d-excess by ECHAM5-wiso for the same time period as the observations for precipitation and at the annual scale for the ice core data (i.e. yearly averages based on daily precipitation isotopic composition weighted by the amount of daily precipitation). Mean values which are overestimated by ECHAM5-wiso are written in italic." - In Table 8: "Table 8: d-excess mean seasonal amplitude (in ‰ calculated for precipitation at daily or monthly scale (when the name of the station is associated with an asterisk) for the precipitation at the lowest time resolution for the ice core data and sub-annual ice core data at the lowest time resolution, as well as simulated by ECHAM5-wiso for the same time period as each record. The data type is identified as 1 for precipitation samples and 2 for ice core records. Amplitude values that are overestimated by ECHAM5-wiso are written in italic."

"* The authors state once that ". . . a stationary isotope-temperature slope cannot be applied for the climatic interpretation of Antarctic ice core." (pg 3, line 1-2). This is an important point. I think this point should be made in the Conclusion as well, specifically that the results of this study (or at least some of the results) may not hold in the deeper past (greater than a few decades). Please be clear in your assessment of the relevance for paleoclimate interpretations. This has the potential to be misunderstood."

We thank the reviewer for highlighting this conclusion, which was taken into account to propose temperature reconstructions spanning the last 2000 years (Stenni et al, 2017). We have added this finding in the conclusions, stressing that it only applies to inter-annual to decadal changes: p.26 l.21: "Expanding earlier site-specific studies, we show that the strength and slope of the $\delta$18O-temperature linear relationship is not stationary in Antarctica over the last four decades. This finding has implications for past temperature reconstructions using ice core records."

"* In many instances, the citations are dated. There are many more recent studies that should be cited in this manuscript. I encourage the authors to provide citations of more recent studies." We thank Reviewer 1 for this suggestion. We have used the most recent references for our observation database. The peer-review literature for isotopic

modeling, post-deposition processes and model-data comparisons has been screened to identify recent works which we had not cited. The following references have been added in the text We have thus cited: - In the introduction: o p.4 l.6 (Schoenemann et al., 2014) o p.4 l.10 (Jones et al., 2017;Münch et al., 2017;Sokratov and Golubev, 2009;Laepple et al., 2018) o p.4 l.14 (Smith and Stearns, 1993;Turner, 2004;Stammerjohn et al., 2008;Schroeter et al., 2017) o p.4 l.21 (Hoshina et al., 2014) o p.4 l.25 (Grazioli et al., 2017) o p.5 l.14 (Fernandoy et al., 2018) - Section 3.2.1 o p.16 l.8 (Waddington et al., 2002;Neumann and Waddington, 2004) - Section 4.1 o p.24 l.1 (Steiger et al., 2017;Sturm et al., 2010) - Section 4.2 o p.24 l.8 (Pfahl and Sodemann, 2014;Jouzel et al., 2013;Kurita et al., 2016) o p.24 l. 9 (Schlosser et al., 2017)

Technical Corrections "pg 2 line 7 - nudged? please explain what this means somewhere in the introduction, and possibly change the wording." The text has been modified to refer to Butzin et al. (2014), p.6 l.12.

"pg 2 line 15-17 - the description is unclear" The description has been clarified: "The comparison with accumulation and water stable isotope data is thus restricted to the period 1979-2013, for accumulation and water stable isotope data from snow and firn/ice core but not for the isotopic composition from precipitation data that would consist in a too few number of points."

"pg 2 line 28 - slopes? "We show that local spatial or seasonal slopes" the relevance of slopes should be defined in the introduction so certain readers are not left wondering what this means" We did not detail in the abstract the relevance of the ïA̧d'18O-temperature relationship and particularly its slope, due to space limitations. Nevertheless, we explained it in the introduction from p.4 l.22 to l.27: "Pioneer studies evidenced a close linear relationship between the spatial distribution of water stable isotopes and local temperature (e.g. Lorius and Merlivat, 1975), and explained this feature as the result of the distillation along air mass trajectories. Thereupon, local temperature (i.e. at a specific site) was reconstructed using $\delta$18O measurements and based on the slope of the aforementioned empirical relationship (...)Evaporation conditions, transport and boundary layer processes may vary through time, from seasonal (Fernandoy et al., 2018) to annual or multi-annual scale, thereby potentially distorting the quantitative relationship between snow isotopic composition and local surface air temperature estimated empirically for present day conditions (Jouzel et al., 1997)."

"pg 3 line 6 - "This work valuates" - evaluates?" We replaced it.

"pg 4 line 4 - consider saying "the hydrologic cycle" rather than "water cycle"" p.4 l.4: we replaced "water cycle" by "hydrological cycle".

"pg 4 line 6-8: "Their climate interpretation is however limited, first by the alteration of the signal due to deposition and post-deposition processes, and second by the complexity of all parameters affecting the Antarctic snowfall isotopic composition" cite sources. For Antarctica, one of the more in-depth studies of "post-depositional processes" is Jones et al., 2017 "Water isotope diffusion in the WAIS Divide ice core during the Holocene and last glacial" doi:10.1002/2016JF003938. Also provide citations for depositional processes and "complexity of all parameters" - perhaps you mean isotopic recharge, etc?"

For post-deposition effects, we referred to Sokratov and Golubev (2009), Jones et al. (2017), Laepple et al. (2018) and Münch et al. (2017). By "complexity of all parameters", we meant the interplay of all parameters driving the isotopic composition such as the origin of moisture or the intermittency of precipitation, as stressed by Krinner and Werner, (2003); Sime et al., (2009); Hoshina et al., (2014) and Touzeau et al. (2016). "The climate signal potentially recorded in precipitation isotopic composition is however difficult to disentangle. First the original signal from precipitation may be altered due to deposition and post-deposition processes (e.g. Jones et al., 2017;Münch et al., 2017;Sokratov and Golubev, 2009;Laepple et al., 2018), which cannot yet be quantified. Second, the Antarctic snowfall isotopic composition may be affected by the origin of moisture and the associated evaporation conditions, or by changes in the relationships between condensation and surface temperature, as well as by changes in the intermittency of precipitations (e.g. Sime et al., 2009;Krinner and Werner, 2003;Hoshina et al., 2014;Touzeau et al., 2016)." pg 4 line 6: "Their"? Who are they? "Their" stand for "Water stable isotope measured along ice cores" from the sentence before. We thus made more explicitly the link: "The climate interpretation of such records".

"pg 4 line 6-8: "Their climate interpretation is however limited, first by the alteration of the signal due to deposition and post-deposition processes, and second by the complexity of all parameters affecting the Antarctic snowfall isotopic composition." - please use another word other than limited. I think you mean to say that post depositional processes alter the original signal, which must be accounted for in climate interpretations?" We have reformulated the sentence for clarity (see above).

"pg 4 line 13-16, "However, recent studies cast doubt on this assumption, evidencing isotopic exchanges between the Antarctic snow surface and the atmosphere associated with snow metamorphism occurring at the diurnal and sub-annual scales (Ritter et al., 2016;Casado et al., 2016;Touzeau et al., 2016)." consider citing Steen- ĚĞ Larsen et al.?" We thank the referee who pointed the work of Steen-Larsen et al., very complementary to our citations: "However, recent studies cast doubt on this assumption, evidencing isotopic exchanges between the Antarctic snow surface and the atmosphere associated with snow metamorphism occurring at the diurnal and sub-annual scales (Ritter et al., 2016;Casado et al., 2016;Touzeau et al., 2016;Jones et al., 2017;Steen-Larsen et al., 2014)."

"pg 4 line 16-18: Again, the most recent diffusion study I have seen is Jones et al. 2017, it provides important information with an Antarctic perspective, and it should be cited here. There are important points in Jones et al. 2017 that improve on Sigfus Johnsen's 2000 paper." We thank the referee for his suggestion to refer to the recent work of Jones et al., 2017. This contribution is indeed very important to improve the knowledge of post-deposition processes, especially the ice-deformational thinning along the ice core, and the crystal-type acting in diffusion. We have thus cited it: "Other processes such as melt and diffusion processes can also alter the preservation of isotopic signals in

firn and ice and cause smoothing of the initial snowfall signals (Johnsen, 1977;Whillans and Grootes, 1985;Johnsen et al., 2000;Jones et al., 2017)."

"pg 4 line 18-19: "So far, the overall importance of such post-deposition processes on the alteration of the initial precipitation signals cannot be quantified." This is not true. The alteration of the initial precip signal can be determined reasonably well by fitting a Gaussian to the data. Similarly, the Johnsen firn diffusion model, to the first order, is also a reasonable model for signal alteration. However, there are physical mechanisms that are still not understood." We agree that post-deposition processes can be described using an empirical spectral analysis, but cannot yet by fully understood based on a mechanistic model (see Touzeau et al., 2017). Thus, we rewrote our sentence as follows: "So far, the mechanisms of such post-deposition processes on the alteration of the initial precipitation signals are not fully understood and quantified."

"pg 5 line 7: "$\delta$18O and deuterium", should be "$\delta$18O and $\delta$D (D refers to deuterium)" - something like this would be more consistent" We replaced "deuterium" by "$\delta$D".

"pg 5 line 19: is this really the only exception??? "with one exception (Lee et al., 2008).""" Among the studies focusing on the stationary of the isotope-temperature relationship using simulations (Jouzel et al., 1997), Lee et al. (2008) is the only study at our knowledge, that shows, that the spatial isotope-temperature relationship is not a good approximation for glacial conditions.

"pg 6 line 2: what is motivating "interannual scale" research, I suggest mentioning why this matters in the introduction" As suggested, we deepened our motivation p.4 l.8 : "The focus on inter-annual variations is motivated by the goal to quantify temperature changes at the Earth's surface, including Antarctica, during the last millennia, to place current changes in the perspective of recent natural climate variability (Jones et al., 2016), to understand the drivers of this variability, and to test the ability of climate models to correctly represent it (Stenni et al., 2017). This timescale is relevant for the response of the Antarctic climate to e.g. volcanic forcing, and for the Antarctic climate

fingerprint of large-scale modes of variability such as ENSO and the Southern Annular Mode (Smith and Stearns, 1993;Turner, 2004;Stammerjohn et al., 2008;Schroeter et al., 2017). "

"pg 7 line 11: "cautious" - caution?" Thanks for having highlighted this mistake. We replaced "cautious" by "caution".

"pg 7 line 23: nudged to, what does this mean?" Please consider our response above.

"pg 8 line 10: just use dD rather than deuterium to avoid confusion, and make sure to define D, see above comment" As previously, we replaced "deuterium" by "$\delta$D".

"pg 8 line 18-19: unclear what this means, "the averaging period may be heterogeneous, including subintervals within 1960-2013, or longer time periods."" We changed the sentence for a better understanding: "(...) in this case, the averaging period is based on different periods, with potential not continuous records."

"pg 11 line 3-5: "While this bias is small (less than 2âŬęC)" - this is not small, please re-word" The 2°C bias for Dronning Maud Land is smaller compared to the 15°C bias for McMurdo. We have rewritten the sentence to: "While this bias is less than 2°C for Droning Maud Land (Mawson and Neumayer) and over the Peninsula (Palmer and Esperanza), it reaches 7°C for the Coastal Indian region (Casey and Dumont d'Urville) and is very strong over the Victoria Land region (McMurdo), reaching 15°C."

"pg 11 line 8: "above the ice sheet" - what does this mean?" We meant "inland", so we substitute "above the ice sheet" by this word. "In contrast, ECHAM5-wiso has a warm bias for all the stations located inland (Vostok, Dome C and Byrd)."

"pg 12 line 1-2: "despite the nudging technique (not shown)." - what exactly is not shown? As mentioned previously, please explain nudging." What is not shown is the mean values and the amplitude of inter-annual variations simulated by ERA-interim. We have rewritten this sentence to: "We note that mean values and the amplitude of inter-annual variations are different for ECHAM5-wiso and ERA (not shown), as expected from different model physics, despite the nudging technique." For the "nudging" term, we refer the referee to the above response.

"pg 14 line 21-24: "The largest deviations are encountered in coastal regions, where either the model resolution is too low to resolve advection and boundary layer processes (e.g. katabatic winds), or where post-deposition processes may have a larger influence." Why would post deposition processes have a larger influence? Larger compared to what?" Larger deviations are observed in coastal regions, affected by katabatic winds. Such processes are not resolved in the model, which thus do not account for the associated deposition effects (e.g. snow drift by the winds). Strong winds also enhance ventilation and thus the equilibration between surface snow and water vapor in the boundary layer, one of the component of post-deposition effects (See Waddington et al., 2002). We have thus rewritten the sentence to: "The largest deviations are encountered in coastal regions, where the model resolution is too low to resolve advection and boundary layer processes (e.g. small scale storms, katabatic winds). Katabatic winds also have the potential to enhance ventilation-driven post-deposition processes (Waddington et al., 2002;Neumann and Waddington, 2004)."

"pg 15 line 23: "We have calculated the mean amplitude of the $\delta$18O sub-annual variations" - please clarify what amplitude you are calculating? Monthly?" Ice core data available at sub-annual resolution are dated by annual layer counting, at best at the seasonal scale (through the identification of summer peaks). For each year, an annual amplitude can be estimated through the difference between the corresponding minimum and maximum values. The mean amplitude of $\delta$18O sub-annual variations correspond to the mean $\delta$18O annual amplitude. We rephrased the sentence p.15 l.23: "We have calculated the mean of the $\delta$18O annual amplitude (i.e. maximum – minimum values within each year) in ice core records (…)."

"pg 16 line 2-4: "ECHAM5-wiso underestimates the seasonal amplitude (by 14 to 69%) when compared to precipitation data, but overestimates the seasonal amplitude when compared to ice core data (from 11 to 71%)." âAËŸT could the seasonal amplitude

over- estimation in the model be related to diffusion? These overestimations are similar to the annual signal attenuation examples I gave above." Please refer to the response to your first comment concerning diffusion/post-deposition effects, which is related to post-deposition effects damping the seasonal amplitude.

"pg 18 line 10-11: this needs more explaining and/or a citation - "Due to the temperature dependency of equilibrium fractionation coefficients, dexcess increases when temperature decreases."" As temperature decreases, the difference between equilibrium fractionation coefficients increases, leading to a gradual deviation from the meteoric water line (calculated at the global scale, where the coefficient of 8 results from the average equilibrium fractionation coefficients), and thus to a gradual increase in d-excess. We referred to Masson et al., 2008 and Touzeau et al., 2016, which deal with the temperature dependency of the deuterium excess: "Due to the temperature dependency of equilibrium fractionation coefficients leading to a gradual deviation from the meteoric water line (calculated at the global scale, where the coefficient of 8 results from the average equilibrium fractionation coefficients), d-excess increases when temperature decreases (Masson-Delmotte et al., 2008;Touzeau et al., 2016)."

"pg 19 lines 16-18: "ECHAM5-wiso systematically underestimates the d-excess mean seasonal amplitude when compared with precipitation data, while it systematically overestimates it when compared with ice core data." could the overestimation be due to diffusion, which would decrease the dxs amplitude? what is the range of overestimation (in percent)?" We have added a discussion of post-deposition effects (see above) and the potential associated loss of amplitude. "ECHAM5-wiso systematically underestimates the d-excess mean seasonal amplitude when compared with precipitation data, while it systematically overestimates it when compared with ice core data (from 9.4 to 15.5 ‰, with the exception of the GIP ice core. Again, we cannot rule out a loss of amplitude in ice core data compared to the initial precipitation signal, due to the temporal resolution and to post-deposition effects."

"pg 19 lines 26-27, pg 20 lines 1-2: "ECHAM5-wiso always underestimates seasonal

amplitude of $\delta$ 18O and d-excess in precipitation but always overestimates seasonal amplitude of $\delta$ 18O and d-excess in firn/ice cores (Table 4 and 8). Differences between the model and firn/core data might be due to diffusion processes, but no clear reason can be given for the other isotopic biases." - it is not accurate to say "might be due to diffusion", because diffusion must have a substantial effect" We agree that the potentiality of the effect of diffusion is inappropriate here. We thus turned the sentence to: "Differences between the model and firn/core data are at least partially due to diffusion processes, but no clear reason can be given for the other isotopic biases."

[revised manuscript text omitted]

Please also note the supplement to this comment:
https://www.clim-past-discuss.net/cp-2017-118/cp-2017-118-AC3-supplement.pdf

[Figure]

Fig. 1 : Ratio of the three first seasonal amplitude average by the mean seasonal amplitude in function of the mean accumulation

**Fig. 1.**

**Supplement:**

S1: Mean of the three first seasonal amplitudes ("3-points mean seasonal amplitude", in ‰), mean seasonal amplitude (in ‰), ratio of the mean of the three first seasonal amplitudes by the mean seasonal amplitude over the whole available covered period of ice cores dated at the sub-annual scale ("ration"), and yearly mean accumulation ("Accumulation", in cm w.e. y-1). Blank cells remain in the "Accumulation" column when no provided.

| | 3-points mean seasonal amplitude (‰) | Mean seasonal amplitude (‰) | Ratio | Accumulation (cm w.e. y-1) |
|---|---|---|---|---|
| USITASE-2000-1 | 6.46 | 3.92 | 1.65 | 14 |
| USITASE-2000-2 | 8.43 | 4.27 | 1.98 | 26 |
| **USITASE-2000-3** | **7.18** | **7.78** | **0.92** | |
| USITASE-2000-4 | 97.98 | 61.04 | 1.61 | 19 |
| USITASE 2000-5 | 4.55 | 4.33 | 1.05 | 12 |
| **USITASE-2000-6** | **2.40** | **5.64** | **0.43** | |
| USITASE-2001-1 | 4.23 | 2.55 | 1.66 | 26 |
| USITASE-2001-2 | 9.80 | 6.73 | 1.46 | 43 |
| USITASE-2001-3 | 11.02 | 6.80 | 1.62 | 33 |
| USITASE-2001-4 | 67.36 | 46.87 | 1.44 | 19 |
| **USITASE-2001-5** | **5.88** | **6.31** | **0.93** | **38** |
| USITASE-2002-1 | 11.06 | 6.13 | 1.80 | |
| USITASE-2002-2 | 6.88 | 3.83 | 1.80 | |
| **USITASE-2002-4** | **5.68** | **6.40** | **0.89** | **15** |
| NUS 08-7 | 6.65 | 5.15 | 1.29 | |
| NUS 07-1 | 6.09 | 3.00 | 2.03 | |
| **WDC06A** | **2.00** | **2.15** | **0.93** | |
| IND25 | 90.59 | 42.22 | 2.15 | 28 |
| GIP | 5.50 | 5.44 | 1.01 | 112 |

---

## Author Response (AR1)

Dear editor,

We thank you for your positive appreciation of our responses to the reviewers of our submitted manuscript, and are ready to upload the revised manuscript that we have prepared and which addresses all comments from the reviewers.

5

Here are our answers to your remarks:

- For the distribution of our database, we propose to upload a zipped file on the PANGAEA data publisher web site with (i) one file consisting of information data for each site (e.g. latitude, longitude, covered period, resolution) and processed isotopic data (mean, standard deviation); (ii) one file with raw ice core data at sub-annual scale; and (iii) one file with precipitation data.

10

- We chose to compare the first three seasonal cycles to the mean seasonal amplitude as the smallest period covered by one of our sub-annual ice core records is 7 years. This choice of three years is a compromise between the length of the existing records, the year-to-year variability of the seasonal amplitude, the variability of the accumulation rates, and the expected loss of amplitude in the upper part of the firn.

15

- Finally, we have revised the captions of Tables 6 to 8 to make them shorter.

We hope that you will enjoy the revised manuscript, which has been significantly improved thanks to the constructive comments of our reviewers.

20  Best regards,
Sentia Goursaud

---

## Author Response (AR2)

Dear Editor,

We thank you for your meticulous reading of our manuscript, and for your editorial suggestions.

5     We have corrected the English wordings when asked to, and reworded the unclear sentences. We have clarified, at this end of Section 2.1.3 presenting our isotopic database, that it is available on the PANGAEA data archive. We have added an acknowledgement section. Finally, we have gathered means and standard deviations in the same column in all tables.

We hope that this revised version will be appropriate for publication in Climate of the Past.

10

Best regards,

Sentia Goursaud